# Three-dimensional ultrafast charge-density-wave dynamics in CuTe

Nguyen Nhat Quyen[1], Wen-Yen Tzeng[2], Chih-En Hsu [3], I-An Lin[3], Wan-Hsin Chen[1], Hao-Hsiang Jia[4], Sheng-Chiao Wang[1], Cheng-En Liu[1], Yu-Sheng Chen[4], Wei-Liang Chen[5], Ta-Lei Chou[5], I-Ta Wang[5], Chia-Nung Kuo[6], Chun-Liang Lin [1], Chien-Te Wu[1,7], Ping-Hui Lin[4], Shih-Chang Weng [4], Cheng-Maw Cheng [4], Chang-Yang Kuo[1,4], Chien-Ming Tu[1,8,9], Ming-Wen Chu [5,10], Yu-Ming Chang[5,10], Chin Shan Lue[6,11] ✉, Hung-Chung Hsueh [3] ✉ & Chih-Wei Luo [1,4,11,12,13] ✉

Charge density waves (CDWs) involved with electronic and phononic subsystems simultaneously are a common quantum state in solid-state physics, especially in low-dimensional materials. However, CDW phase dynamics in various dimensions are yet to be studied, and their phase transition mechanism is currently moot. Here we show that using the distinct temperature evolution of orientation-dependent ultrafast electron and phonon dynamics, different dimensional CDW phases are verified in CuTe. When the temperature decreases, the shrinking of $c$-axis length accompanied with the appearance of interchain and interlayer interactions causes the quantum fluctuations (QF) of the CDW phase until 220 K. At $T < 220$ K, the CDWs on the different $ab$-planes are finally locked with each other in anti-phase to form a CDW phase along the $c$-axis. This study shows the dimension evolution of CDW phases in one CDW system and their stabilized mechanisms in different temperature regimes.

Low-dimensional materials have been the subject of many studies because they feature an unusual quantum state of charge density waves (CDWs), which renders them suited to applications in switchable electronics[1,2]. The CDW materials include quasi-1D blue bronzes $(K_{0.3}MoO_3)$[3], two-dimensional (2D) transition metal dichalcogenides (TMDCs) ($2H$-TaSe$_2$[4], 3R-Ta$_{1+x}$Se$_2$[5], NbSe$_2$[6]), rare-earth tritellurides $R$Te$_3$ family ($R$ = Y, La-Sm, Gd-Tm)[7] and the three-dimensional (3D)-CDW material 1$T$-TiSe$_2$[8]. Recently, the CDW transition temperature has significantly increased in NbSe$_2$ as dimensionality is reduced from 3D

bulk to a monolayer 2D material[9], which implies that the CDW phase transition is sensitive to dimensionality. For an ideal 1D system, the Peierls model shows that the change of periodicity of the chain atoms and band gap opening at the zone boundary of a new unit cell that contains more atoms will reduce the energy of this unstable system[10]. The Fermi surface nesting effect, whereby segments of the Fermi contours are connected by $q_{CDW}$, screens phonons to induce the Kohn anomalies[11] at the $q_{CDW}$ point for phonon dispersion and softens the coherent lattice vibrations, which is the collective mode of phonons.

[1]Department of Electrophysics, National Yang Ming Chiao Tung University, Hsinchu 30010, Taiwan. [2]Department of Electronic Engineering, National Formosa University, Yunlin 632, Taiwan. [3]Department of Physics, Tamkang University, New Taipei City 251301, Taiwan. [4]National Synchrotron Radiation Research Center, Hsinchu 30076, Taiwan. [5]Center for Condensed Matter Sciences, National Taiwan University, Taipei 10617, Taiwan. [6]Department of Physics, National Cheng Kung University, Tainan 70101, Taiwan. [7]Physics Division, National Center for Theoretical Sciences, Taipei, Taiwan. [8]Undergraduate Degree Program of Systems Engineering and Technology, National Yang Ming Chiao Tung University, Hsinchu 30010, Taiwan. [9]Chung Cheng Institute of Technology, National Defense University, Taoyuan 335009, Taiwan. [10]Center of Atomic Initiative for New Materials (AI-MAT), National Taiwan University, Taipei 10617, Taiwan. [11]Taiwan Consortium of Emergent Crystalline Materials (TCECM), National Science and Technology Council, Taipei 10601, Taiwan. [12]Institute of Physics and Center for Emergent Functional Matter Science, National Yang Ming Chiao Tung University, Hsinchu 30010, Taiwan. [13]Department of Physics, University of Washington, Seattle, Washington 98195, USA. ✉e-mail: cslue@ncku.edu.tw; hchsueh@gms.tku.edu.tw; cwluoep@nycu.edu.tw

Many recent theories and experiments, however, are incompatible with this simple 1D-CDW scenario, and its nature still needs to be clarified[12–14].

In correlated electron systems, the order parameter instability with spontaneously broken symmetry (SBS) has gained attention recently. Owing to the interaction among the charge, spin, orbital, and lattice degrees of freedom, the symmetry-breaking ground state, e.g., charge/spin order and superconductivity, is caused. Usually, the fluctuations in the SBS state exhibit a collective oscillation with the modes of phase (so-called Nambu-Goldstone mode or phason)[15] or amplitude (so-called Higgs amplitude mode or amplitudon)[16]. As shown in Supplementary Fig. 1, the phason is associated with the in-phase motion of ions, which causes the phase ($\phi$) modulation of CDW ($\rho$), and the phase ($\phi$) changes with the same $V$ in the minimum of Mexican hat-like (double-well) potential[17]. The amplitudon is associated with the out-of-phase motion of ions, which causes the amplitude ($\Delta$) modulation of CDW ($\rho$) and the changes of $V$ with the same $\phi$ in the double-well potential. As a result, the phase and amplitude modes are IR- and Raman-active modes, respectively[18].

Regarding CDW materials, the vulcanite-phase CuTe[19] (with a gap of ~190 meV)[20] is structurally simple. It consists of Te-Cu-Te that is stacked in layers with a configuration of Te chains-puckered square Cu network-Te chains (see Fig. 1), which could be used to reveal the CDW phases with various dimensions simultaneously in one system[21] and the origin of CDW phase transitions. All thermal transport measurements - specific heat, thermal conductivity, and the Seebeck coefficient - demonstrate remarkable features near $T_c = 335$ K,[22] which show electron-phonon coupling in CuTe[23]. Using density functional theory calculations, Kim et al.[24] proposed that the Coulomb correlation of Cu-$3d$ electrons is essential for activating the CDW phase transition. However, this scenario has been revisited and challenged recently by more careful calculations, including quantum anharmonicity.[25] Besides, the electronic anisotropy in CuTe is revealed by infrared reflectivity measurements[26]. Two distinct CDW phases with different mechanisms and superconductivity were recently discovered in a pressurized CuTe[27]. There is debate over the interplay between electronic correlation and electron-phonon interaction, and a possible CDW phase diagram in CuTe. Further studies of their dynamics are required to determine the CDW phases with various dimensions and the origin of CDW phase transitions. Additionally, time-resolved ultrafast spectroscopy allows direct observation of transient interactions between quasiparticles to determine the strength of the interaction between different degrees of freedom[28,29]. There is a unique characterization ability in the time domain, so time-resolved ultrafast spectroscopy is promising to disentangle various spectrally overlapping components (excitations) based on their different temporal characteristics and to study the dynamics of collective modes in CDW materials[4,30–39] (see Supplementary Note 1).

Here, we show that this study uses orientation-/time-resolved ultrafast spectroscopy (otrUS) in (001) CuTe single crystals (see Fig. 1) to simultaneously elucidate electron/collective mode dynamics in three dimensions and the origin of CDW phase transitions. At temperatures below 280 K, the oscillation components in transient reflectivity change ($\Delta R/R$) spectra are observed on the *ab*-plane. These are associated with the formation of collective modes. The electron relaxation time extracted from $\Delta R/R$ spectra diverges near 320 K because the CDW gap opens, significantly higher than the onset temperature (280 K) for collective modes along the *a*-axis[28,33–35]. STM mapping shows the spatial modulation of CDW with $q_{CDW} = 0.4a^*$. Moreover, a CDW along the *c*-axis was first observed at low temperatures using otrUS and verified by first-principles calculations for a lateral shift between two neighboring CDW planes.

## Results and discussion
### Ultrafast dynamics of CDW orders

Figure 1 (left part) shows the typical $\Delta R/R$ spectra on the CuTe *ab*-plane, a composite of the damped oscillations and the exponential decay background. To figure out the correlation between the oscillation components in the $\Delta R/R$ and CDW phases, pump-probe measurements were made along the *a*- and *b*-axes (see Supplementary Fig. 3a) at various temperatures, as shown in Fig. 2a. After pumping by 3.1-eV photons, there is bleaching (reduction of the electronic population) around the Fermi surface ($E_F$), which causes a positive change in $\Delta R/R$ for 1.55-eV probing photons at zero delay time to reveal the variations of physical characteristics associated with the changes of $E_F$. Since electrons have a small heat capacity and most photons from the pumping pulses are absorbed by the electrons around $E_F$, the electron temperature ($T_e$) quickly increases to thousands of K, much higher

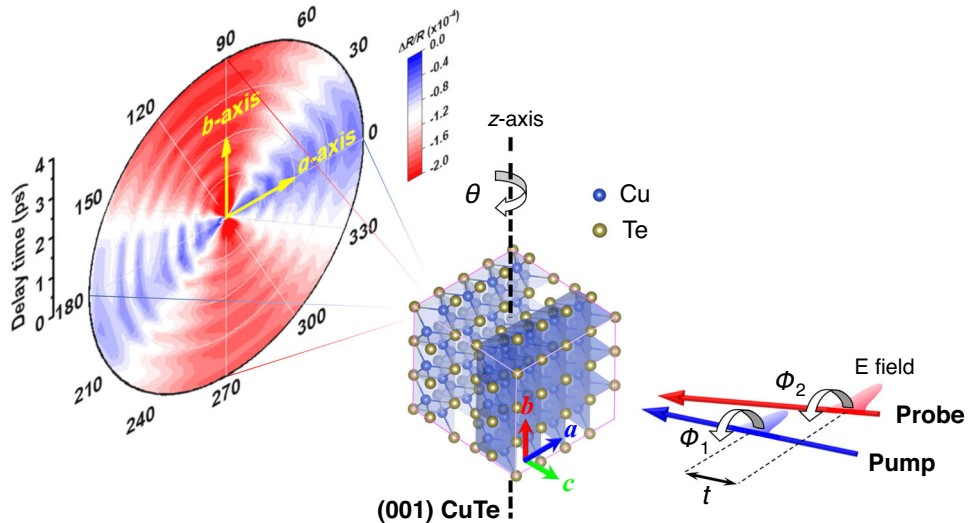

**Fig. 1 | Experimental scheme and orientation-resolved ultrafast spectra for CuTe.** Left: Orientation-dependent transient reflectivity change ($\Delta R/R$) spectra on the *ab*-plane of a CuTe single crystal from $\phi_1$ ($\phi_2$) = 0° (*a*-axis) to 90° (*b*-axis) at 37 K. Right: Crystal structure of CuTe and schematics of the orientation-/time-resolved ultrafast spectroscopy. *a*: *a*-axis. *b*: *b*-axis. *c*: *c*-axis. *t* is the delay time between the pump and probe pulses. $\theta$ is the angle between the *c*-axis of a (001) CuTe single crystal and the incident direction of the probe beam. $\phi_1$ and $\phi_2$ are the angles between the *a*-axis of a (001) CuTe single crystal and the respective polarization of the pump and probe pulses.

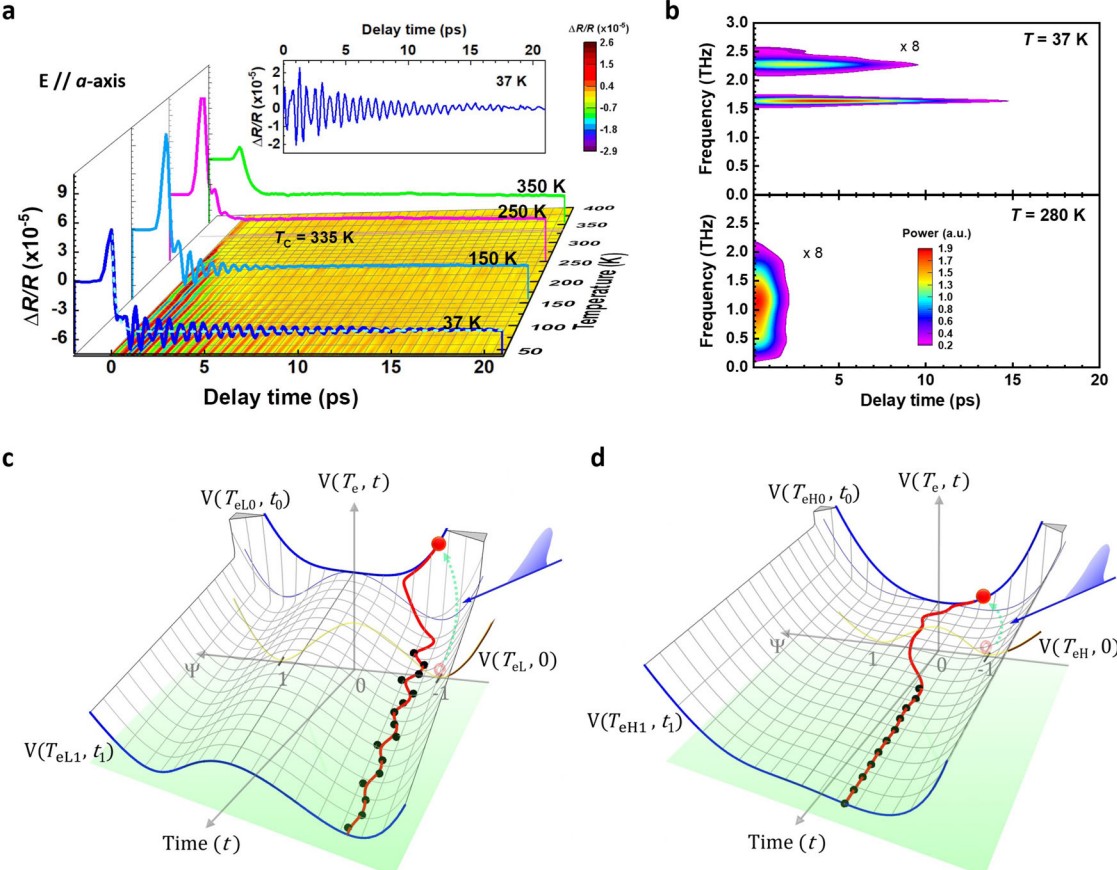

**Fig. 2 | Ultrafast dynamics at various temperatures. a** $\Delta R/R$ along $a$-axis ($\theta = 8°$, $\phi_1$ and $\phi_2 = 0°$) of a CuTe single crystal at various temperatures. Inset: a typical oscillation component of $\Delta R/R$ spectra at 37 K after subtracting the decay background (dashed line fitted by Eq. (1)). **b** Spectrogram of the oscillation components at 37 K and 280 K in (**a**) after short-time Fourier transformation. **c, d** Simulated the oscillation components of $\Delta R/R$ spectra (black and solid circles are experimental data in (**a**)) at low (e.g., $T_{eL} = 37$ K) and high (e.g., $T_{eH} = 280$ K) temperatures in

(**a**) using the TDGL equation, Supplementary Eq. (1). $V(T_e, t)$: the ground-state double-well potential as a function of electron temperature $T_e$ and delay time $t$. $V(T_{eL0(eH0)}, t_0)$: the high-symmetry state at zero delay time $t_0$ for lower (higher) electron temperature $T_{eL0}$ ($T_{eH0}$) after pumping. $V(T_{eL1(eH1)}, t_1)$: the double-well potential at delay time $t_1$ for lower (higher) electron temperature $T_{eL1}$ ($T_{eH1}$) after relaxation.

than the initial lattice (sample) temperature. According to the three-temperature model[40], the hot-electron temperature decreases within several hundreds of fs via the electron-phonon coupling between hot electrons and phonons; meanwhile, hot-electron energy is transferred to phonons so their temperature ($T_{ph}$) increases on the same time-scale. This results in the melting of the CDW phase in CuTe (at a lower excitation fluence, the CDW phase is not completely destroyed and only causes damped oscillations around the equilibrium points of the order parameters[41]), which is obtained by subtracting the exponential decay background (dashed line in Fig. 2a), as shown in the inset of Fig. 2a. The decay background is expressed as:

$$\frac{\Delta R}{R} = A_e e^{-t/\tau_e} + A_{ph}\left(1 - e^{-t/\tau_{ph,r}}\right)e^{-t/\tau_{ph}} \qquad (1)$$

where $A_e$ and $\tau_e$, respectively, denote the number and relaxation time for photoexcited electrons. The second term describes the dynamics of phonons with an initial number $A_{ph}$, a corresponding rise time $\tau_{ph,r}$, and a decay time $\tau_{ph}$. Finally, hot electrons and phonons, that reach thermal equilibrium at $T_e = T_{ph}$, simultaneously transfer their energy to the lattice and increase the lattice temperature $T_L$. During this stage (after ~1.5 ps), the CDW phases appear and recover with the so-called collective mode oscillation, which is a fingerprint of CDW orders and is detected by the 1.55-eV photons via changes in the dielectric constant due to the dynamics of CDW orders[30].

Figure 2b shows a spectrogram of the damped oscillation in the inset of Fig. 2a, obtained using a short-time Fourier transformation. There are two collective modes with frequencies of 1.64 and 2.26 THz at 37 K, but only one collective mode with a much broader width and short lifetime is observed at 280 K. To determine the nature of collective modes in CuTe, the time-dependent Ginzburg Landau (TDGL) equations[30,40,42] are used to quantitatively analyze the experimental data in Fig. 2c, d (see Supplementary Section S5). After excitation by a pump pulse, the electrons are excited from a double-well potential (DWP) $V(T_{eL}, 0)$ at a low electron temperature $T_{eL}$ (e.g., 37 K) to the shallow single-well potential (SWP) $V(T_{eL0}, t_0)$ at a higher electron temperature $T_{eL0}$ (probably thousands of K), which is slightly higher than the melting temperature of the CDW phase. SWP $V(T_{eL0}, t_0)$ then quickly relaxes back to a DWP with shallow wells; meanwhile, hot electrons drop from the edge of a SWP to one well of a DWP. Finally, hot electrons begin to oscillate inside one well of a DWP $V(T_{eL1}, t_1)$ and gradually cool, accompanied by the evolution of DWP $V(T_{eL}, t)$ as a function of time. The simulation results (red-solid lines in Fig. 2c, d) fit the experimental data (black-solid dots in Fig. 2c, d obtained from Fig. 2a) very well for low- and high-temperature cases ($T_{eL} = 37$ K, $T_{eH} = 280$ K).

**Temperature evolution and spatial modulation of CDW orders**

Figure 3a shows the frequencies of two collective modes from Fig. 2b at various temperatures. These modes significantly shrink and

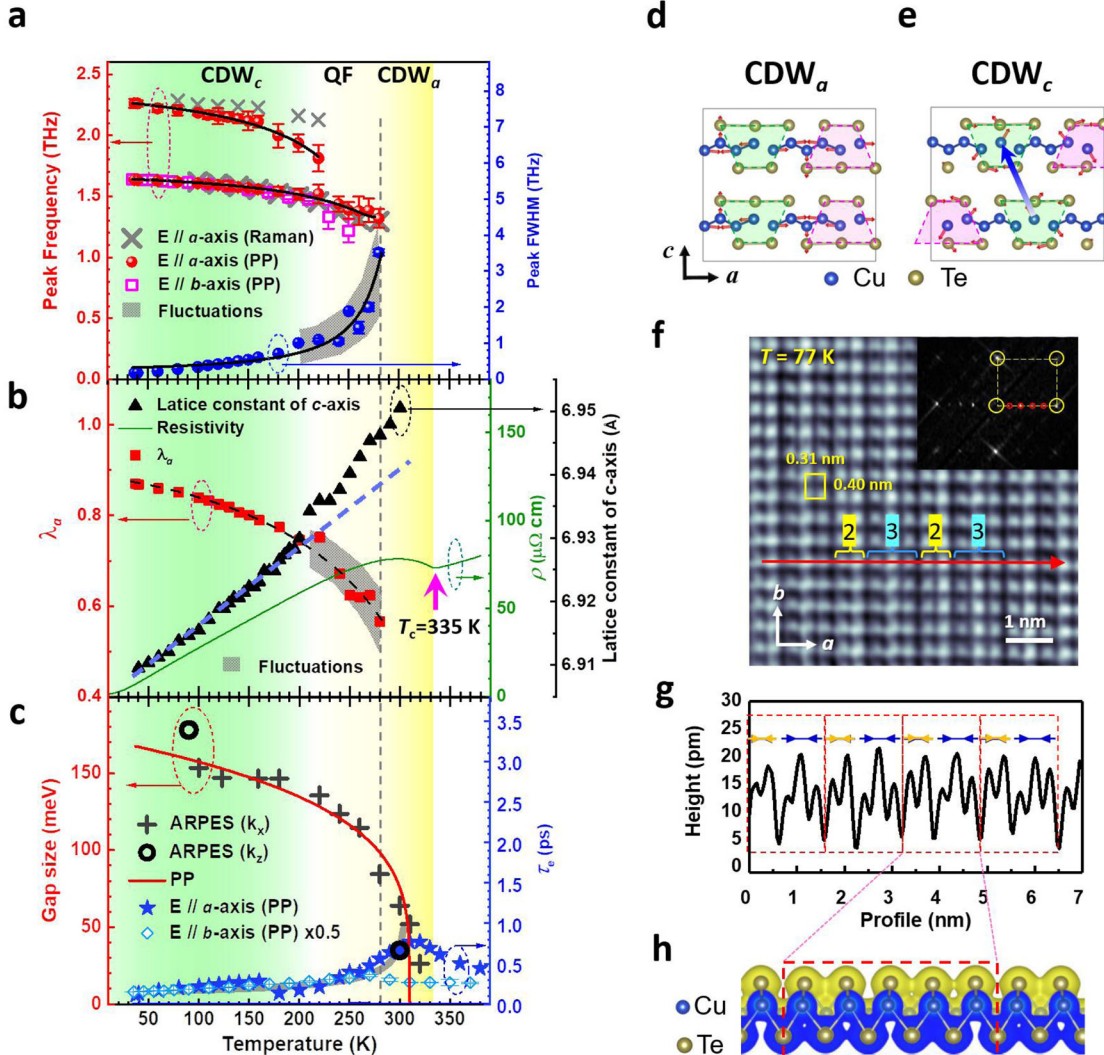

**Fig. 3 | Temperature evolution and spatial modulation for CDW orders.**
**a** Frequency/full width at half maximum (FWHM of 1.64-THz peak at 37 K) for peaks in the Fourier transform spectra in Fig. 2a (for E//b-axis from Supplementary Fig. 3) and Raman modes (obtained from Supplementary Fig. 4) as a function of temperature. Solid lines show the fitting results using the TDGL equation, Supplementary Eq. (2). QF: quantum fluctuations. PP: pump-probe spectra. The shaded area marks the fluctuations in the data. The error bars are obtained from the standard deviation of the least square fitting. **b** Electron-phonon coupling constant $\lambda_a$ (solid squares) as a function of temperature, which is derived by fitting the data in (**a**) with Supplementary Eq. (2). The solid line represents the resistivity of CuTe as a function of temperature. Solid triangles show the temperature-dependent lattice constant for the c-axis of CuTe. Dashed lines are guides for the eyes. The shaded area marks the fluctuations in data. **c** Relaxation time $\tau_e$ for photoexcited electrons as a function of temperature and obtained by fitting the $\Delta R/R$ spectra in Fig. 2a

(Supplementary Fig 3a for b-axis) with Eq. (1). CDW gap size as a function of temperature. PP: obtained by fitting the $\tau_e$ data with the mean-field-like model[49] (a gray-thick line). ARPES ($k_x$): obtained from the ARPES images in Supplementary Fig. 5. ARPES ($k_z$): obtained from the ARPES images in Supplementary Fig. 6. **d, e** Modulated structure and eigenvectors in the CDW along the a-axis ($CDW_a$) and CDW along the c-axis ($CDW_c$) that is proposed by first-principles simulations. **f** High-resolution STM image of as-cleaved CuTe surface ($V_s = 300$ mV, $I_t = 6.79$ nA) in real space. The yellow rectangle indicates a unit cell of CuTe. Inset: 2D fast Fourier transform image of the STM image in (**f**). Yellow circles mark the points of a $1 \times 1$ unit cell, and the red circles represent the $5 \times 1$ CDW modulated superstructure. **g** The line profile along the red-dashed arrow in (**f**). **h** Charge density isosurface (with a value of 0.032 e/bohr³) of a CuTe layer building block, which is obtained using first-principles calculations.

broaden as temperature increases. The 2.26-THz and 1.64-THz modes disappear at temperatures above 220 K and 280 K, respectively. The onset temperature of 280 K for the 1.64-THz mode is similar to the onset temperature (~280 K) for the metallic state and significantly less than $T_c = 335$ K, which is determined by the transport measurements in Fig. 3b. This characteristic temperature of 280 K is also observed in the temperature-dependent Seebeck coefficient $S$ (i.e., the sign-changing temperature) and the lattice thermal conductivity $\kappa_L$ (i.e., the max-$\kappa_L$ temperature in the high-temperature region)[22]. This implies that there is a coupling between the electronic and lattice subsystems. The 1.64-THz mode along the a-axis also softens dramatically as the temperature approaches $T_c$ from the low-

temperature side, which is consistent with the results for the quasi-1D $K_{0.3}MoO_3$[43,44] and rare-earth tri-tellurides $R\mathrm{Te}_3$ ($R$ = Ho, Dy, Tb)[45]. This shows that the 1.64-THz mode is a typical amplitudon (amplitude mode) associated with the $A_1$ ($A_{am}$) mode, as verified by first-principles calculations (see Fig. 3d & Supplementary Note 1 and 6, the arrows indicate the motion of atoms to form the amplitudon) and the Raman spectra in Fig. 3a (see Supplementary Note 3). Besides, this 1.64-THz amplitudon is also observed along the b-axis at temperatures below 250 K, as shown by the open squares in Fig. 3a. This implies that there is interchain coupling between neighboring amplitudons along the a-axis due to fluctuations in the CDW order[46,47], which is visualized using STM in Fig. 3f.

The STM image clearly shows the $1 \times 1$ unit cell of 0.31 nm $\times$ 0.40 nm, enclosed by the yellow rectangle in Fig. 3f[19]. The line profile along the red-solid arrow at the bottom of Fig. 3f has another periodicity. The topmost Te atomic rows are modulated and present groups with five Te atomic rows, as denoted by the red-dashed rectangle in Fig. 3g. Each group contains a basis of two atomic rows (i.e., the number "2" in Fig. 3f), followed by a base of three atomic rows (i.e., the number "3" in Fig. 3f). The yellow and blue arrows indicate the displacement of each basis of atomic rows, respectively. This specific periodicity is also present in the 2D fast Fourier transform (FFT) image, as shown in the inset of Fig. 3f. The yellow-dashed rectangle is the reciprocal lattice of the $1 \times 1$ unit cell, and the red circles indicate another $5 \times 1$ periodicity. Namely, the CuTe single crystal at 77 K exhibits the $5 \times 1$ CDW quantum phase with a spatial modulation of $q_{CDW} = 0.4a^*$, which agrees with the charge density isosurface profile obtained using first-principles calculations (see Supplementary Note 6) and several previous predictions that use other methods[19,20,24]. In the STM image, the 5-Te-atoms modulation chain along the $a$-axis is repeated well along the $b$-axis. It demonstrates a CDW plane formed via the interchain coupling, as verified by the otrUS along the $b$-axis in Fig. 3a.

The 2.26-THz mode is eliminated at $T > 220$ K. First-principles calculations show that the 2.26-THz mode is associated with a new $A_g$ ($A_{ZF1}$) mode (see Fig. 3e, Supplementary Fig. 4 & Supplementary Note 6, the arrows indicate the motion of atoms to form the amplitudon), for which the atomic configuration is different to that of the 1.64-THz mode. There is a lateral shift between two neighboring layers, as shown by the thick arrow in Fig. 3e. This indicates that there is an additional interaction between two adjacent layers along the $c$-axis to form a CDW along the $c$-axis (CDW$_c$) at $T < 220$ K with the atomic configuration shown in Fig. 3e. The dimension transition from a CDW along the $a$-axis (CDW$_a$) to CDW$_c$ is also accompanied by a reduction in the $c$-axis shrinking rate as temperature decreases, as shown in Fig. 3b (solid triangles). Namely, the lattice along the $c$-axis becomes more stable and rigid (lower shrinking rate) as the interval between two neighboring layers (CDW planes) decreases. Sufficient interlayer interactions between two neighboring CDW planes establish a more stable CDW at $T < 220$ K, consistent with XRD results in Supplementary Fig. 14a. Recently, Kuo et al.[22] reported that the temperature-dependent specific heat $C_p$ for CuTe is fitted well by the lattice-specific heat at $T < ~ 220$ K, so the CDW phase at low temperatures is mainly determined by the lattice (or atomic structure). Therefore, the CuTe first suffers dimensional fluctuations (from CDW$_a$ to CDW$_c$ via the formation of a CDW plane on $ab$-plane) in the intermediate temperature region of 220-280 K, in which the peak width of the 1.64-THz mode fluctuates significantly, as denoted by the gray area and QF in Fig. 3a. CuTe then forms a stable CDW$_c$ at $T < 220$ K.

To gain further insights into the nature of amplitudons in CuTe, the TDGL equations[30,40,42] are used to quantitatively analyze the data in Fig. 3a (see Supplementary Note 5). Through the equation of $\omega_{AM}(T) = \sqrt{\lambda}\omega_0$ [40] (where $\omega_0$ is obtained from the solid-line-fitting in Fig. 3a), the electron-phonon coupling constant of $\lambda$ is derived as a function of temperature (red-solid squares in Fig. 3b) using the data of $\omega_{AM}(T)$ in Fig. 3a. At $T = 37$ K, the electron-phonon coupling constant along the $a$-axis is $\lambda_a = 0.87$, which is < 1 and is consistent with the results that are obtained from the Raman spectra[23]. This result is in good agreement with those of previous studies: $\lambda = 0.5$ (with a bare critical phonon frequency of $\omega_0 = 2.2$ THz) in LaTe$_3$ single crystals[40] and $\lambda = 0.80$ (0.85) in 1T-TiSe$_2$ (1T-TaS$_2$)[48]. This electron-phonon constant $\lambda_a$ decreases dramatically as the temperature increases and becomes zero at around 290 K (i.e., the temperature with the greatest resistivity and significantly less than $T_c$). This study proposes that electron-phonon coupling does not dominate the formation of the CDW phase just at $T_c$. The mechanism that drives the CDW phase transition at $T_c$ must be determined.

Figure 3c shows the relationship between temperature and relaxation time $\tau_e$ for photoexcited electrons, which is obtained by fitting the pump-probe spectra in Fig. 2a with Eq. (1). The value for $\tau_{e,a}$ along the $a$-axis suddenly increases at just less than $T_c$ and reaches maximum value within the insulating region (the area between a dashed line and $T_c$ in Fig. 3b) due to gap opening. According to the theoretical band structure calculations, the formation of CDW$_a$ (or CDW$_c$) only partially opens gaps but not the full Brillouin Zone as predicted by Peierl's model for ideal 1D systems[10]. At $T < ~ 300$ K, which is the temperature at which the resistivity is greatest, the $\tau_{e,a}$ value gradually reduces as temperature decreases. The divergent behavior in the vicinity of 320 K is a typical phenomenon of phase transition and gap opening. Therefore, the mean-field-like model that was developed by Kabanov et al.[49] can be used to fit the temperature-dependent $\tau_{e,a}$ value (thick-solid line in Fig. 3c). The temperature-dependent gap size that is obtained by otrUS is consistent with the results for angle-resolved photoemission spectroscopy (ARPES) (see Supplementary Note 4). Additionally, the reduction of the density of states around $E_F$ and along the $b$-axis[26] (pseudogap-like feature[4]), caused by a certain ordering due to the interchain coupling of CDW chains along the $a$-axis, further results in a slight increase of $\tau_{e,b}$ (along the $b$-axis) around 270 K.

## Electronic-driven phase transition and CDW order along the $c$-axis

The most important point delivered by Fig. 3c is that the electronic phase transition temperature indicated by $\tau_{e,a}$ is much closer to $T_c = 335$ K, significantly higher than the onset temperature of 280 K for the $a$-axis amplitudon. This implies that the CDW-ordered phase transition (from metallic to insulating states due to gap opening around $E_F$) is driven by the electronic subsystem in CuTe with a strong Coulomb correlation of Cu-3$d$ electrons[24]. While $T < 280$ K, the electronic order gradually couples with the phonon subsystem, such as the $A_{am}$ mode, through electron-phonon coupling $\lambda_a$ (see Fig. 3b) to form an amplitudon with 1.64 THz and the high-temperature CDW$_a$. This causes metallic behavior in the transport again. In the gray area (220-280 K) of Fig. 3b, the 1.64-THz amplitudons are rather unstable due to the significant fluctuations in the electron-phonon coupling constant $\lambda_a$. Finally, the appearance of 2.26-THz mode indicates that the CDW$_c$ is stabilized due to additional interlayer interaction along the $c$-axis at $T < 220$ K.

To determine whether the 2.26-THz $A_{ZF1}$ mode in Fig. 3a is associated with the CDW$_c$, pump-probe measurements were further conducted by increasing the incident angle $\theta$ of a probe beam, as shown in Fig. 1 (also Supplementary Fig. 12). In this configuration, the $\mathbf{E}$ field of probe pulses can be decomposed into two perpendicular components: $\mathbf{E}_1$ and $\mathbf{E}_2$. $\mathbf{E}_2$ is parallel to the $a$-axis, and $\mathbf{E}_1$ is parallel to the $c$-axis, which discloses the CDW along the $c$-axis, $q_c$. Figure 4a shows the $\Delta R/R$ spectra for $\theta$ from 8° to 45°. After FFT, two modes are observed for $\theta = 8°$, 22°, 30°, and 45°. Peak fitting reveals two modes for $\theta = 8°$ with frequencies of 1.61 and 2.32 THz. The high-frequency peak ($A_{HF}$) is much shorter than the low-frequency peak ($A_{LF}$): $A_{HF}/A_{LF} \sim 0.03$. As the $\theta$ increases from 8° to 45°, the $A_{HF}$ undergoes a significant redshift from 2.32 to 1.92 THz due to the variation of the propagation vector ($\mathbf{k}$) of the $\mathbf{E}_1$ field and its coupling with the zone-folded (ZF) phonon bands (see Supplementary Note 7). The increase of the ratio of $\mathbf{E}_1$ to $\mathbf{E}_2$ by increasing $\theta$ further enhances the value of $A_{HF}/A_{LF}$ to 0.52, which is 17.3 times larger than that of $\theta = 8°$. This demonstrates that the $c$-axis CDW $q_c$ can be unambiguously revealed via the coupling between $q_c$ and $\mathbf{E}$ field along the $c$-axis. Moreover, the width of the $A_{HF}$ mode is wider than that of the $A_{LF}$ mode to show a shorter lifetime, which is consistent with the result in Fig. 2b and implies a stronger CDW interaction along the $a$-axis compared with the $c$-axis.

The CDW$_c$ structure of CuTe in real space is illustrated in Fig. 4c. At $T < 220$ K, the interlayer coupling locks two neighboring CDW planes in anti-phase to establish a CDW$_c$, as shown by the dashed

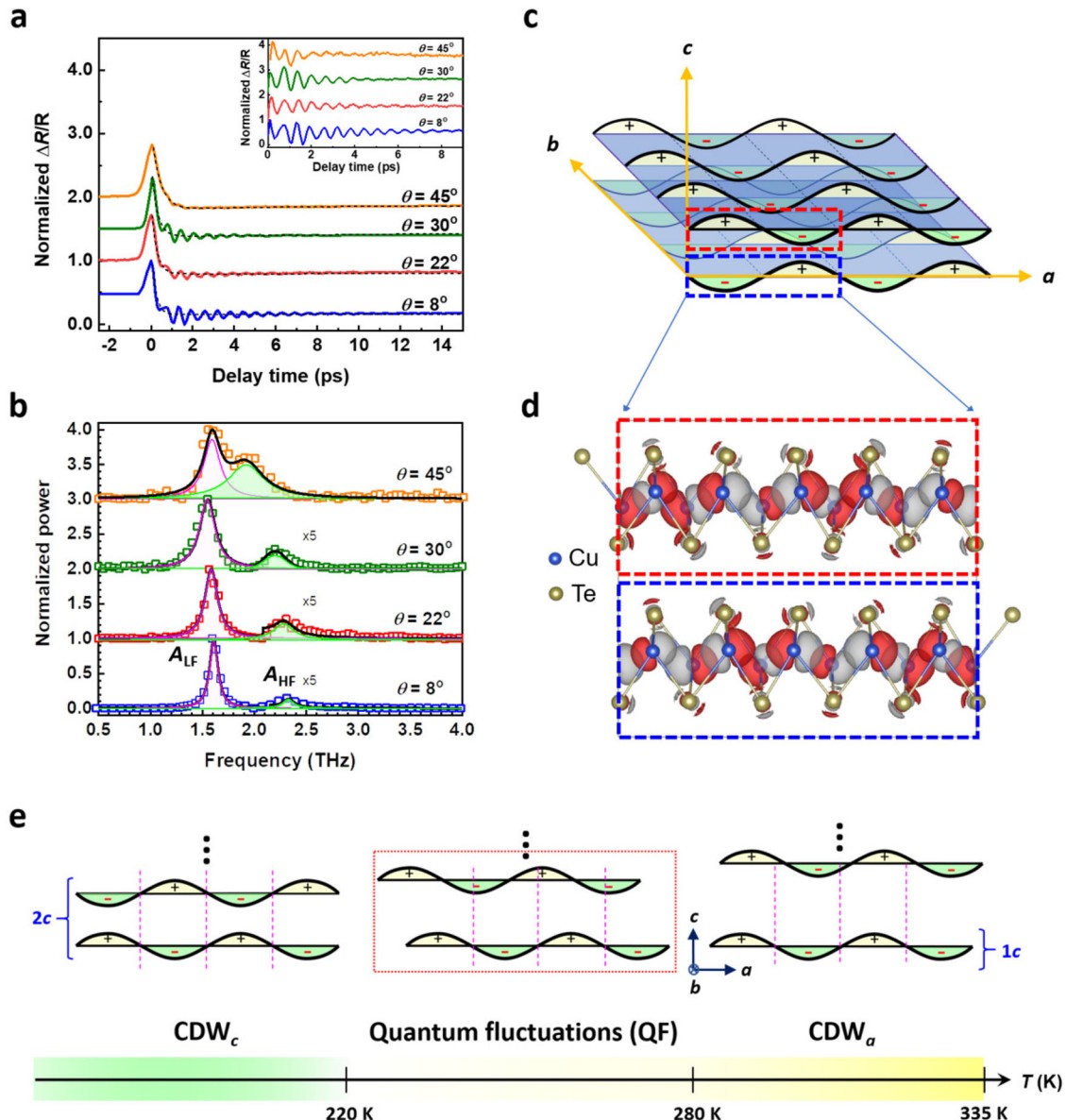

**Fig. 4 | Ultrafast dynamics along the *c*-axis of CuTe and schematics of a CDW in various dimensions. a** Normalized $\Delta R/R$ spectra of a CuTe single crystal at different incident angles, $\theta$ ($\phi_1$ and $\phi_2 = 0°$) at $T = 100$ K. Inset: the oscillation components of normalized $\Delta R/R$ spectra in (**a**) after subtracting the decay background (dashed line fitted using Eq. (1)). **b** Fourier transform spectra of the oscillation components in the inset of (**a**). **c** Schematics of a CDW in various dimensions for which two layers along the *c*-axis are anti-phase. **d** Modulated structure of the CDW in (**c**) with charge density difference between $CDW_c$ and non-CDW (normal) states obtained by first-principles calculations. An absolute isosurface value of 0.007 e/bohr$^3$ is adopted, whereas the positive and negative charge differences are, respectively, denoted as red and gray clouds (for details, see Supplementary Note 6). **e** The dimension evolution of CDW phases from high to low temperatures in CuTe.

squares in Fig. 4c. This is similar to the behavior for the 3D-CDW material, 1*T*-TiSe$_2$,[8] and is verified by first-principles calculations (see Supplementary Note 6). Figure 4d represents the modulated structure in the $CDW_c$ with charge distribution difference between non-CDW and CDW states. We emphasize that the Cu atoms dominate the charge distribution difference with anti-phase features and have a key role in the interlayer coupling between two neighboring CDW planes. The dimension evolution of the CDW phase from high to low temperatures in CuTe can be concluded in Fig. 4e. At 335 K, a $CDW_a$ along the *a*-axis forms with 1*c* periodicity. When the temperature decreases, the shrinking of the *c*-axis length (see Fig. 3b) causes the quantum fluctuations of the CDW phase in dimension (from $CDW_a$ to $CDW_c$ accompanying the formation of a CDW on *ab*-plane) until 220 K.[18,21] The CDW will be further stabilized with 2*c* periodicity and anti-phase along the *c*-axis below 220 K to form a $CDW_c$[18] due to locking the in-plane CDW phase among layers in the out-of-plane direction.

## Methods

### Sample preparation

Single crystals of CuTe were prepared using the Te-flux method. High-purity Cu shots (99.9%) and Te ingots (99.9999%) were mixed with a molar ratio of 35:65 and sealed in a quartz tube under a high vacuum. The quartz tube was then heated to 600 °C for 24 h, cooled to 410 °C for 2 h, and slowly cooled to 345 °C at a rate of 1 °C/h with a final dwell of another 72 h. Excess Te was removed by centrifugation at 345 °C, and several plate-like crystals with a typical size of $3 \times 3 \times 0.2$ mm$^3$ were mechanically removed from the quartz tube. The XRD measurements were carried out at TPS 09 A in NSRRC with an x-ray energy of 12 keV

($\lambda = 1.0332$ Å) at room temperature (the results in Supplementary Note 8). The chemical composition of the single crystals was determined using an energy-dispersive x-ray (EDX) spectroscopy (the results in Supplementary Note 8).

### Orientation-/time-resolved ultrafast spectroscopy

The orientation- and temperature-dependent ultrafast quasiparticle and collective mode dynamics in CuTe single crystals were studied using 400-nm pump and 800-nm probe spectroscopy. The light source is a Ti:sapphire laser (Femtosource scientific XL300, Femtolaser) with a central wavelength of 800 nm, a repetition rate of 5.2 MHz, and a pulse width of 100 fs. The laser pulse was split into one pump pulse and one probe pulse by a beam splitter (BS). For the pump pulses, a $\beta$-BaB$_2$O$_4$ (BBO) nonlinear crystal was used to convert the wavelength from 800 nm to 400 nm after an acousto-optic modulator (AOM), which was used to modulate the pump beam and generated a reference frequency for a lock-in amplifier[27]. A delay stage was used to adjust the pump pulses' optical path length and control the time delay between the pump and the probe pulses. The polarization (**E**) of the pump and probe pulses was controlled by adjusting the fast-axis angles of the half-wave plates in front of the samples[50]. For measurements of the orientation-dependent transient refractivity change ($\Delta R/R$) spectra, probe pulses were polarized along the *a*-axis (*b*-axis) of the CuTe crystals with $\phi_2 = 0°$ (90°) (see Fig. 1). To prevent a coherent spike around zero delay time, the pump and probe pulses were polarized perpendicular to each other (e.g., $\phi_2 = 0°$ and $\phi_1 = 90°$). Moreover, the $\Delta R/R$ spectra along the *c*-axis of the CuTe crystals were obtained by varying the incident angle $\theta$ for $\phi_2 = 0°$, as shown in Fig. 1. The typical pump and probe fluences are 32 μJ/cm$^2$ and 7 μJ/cm$^2$, respectively. The CuTe crystals were mounted inside a cryostat for temperature-dependent measurements and were cleaved using scotch tape before the measurements to ensure the surface was clean and bright. The fluctuations of laser power and sample temperatures would further cause the error in $\Delta R/R$ spectra, e.g. the error bars in Fig. 3c. The error bars in Fig. 3a are obtained from the standard deviation of the least square fitting.

### Scanning tunneling microscope (STM)

STM images were acquired at 77 K using an electrochemically etched tungsten tip after the CuTe single crystal was cleaved in situ. The base pressure of the STM chamber was maintained at $1 \times 10^{-10}$ Torr during the entire experiment.

### Temperature-dependent polarized Raman spectroscopy

The polarized Raman spectroscopy was performed by varying the sample temperature between 77 K and 300 K. The Raman spectra were curve-fitted to determine the peak frequency and the spectral linewidth as a function of temperature.

### First-principles electronic structure calculations

Based on the density functional theory (DFT), the first-principles calculations for the electronic structures of quasi-one dimensional CuTe were performed using the Quantum Espresso code (for details, see Supplementary Note 6) with ultrasoft pseudopotentials. The generalized gradient approximation in the Perdew-Burke-Ernzerhof (PBE) functional (for more information, see Supplementary Note 6) was employed to approximate the exchange-correlation interaction. An energy cutoff of 45 Ry was adopted to expand the plane-wave basis of the wavefunction. The experimental crystal structure of the primitive cell (orthorhombic, Pmmn, $a = 3.1537$ Å, $b = 4.0933$ Å, $c = 6.9621$ Å) was used. Atomic configurations of possible CDW modulations, CDW$_a$ and CDW$_c$, guided by the eigenvectors of the soft modes with calculated imaginary phonon frequencies along the $\Gamma$-X and Z-U of the normal-phase Brillouin Zone (BZ) were respectively optimized in $5 \times 1 \times 1$ and $5 \times 1 \times 2$ supercells. The convergence criteria of all atomic forces are

better than $10^{-5}$ Ry/Bohr. The k-point grid of the normal phase with a primitive unit cell ($20 \times 16 \times 8$), CDW$_a$ ($4 \times 16 \times 8$), and CDW$_c$ ($4 \times 16 \times 4$) were set to perform corresponding BZ integrations. In order to explore the lattice dynamics of CuTe, phonon dispersions of the normal, CDW$_a$, and CDW$_c$ phases were also calculated on the q grids of $10 \times 4 \times 2$, $2 \times 4 \times 2$ and $2 \times 4 \times 1$, respectively, within the framework of density functional perturbation theory (DFPT) (for details, see Supplementary Note 6). Moreover, the effects of CDW modulation on the electronic band structures and phonon dispersions were investigated by the unfolding method implemented in a modified unfold-x code (for details, see Supplementary Note 6). The wave character of the charge density in a CDW phase was clarified via the charge density difference ($\rho_{\mathrm{diff}}$) between the CDW and normal (non-CDW) phases, $\rho_{\mathrm{diff}} = \rho_{\mathrm{CDW}} - \rho_{\mathrm{normal}}$.

## Data availability

Data that support the findings of this study are presented in the main article and Supplementary Information files. Source data are provided with this paper.

## Code availability

The first-principles calculations were performed by using the Quantum ESPRESSO codes (https://www.quantum-espresso.org/). The visualizations of crystal models and Fermi surfaces were carried out by using VESTA program (https://jp-minerals.org/vesta/en/) and FermiSurfer tool (https://mitsuaki1987.github.io/fermisurfer/), respectively.

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

## Acknowledgements

This work was supported by the National Science and Technology Council of the Republic of China, Taiwan with Grant: 109-2112-M-009-020-MY3, 109-2124-M-009-003-MY3, 112-2119-M-A49-012-MBK (C.W.L.) and the Center for Emergent Functional Matter Science of National Yang Ming Chiao Tung University from The Featured Areas Research Center Program within the framework of the Higher Education Sprout Project by the Ministry of Education (MOE) in Taiwan. S.C.W. acknowledges the research support from NSTC, Taiwan, under Grant: 111-2124-M-213-001. H.C.H. acknowledges the support from NSTC, Taiwan, under Grant: 110-2112-M-032-014-MY3. The authors also thank the National Center for High-Performance Computing (NCHC) in Taiwan for providing computational resources. W.L.C. and Y.M.C. acknowledge the research support from NSTC, Taiwan, under Grant: 108-2112-M-002-013-MY3 and 109-2119-M-002-026-MY3. C.T.W. acknowledges the support from NSTC, Taiwan, under Grant: 112-2112-M-A49-042. C.Y.K. acknowledges the support from NSTC, Taiwan, under Grant: 110-2112-M-A49-002-MY3. C.S.L. acknowledges the support from NSTC, Taiwan, under Grant: 112-2124-M-006-009.

## Author contributions

C.W.L. proposed the project. N.N.Q., W.Y.T., C.M.T. and C.W.L. performed the pump-probe experimental measurements and analyzed the data. C.N.K. and C.S.L. prepared the samples. C.L.L. and W.H.C. performed the measurements of STM. C.Y.K., C.E.L., Y.S.C., S.C.W. and

C.M.C. performed the measurements of XRD. W.L.C., T.L.C., I.T.W. and Y.M.C performed the temperature-dependent Raman spectroscopy and analysis. C.T.W. performed the analysis of TDGL theory. P.H.L., H.H.J. and S.C.W. performed the measurements of ARPES. M.W.C. performed the structure characterization. C.E.H., I.A.L. and H.C.H. performed the first-principles calculations. N.N.Q. and C.W.L wrote the manuscript. All authors edited and approved the final manuscript.

## Competing interests

The authors declare no competing interests.
