## [Peer Review File · Nature Communications]

REVIEWER COMMENTS

Reviewer #1 (Remarks to the Author):

Review report for Nature, "Three-dimensional ultrafast charge-density-wave dynamics in CuTe", by Nguyen et al, submitted for review purpose.

The authors have reported on the various CDW phases in CuTe using the temperature and orientation-dependent ultrafast pump-probe and transient reflectivity measurements. They claim that CuTe has all 1D, 2D, and 3D CDW. At $T = 280$ K, author have reported the evidence of electron-phonon coupling with collective modes along the a-axis, which synchronize via an interchain interaction to establish a two-dimensional (2D) CDW phase on the ab-plane. The 2D CDW phase planes are finally locked with each other in anti-phase to form a three-dimensional (3D) CDW phase at temperatures, $T < 220$ K. The data seems interesting to understand the cause and effect of CDW. However, following are major concerns and suggestions which can be addressed before publication.

1. Page 1, Line 11: the sentence, "At 280K.....(2D) CDW phase on the ab-plane while $T < 250$ K" needs revision for the use of "while".

2. Page 2, Line 13: The description of "collective mode of phonons", is required for readers. The description of the collective modes (Amplitude, phason) is important to understand the objectives of the paper. Thus, paper required major modification with proper references on this aspects.

3. Page 2, and 3: Author mention that "11" system could be the key structure to disclose the CDW phase which describe the importance of understanding the CDW in detail. However, the system "11" is not defined clearly in the text.

4. Page 3, Line 3: Author mention that all thermal transport measurements in CuTe show, $T_c = 335$ K because of electron-phonon coupling (EPC). However, the equally probable Fermi surface nesting (FSN) also can play an important role for CDW and can be observed using resistivity and thermopower measurements. Discussion about ruling out the possibility of FSN is required.

5. Page 4, line 4, It is not clear how oscillation components in transient reflectively changes are associated explicitly with the formation of collective modes?

6. Page 4, line 7-9, Author have discussed about the 1D-CDW chain, 2D- and 3D- CDW phase but wave vector is provided only for 1D-CDW ($q_{CDW} = 0.4a^*$) chain. What will be case for 2D and 3D?

7. Page 4, line 17: Author has mentioned the “bleaching around the Fermi surface”, what is the important significance of word “bleaching” here?

8. Page 11, line 11: in description of gray area in Fig.3b, which is not visible clearly. Author may like to modify the figure accordingly.

9. Figure 3f: the red line profile seems to be very regular arrangements of atoms. The text on page 7 mentioned that the line profile has different periodicity. Perhaps quantifying the comparisons (changes in numbers) would be good, if possible.

10. The technical writing requires some careful revision. The summary is just bullets of the paper. In addition, there are several typo and unclear statements which need attentions for publications.

The paper require some revisions.

Reviewer #2 (Remarks to the Author):

In the manuscript by Quyen et al., the authors investigate the CDW phases in CuTe by employing ultrafast optical pump probe measurements in combination of first principle calculations, Raman spectroscopy, scanning tunneling microscopy. The authors report distinct temperature and polarization dependent ultrafast electron and phonon dynamics, and attribute them to the formation of 2D and 3D CDW phases.

Overall, the study is comprehensive, the experimental results are clearly presented, and the manuscript is well-organized. However, as a peer reviewer, I have concerns regarding the interpretation of the experimental results, which have led me to question whether the manuscript warrants publication in Nature Communications. Below are some major issues:

1. The authors claim the formation of 2D CDW phase below $T=250$ K and 3D CDW phase below $T=220$ K. According to the current manuscript, the main experimental evidence supporting these

claims is from the dynamics of the 1.64 and 2.26 THz collective modes. The 250 K transition is claimed simply due to the appearance of the 1.64 THz mode observed in the b-axis polarization. The 220 K transition is claimed because the 2.26 THz mode start to be observed below this temperature. However, I doubt whether these observations are overclaimed.

The CDW transition brings new phonon modes as a result of Brillouin zone folding induced by the new periodicity. However, the different onset temperatures below which new modes are observed should not be appropriate indicators of new phases. When approaching T_c from below, the modes may be quickly damped within one period, or the oscillation strength becomes too weak to be resolved. Different modes have different e-ph coupling strength, and may exhibit anisotropic optical polarization response due to the low dimensional electronic structure. Therefore, the modes may not be observed instantly below T_c and they may show different onset temperatures. In existing literature, there are many examples of onset temperatures that is below T_c , and different onset temperatures of different modes/same mode observed in different polarizations (see details in [Demsar et al., PRL 83, 800 (1999)], [Schmitt et al., Science 321, 1649 (2008)], [Lin et al., PRB 101, 205112 (2020)], [Yue et al., PRB 107, 165115 (2023)]). In these papers, the onset temperatures are not viewed as indicators of changes of CDW phases.

Based on the dynamics of the collective modes, I think one can at most say that the charge correlations in the 2nd/3rd dimensions may be enhanced below these onset temperatures. The manuscript also presents the first principle calculation results as another supporting evidence of the 2D/3D CDW phases. Without experimental evidence, the calculations can only be viewed as “theoretical predictions”. Other experimental observations presented in the manuscript, such as STM, Raman, ARPES, do not show any supportive results of the 2D/3D phases either. Based on current results, the authors can only claim there are “fingerprints of the 2D/3D CDW phases”, unless more powerful evidence of the 2D/3D CDW phases are provided.

2. In Figs. 4(a-b), the authors show that the high-frequency amplitude peak (AHF) is located at different frequencies at $\theta = 8^\circ$ and 45° , which serves as another important evidence of the 3D correlations of the charge order. Is it possible that two frequencies are actually related to two different modes which are selectively observed by the two polarizations? If they are one mode, one can expect to see the gradual shift of the AHF peak position in the FFT spectra by continuously changing θ from 8° to 45° , so the author should show such data to support their claim. In the current manuscript, the author only show data at $\theta = 8^\circ$ and 45° , which is not enough to ensure the two peaks to be related to the same mode.

Moreover, if the two peaks are confirmed to be the same mode, then I think the redshift with increasing θ is one of the most interesting results of this study. Usually, the mode intensity can vary between different measurement polarizations but the frequency should be nearly the same. The authors should talk more about this result and explain to readers the underlying mechanism why the peak frequency shifts with θ .

In my opinion, addressing the above concerns is necessary for the manuscript to meet the calibre of Nature Communications. I have additional minor comments as follows:

1. In the 2nd paragraph of the introduction part, the authors write “ This so-called 11 system is so simple that it could be the key structure to disclose the CDW phases with various dimensions

simultaneously in one system” with bold and italic font. I don’t see there is any relation between a system being “simple” and for “disclosing the CDW phases with various dimensions simultaneously in one system”. Also here no reference suggesting that CuTe may host CDW phases with various dimensions is provided. Therefore, I think this sentence is not logically reasonable at here.

2. In the paragraph before the section “Ultrafast dynamics of CDW orders”, there is a sentence “The electron relaxation time extracted from $\Delta R/R$ spectra diverges near 320 K because the CDW gap opens. This is significantly higher than the onset temperature (280 K) for collective modes along the a-axis, which indicates the electronic-driven 1D-CDW-chain phase transition in CuTe.” Does the onset temperature (280 K) really means that the 320 K transition is purely electronic driven? As I mentioned in the previous comments (in the 4th graph of this review report), the onset temperature below which the folded modes can be observed are often lower than T_c in many studies, because near T_c the modes may be quickly damped or the oscillation strength becomes too weak.

3. In Fig. S2, the ~ 2.3 THz mode is present in the Raman spectra above 220 K, even though the peaks are lower and broader than at low temperatures. This is against Fig. 3a where this mode is absent above 220 K in the plotted Raman data. The authors should explain such contradiction.

Reviewer #3 (Remarks to the Author):

The authors use pump-probe spectroscopy (transient reflectivity) and Raman spectroscopy at various temperature in order to elucidate the complete phase diagram of CuTe. In addition to the known (5,1,2) CDW (charge-density wave) phase (also called the quasi-1D CDW) whose transition from the high-symmetry phase takes place at 335K, they infer transition to a 2D CDW at 280 K and a 3D CDW at 250K. Experimental evidence is given in terms of softening of measured phonon modes.

The findings are plausible and deserve to be published. Unfortunately, the physical explanation of the driving mechanisms between the different CDW regimes remains rather obscure. In the supporting information, the authors show that two of the CDW instabilities are visible as dips or even soft modes in the phonon dispersion of the primitive (high-symmetry) unit cell of CuTe. Using this dispersion as starting point, the authors should explain why at different temperatures the different soft modes become dominant. Without a clear and understandable explanation of the driving mechanism, the paper does not meet the high criteria of importance for a general audience that one expects from nature communications.

Other comment:

The last phrase of the abstract ("This study shows that the hidden CDW phases with various dimensions and their transition mechanisms are critical for CDW materials.") does not make sense. What does it mean that the hidden CDW phases are critical for CDW materials? Furthermore, there are materials with just one CDW phase where there are no "hidden" CDW phases.

Responses to Reviewer #1:

We appreciate the Reviewer's time and suggestions in the comment letter, which did help us to polish this research work. The detailed responses to the Reviewer's concern are provided as follows.

(1) Reviewer's comments:

Page 1, Line 11: the sentence, "At 280K.....(2D) CDW phase on the ab-plane while $T < 250\text{K}$ " needs revision for the use of "while".

Responses:

Thank you for the reviewer's correction. The "while" in line 11 on page 1 (in previous manuscript) has been replaced by "at" in this revised manuscript.

(2) Reviewer's comments:

Page 2, Line 13: The description of "collective mode of phonons", is required for readers. The description of the collective modes (Amplitude, phason) is important to understand the objectives of the paper. Thus, paper required major modification with proper references on this aspects.

Responses:

Thank the reviewer for reminding us of the description of the collective mode of phonons. *The following discussion and references have been added to the 2nd paragraph on page 4 of this revised manuscript and the Supplementary Information (section S1).*

In correlated electron systems, the order parameter (OP) instability with the spontaneously broken symmetry (SBS) has gained attention recently. Owing to the interaction among the charge, spin, orbital, and lattice degrees of freedom, the symmetry-breaking ground state, *e.g.*, charge/spin order and superconductivity, is caused. Usually, the fluctuations in the SBS state exhibit a collective oscillation with the modes of phase (so-called Nambu-Goldstone mode or phasor) [1] or amplitude (so-called Higgs amplitude mode or amplitudon) [2]. As shown in Fig. R1, the phasor is associated with the in-phase motion of ions, which causes the phase (ϕ) modulation of charge density (ρ) wave, and the phase (ϕ) changes with the same V in the minimum of Mexican hat-like (double-well) potential [3]. The amplitudon is associated with the out-of-phase motion of ions, which causes the amplitude (Δ) modulation of charge density (ρ) wave and the changes of V with the same ϕ in the Mexican hat-like (double-well) potential. As a result, the phase and amplitude modes are IR-active and Raman-active modes,

respectively [4]. The presence of the amplitude mode in charge density wave (CDW) systems has been observed by time-resolved spectroscopy [5-7].

Fig. R1. Schematics of spatial modulation (left) of ions/charge density (ρ) and phase space potentials (right) associated with 1D crystal chain. **a** Phase mode (Phason) with the phase changes as a function of time, $\phi(t)$. **b** Amplitude mode (Amplitudon) with the potential changes as a function of time, $\Delta(t)$.

(3) Reviewer's comments:

Page 2, and 3: Author mention that “11” system could be the key structure to disclose the CDW phase which describe the importance of understanding the CDW in detail. However, the system “11” is not defined clearly in the text.

Responses:

Thank the reviewer for reminding us. The “11” system means that there is one cation (*e.g.*, one Cu ion in CuTe) and one anion (*e.g.*, one Te ion in CuTe) in CDW compounds. In order to avoid confusion for readers, we have removed the sentence “This so-called ”11” system is so simple that it could be the key structure to disclose” in this revised manuscript. *(line 3 in the 2nd paragraph, page 5)*

(4) Reviewer's comments:

Page 3, Line 3: Author mention that all thermal transport measurements in CuTe show, $T_c = 335$ K because of electron-phonon coupling (EPC). However, the equally probable Fermi surface nesting (FSN) also can play an important role for CDW and can be observed using resistivity and thermopower measurements. Discussion about ruling out the possibility of FSN is required.

Responses:

The authors appreciate the reviewer for raising the crucial point about the role of Fermi surface nesting (FSN) in forming the CDW phase of CuTe. To address this issue without dealing with a theoretically complicated thermodynamical process, we have performed the calculations of Fermi surfaces, a nesting factor which is the low-frequency limit of the bare electronic susceptibility, and phonon spectra of the non-CDW phase under ambient and high (9 GPa) pressure. As shown in Fig. R2ab, all the imaginary frequencies of the phonon spectrum at ambient conditions can be stabilized. Therefore, the corresponding CDW instability can be destroyed under external hydrostatic compression. However, both the peak features at CDW ordering q -vectors [$q_1 = (0.4, 0, 0)$ and $q_2 = (0.4, 0, 0.5)$ along the Γ -X and Z-U paths, respectively in Fig. R2cd] of the nesting factors (N_f) and corresponding contours of FS topologies, as illustrated in Fig. R2ef, demonstrated the nature of nesting is preserved in both non-CDW and CDW conditions. Therefore, the purely electronic FSN is not the main driving mechanism responsible for developing CDW modulations in CuTe.

The above discussion has been added to the 2nd paragraph (line 7-8) on page 5 of this revised manuscript and the Supplementary Information (section S6).

Fig. R2. The DFPT calculated phonon spectra of the non-CDW phase of CuTe at **a** ambient pressure and **b** 9 GPa. The nesting factors/Fermi surface contours at $k_z = 0$ are shown in **c/e** and **d/f** for ambient and high-pressure (9 GPa) conditions, respectively. DARK GREEN DASHED LINES INDICATE the CDW ordering q -vectors (q_1 and q_2).

(5) Reviewer's comments:

Page 4, line 4, It is not clear how oscillation components in transient reflectively changes are associated explicitly with the formation of collective modes?

Responses:

As we discussed in response to Question 2, there are two collective modes of amplitudon $\Delta(t)$ and phason $\phi(t)$ in correlated electron systems. Recently, a phenomenological time-dependent Ginzburg-Landau (TDGL) model (for the details, please see section S5 in Supplementary Information) was employed to account for the temperature dependence of the amplitudon and non-equilibrium dynamics upon photoexcitation [8]. The TDGL model describes the distortions of CDW amplitudon via classical equations of motion for the complex

coordinates of the electronic order parameter (EOP, $\tilde{\Delta}$) with linear coupling to the bare phonons ($\tilde{\xi}_n$) as shown in Fig. R3. This provides a versatile framework to include effects such as time-dependent perturbations of the potentials and higher spatial dimensions [6, 9].

In terms of pump-probe spectroscopy, the transient reflectivity changes ($\Delta R/R$) of opaque materials can be expressed by [10]

$$\frac{\Delta R}{R} = \frac{4\Delta n}{n^2-1}$$

where n is the refractive index of the materials, and "1" stands for the free-space refractive index. Δn is the pump-induced change in the refractive index n , which can be expressed as

$$\Delta n = \frac{\partial n}{\partial N} \Delta N + \frac{\partial n}{\partial T} \Delta T_L$$

where ΔN and ΔT_L are the pump-induced excess carrier density and lattice temperature change, respectively. After pumping, the ΔN of carrier density changes dominates the Δn and $\Delta R/R$ in sub-ps timescale (the rather small ΔT_L is negligible). The ΔN is then modulated as a function of time by the charge density modulation of amplitudon $\Delta(t)$ (or the oscillation of electronic band positions and intensities of folded electronic bands caused by the driven atomic motion of phonons [6]). Thus, the oscillation feature of collective modes and phonons in the time domain can be further resolved by pump-probe spectroscopy with enough time resolution and phase-matching conditions. Also, the coupling between $\tilde{\Delta}$ and $\tilde{\xi}_n$ as shown in Fig. R3, the oscillation components in $\Delta R/R$ are explicitly associated with the collective modes in CDW materials.

The above discussion has been added to the 1st paragraph (the last line) on page 6 of this revised manuscript and the Supplementary Information (section S1).

Fig. R3. Schematics of spatial modulation (left) and phase space potentials (right) associated with **a** CDW (electronic order parameter, EOP) and **b** a single phonon with complex amplitudes $\tilde{\Delta}$ and $\tilde{\xi}$, respectively.

(6) Reviewer's comments:

Page 4, line 7-9, Author have discussed about the 1D-CDW chain, 2D- and 3D- CDW phase but wave vector is provided only for 1D-CDW ($q_{\text{CDW}} = 0.4a^*$) chain. What will be case for 2D and 3D?

Responses:

Thank you for the reviewer's important comments on q -vectors of various CDW phases of CuTe. According to the diffraction results in Ref. [11], a modulated structure has been developed in the (3+1)-dimensional superspace group $Pm\bar{m}n(0.4, 0, 0.5)000$ with a modulation wave vector $q = (0.4, 0, 0.5)$ using the superspace approach. Therefore, the wave vectors of 2D- and 3D-CDW phases are $q_{\text{CDW}} = 0.4a^*$ and $q_{\text{CDW}} = 0.4a^* + 0.5c^*$, respectively.

The above discussion has been added to the 1st paragraph (line 4) on page 7 of this revised manuscript.

(7) Reviewer's comments:

Page 4, line 17: Author has mentioned the "bleaching around the Fermi surface", what is the important significance of word "bleaching" here?

Responses:

As shown in Fig. R4, the population of electrons around the Fermi surface, E_f , is significantly reduced by pump photons, which is the so-called "bleaching" effect. This is a typical phenomenon in pump-probe processes. Thus, the variations of physical characteristics associated with the changes in the Fermi surface can be exactly revealed by pump-probe spectroscopy. Moreover, the above significance of "bleaching" has been added after the word "bleaching" in the main text. *(See the 2nd paragraph (line 6) on page 7 of this revised manuscript.)*

Fig. R4. Schematics of density of state (DOS) of CuTe obtained by first-principles calculations and the pump-probe processes for **a** 3.1-eV pump at delay time $t = 0$ and **b** 1.55-eV probe at delay time $t > 0$.

(8) Reviewer's comments:

Page 11, line 11: in description of gray area in Fig.3b, which is not visible clearly. Author may like to modify the figure accordingly.

Responses:

Thank the reviewer for reminding us. Fig. 3b has been modified accordingly, which is more visible clearly in this revised manuscript (see Fig. R5).

Fig. R5. Revised Fig. 3 in this revised manuscript.

(9) Reviewer's comments:

Figure 3f: the red line profile seems to be very regular arrangements of atoms. The text on page 7 mentioned that the line profile has different periodicity. Perhaps quantifying the comparisons (changes in numbers) would be good, if possible.

Responses:

Thank the reviewer's suggestion. As mentioned by the reviewer, Fig. 3f displays a regular arrangement of atoms, representing the 1×1 unit cell (see the yellow rectangle in Fig. 3f) on the CuTe surface. However, the different periodicity referenced on page 10 involves a set of five Te atoms in a CDW period, with *two* Te atoms arranged closely (yellow arrows in Fig. 3g) and *three* Te atoms arranged closely (blue arrows in Fig. 3g) in a configuration of ...₂₋₃₂₋₃... (see Fig. 3f marked with the changes of atomic number). This phenomenon has been previously discussed in Refs. [12] and [13]. Our theoretical calculation, Fig. 3h, also demonstrates that the group of 5×1 CDW exhibits an internal charge density modulation. To clarify our statement for the reader, we have modified the manuscript (see the 2nd paragraph (line 5-7) on page 10 of this revised manuscript) and Figs. 3f and 3g by utilizing different colored arrows to assist with comprehension (see Fig. R5).

(10) Reviewer's comments:

The technical writing requires some careful revision. The summary is just bullets of the paper. In addition, there are several typo and unclear statements which need attentions for publications.

Responses:

Thank the reviewer for reminding us and suggestions. We have carefully corrected the typos and unclear statements as the yellow mark in this revised manuscript. We also accordingly revised the summary in the last paragraph of this revised manuscript.

In summary, we have addressed and implemented all points raised by the reviewer. The new manuscript version has been improved significantly and has reached the level of thoroughness and quality suitable for publication in Nature Communications.

References

- [1] Nambu, Y. Energy gap, mass gap, and spontaneous symmetry breaking. *Int. J. Mod. Phys. A* **25**, 4141 (2010).
- [2] Higgs, P. W. Broken symmetries, massless particles and gauge fields. *Phys. Lett.* **12**, 132-133 (1964).
- [3] Thomson, M. D. et al. Phase-channel dynamics reveal the role of impurities and screening in a quasi-one-dimensional charge-density wave system. *Sci. Rep.* **7**, 2039 (2017).
- [4] Grüner, G. Charge Density Wave in Solids, Frontiers in physics Ed D. Pines (Addison-Wesley) (1994).
- [5] Demsar, J., Forró, L., Berger, H. & Mihailovic, D. Femtosecond snapshots of gap-forming charge-density-wave correlations in quasi-two-dimensional dichalcogenides 1T-TaS₂ and 2H-TaSe₂. *Phys. Rev. B* **66**, 041101R (2002).
- [6] Schaefer, H., Kabanov, V. V. & Demsar, J. Collective modes in quasi-one-dimensional charge-density wave systems probed by femtosecond time-resolved optical studies. *Phys. Rev. B* **89**, 045106 (2014).
- [7] Demsar, J., K. Biljaković & Mihailovic, D. Single particle and collective excitations in the one-dimensional charge density wave solid K_{0.3}MoO₃ probed in real time by femtosecond spectroscopy. *Phys. Rev. Lett.* **83**, 800 (1999).
- [8] Dolgirev, P. E. et al. Amplitude dynamics of the charge density wave in LaTe₃: Theoretical description of pump-probe experiments. *Phys. Rev. B* **101**, 054203 (2020).
- [9] Yusupov, R. et al. Coherent dynamics of macroscopic electronic order through a symmetry breaking transition. *Nat. Phys.* **6**, 681-684 (2010).
- [10] Tanaka, T., Harata, A. & Sawada, T. Subpicosecond surface-restricted carrier and thermal dynamics by transient reflectivity measurements. *J. Appl. Phys.* **82**, 4033-4038 (1997).
- [11] Stolze, K. et al. CuTe: Remarkable bonding features as a consequence of a charge density wave. *Angew. Chemie - Int. Ed.* **52**, 862-865 (2013).
- [12] Kim, S., Kim, B. & Kim, K. Role of Coulomb correlations in the charge density wave of CuTe. *Phys. Rev. B* **100**, 054112 (2019).
- [13] Zhang, K. et al. Evidence for a quasi-one-dimensional charge density wave in CuTe by angle-resolved photoemission spectroscopy. *Phys. Rev. Lett.* **121**, 206402 (2018).

Responses to Reviewer #2:

We appreciate the Reviewer's time and suggestions in the comment letter, which did help us to clarify and polish this research work. The detailed responses to the Reviewer's concern are provided as follows.

(1) Reviewer's comments:

The authors claim the formation of 2D CDW phase below $T=250$ K and 3D CDW phase below $T=220$ K. According to the current manuscript, the main experimental evidence supporting these claims is from the dynamics of the 1.64 and 2.26 THz collective modes. The 250 K transition is claimed simply due to the appearance of the 1.64 THz mode observed in the b-axis polarization. The 220 K transition is claimed because the 2.26 THz mode start to be observed below this temperature. However, I doubt whether these observations are overclaimed.

The CDW transition brings new phonon modes as a result of Brillouin zone folding induced by the new periodicity. However, the different onset temperatures below which new modes are observed should not be appropriate indicators of new phases. When approaching T_c from below, the modes may be quickly damped within one period, or the oscillation strength becomes too weak to be resolved. Different modes have different e-ph coupling strength, and may exhibit anisotropic optical polarization response due to the low dimensional electronic structure. Therefore, the modes may not be observed instantly below T_c and they may show different onset temperatures. In existing literature, there are many examples of onset temperatures that is below T_c , and different onset temperatures of different modes/same mode observed in different polarizations (see details in [Demsar et al., PRL 83, 800 (1999)], [Schmitt et al., Science 321, 1649 (2008)], [Lin et al., PRB 101, 205112 (2020)], [Yue et al., PRB 107, 165115 (2023)]). In these papers, the onset temperatures are not viewed as indicators of changes of CDW phases.

Based on the dynamics of the collective modes, I think one can at most say that the charge correlations in the 2nd/3rd dimensions may be enhanced below these onset temperatures. The manuscript also presents the first principle calculation results as another supporting evidence of the 2D/3D CDW phases. Without experimental evidence, the calculations can only be viewed as "theoretical predictions". Other experimental observations presented in the manuscript, such as STM, Raman, ARPES, do not show any supportive results of the 2D/3D phases either. Based on current results, the authors can only claim there are "fingerprints of the 2D/3D CDW phases", unless more powerful evidence of the 2D/3D CDW phases are provided.

Responses:

Thank the reviewer's comments and references (which have been cited accordingly in this revised manuscript with Refs. [28], [33], [34], [35]). We agree with the reviewer's idea, i.e., the charge correlations in the 2nd/3rd dimensions may be enhanced below these onset temperatures. First, we must emphasize that only "one CDW phase transition" is observed in CuTe, around 335 K, as shown in Fig. 3b of the manuscript. The 250 K and 220 K indicated by the 1.64 and 2.26 THz collective modes are the onset temperatures of the dimension changes of the CDW phase that appeared below 335 K.

The green arrow in Fig. R1a shows that the ARPES images were performed along the k_z direction from Γ to Z with different photon energies (E_{ph}), which relate to different k_z points. For example, the E_{ph} was tuned from 90 eV to 106 eV (i.e., $k_z = 12.09 \pi/c$), and the band dispersions of CuTe measured along the k_y direction (momentum cut marked by a green-dashed line at $k_x = 0.41 \text{ \AA}^{-1}$ in Fig. R1b) at 90 K and 300 K, respectively. By comparison of corresponding EDC at $k_y = 0.46 \text{ \AA}^{-1}$ (a green line in Fig. R1e) and the leading edge with the reference spectrum of polycrystalline gold (a black line in Fig. R1e), a CDW gap with 178 meV can be clearly observed at 90 K (also marked by the white arrow in Fig. R1c), which is closed to the gap size along the k_x direction (i.e., along the a -axis in Fig. 3c of this revised manuscript). This unambiguously indicates that a CDW exists at low temperatures and along the k_z direction (i.e., c -axis) to show a 3D CDW. Additionally, as shown in Fig. R1f, the CDW gap significantly shrinks at 300 K to indicate the closing of the CDW gap along k_z direction (i.e., c -axis) above 220 K. Namely, the CDW along c -axis becomes weak at high temperatures even though the 1D CDW phase along the a -axis is still pronounced as shown in Fig. 3c of this revised manuscript.

In summary, the dimension evolution of the CDW phase from high temperatures to low temperatures in CuTe can be concluded in Fig. R2. At 335 K, a 1D CDW along the a -axis forms with $1c$ periodicity and in-phase along the c -axis. When the temperature decreases, the shrinking of the c -axis length (see Fig. 3b in this revised manuscript) causes the *quantum fluctuations of the CDW phase in dimension* until 220 K [1]. The CDW will be further stabilized with $2c$ periodicity and out-of-phase along the c -axis below 220 K to form a 3D CDW due to locking the in-plane CDW phase among layers in the out-of-plane direction.

The above discussion has been added to the 2nd paragraph (line 8-14) on page 15 of this revised manuscript and the Supplementary Information (section S4). Abstract and Fig. 3abc have also been adjusted accordingly (e.g., the "2D" replaced by the "QF").

Fig. R1. **a** Illustration of the bulk Brillouin zone of CuTe with high symmetry labels. **b** Fermi surface mapping in k_x - k_y plane measured at 85 K and $E_{ph} = 90 \text{ eV}$. The dashed rectangle indicates the bulk Brillouin zone with high symmetry labels. **c**, **d** Band dispersions of CuTe measured along the k_y direction (momentum cut marked by a green-dashed line at $k_x = 0.41 \text{ \AA}^{-1}$ in **b**) with $E_{ph} = 106 \text{ eV}$ (which is related to the higher k_z point along the arrow from Γ to Z in **a**) at 90 K and 300 K, respectively. **e**, **f** The comparison of corresponding EDCs at $k_y = 0.46 \text{ \AA}^{-1}$ (green lines) and the leading edge with the reference spectra of polycrystalline gold (black lines) at 90 K and 300 K, respectively, where the largest CDW gap is located. The locations of the largest CDW gaps are indicated by the white arrows in the dispersions. (*This figure has been added to section S4 (Fig. S6) of Supplementary Information.*)

Fig. R2. The dimension evolution of CDW phase from high to low temperatures in CuTe. (*This figure has been added to Fig. 4e of this revised manuscript.*)

(2) Reviewer's comments:

In Figs. 4(a-b), the authors show that the high-frequency amplitudon peak (AHF) is located at different frequencies at $\theta = 8^\circ$ and 45° , which serves as another important evidence of the 3D correlations of the charge order. Is it possible that two frequencies are actually related to two different modes which are selectively observed by the two polarizations? If they are one mode, one can expect to see the gradual shift of the AHF peak position in the FFT spectra by continuously changing θ from 8° to 45° , so the author should show such data to support their claim. In the current manuscript, the author only show data at $\theta = 8^\circ$ and 45° , which is not enough to ensure the two peaks to be related to the same mode.

Moreover, if the two peaks are confirmed to be the same mode, then I think the redshift with increasing θ is one of the most interesting results of this study. Usually, the mode intensity can vary between different measurement polarizations but the frequency should be nearly the same. The authors should talk more about this result and explain to readers the underlying mechanism why the peak frequency shifts with θ .

Responses:

Per the reviewer's suggestions, we have completed more pump-probe measurements with different incident angles, as shown in Fig. R3. The high-frequency amplitudon peak (A_{HF}) unambiguously exhibits the redshift with increasing the incident angle, which can be well described by a cos function in Fig. R4c. Additionally, the peak amplitude ratio of A_{HF} to A_{HF}

($A_{\text{HF}} / A_{\text{LF}}$) rises from 0.03, 0.05, 0.07, to 0.52 for increasing the incident angles from $\theta = 0^\circ$, 22° , 30° , to 45° , respectively.

Fig. R3. **a** Normalized $\Delta R/R$ spectra of a CuTe single crystal at different incident angles, θ (ϕ_1 and $\phi_2 = 0^\circ$, i.e., the configuration of **Fig. R4a**) at $T = 100$ K. Inset: the oscillation components of $\Delta R/R$ spectra in **a** after subtracting the decay background (dashed line fitted using Eq. (1) in this revised manuscript). **b** Fourier transform spectra of the oscillation components in the inset of **a**. (*Figs. 4a and 4b in this revised manuscript have been updated by this figure*)

Fig. R4a represents the configuration of a 3D CDW probed by a probe beam with various incident angles θ . First, in the normal incident case with $\theta = 0^\circ$, a propagation direction of probe pulses $\mathbf{k} // c$ -axis and electric field $\mathbf{E} // a$ -axis, the wavelength λ_0 of CDW along a -axis can be detected by the \mathbf{E} of probe pulses. For the case of oblique incidence, the \mathbf{E} field of probe pulses can be decomposed into two perpendicular components: $\mathbf{E}_{//}$ and \mathbf{E}_{\perp} . $\mathbf{E}_{//}$ is parallel to the a -axis, and \mathbf{E}_{\perp} is parallel to the c -axis, which discloses the contribution of the amplitudon along the c -axis. Under this configuration, the CDW along the a -axis with a wavelength of λ_0 will be detected with a wavelength of λ_θ due to the angle θ between the \mathbf{E} field and the a -axis. Here, the $\lambda_0 = \lambda_\theta \cos\theta$. Then, according to the relation of $\lambda_0 \propto 1/f_0$ ($\lambda_\theta \propto 1/f_\theta$), we can obtain $f_\theta / f_0 = \cos\theta$, which has been clearly presented in **Fig. R4c**. This technique has been applied to study the dichotomy of photoinduced quasiparticle dynamics in high T_c superconductors YBa₂Cu₃O₇ [2-4]. The d -wave symmetry of the superconducting gap and pseudogap in high T_c superconductors was successfully revealed by the distribution of the temperature evolution of $\Delta R/R$ on the CuO₂ planes of YBa₂Cu₃O₇.

Additionally, once the \mathbf{E} field of the probe pulse was set to parallel with the b -axis as the configuration in Fig. R4b, there is only ~ 1.61 -THz peak in all spectra with different incident angles (see Fig. R4d). Therefore, the A_{HF} peaks with ~ 2.31 THz observed in the incident angle-dependent spectra are polarization-sensitive and from the same mode.

The above discussion has been added to the 2nd paragraph (line 3-7) on page 14 of this revised manuscript and the Supplementary Information (section S7)

Fig. R4. **a** Schematics of the configuration of a 3D CDW probed by a probe pulse with various incident angles θ and electric field $\mathbf{E} \perp b$ -axis. \mathbf{E}_{\parallel} ($\parallel a$ -axis) and \mathbf{E}_{\perp} ($\perp a$ -axis) are the components of the electric field \mathbf{E} of a probe pulse. \mathbf{k} is the propagation direction of the probe pulses. λ_0 is the wavelength of CDW probed by a probe pulse with an incident angle of $\theta = 0^\circ$. λ_θ is the wavelength of CDW probed by a probe pulse with an incident angle of $\theta \neq 0^\circ$. **b** Schematics of a 3D CDW probed by a probe pulse with various incident angles θ and electric field $\mathbf{E} \parallel b$ -axis. \mathbf{k} is the propagation direction of the probe pulses. **c** Normalized frequency of A_{HF} peak in Fig. R3b as a function of incident angles θ of the probe pulses. A green solid line is the $\cos\theta$ function. **d** Normalized oscillation components in $\Delta R/R$ spectra of a CuTe single crystal at different incident angles, θ (ϕ_1 and $\phi_2 = 90^\circ$, i.e., the configuration in **b**) at $T = 60$ K. Inset: Fourier transform spectra of the oscillation spectra in **d**. (*This figure has been added to section S7 (Fig. 12) of Supplementary Information.*)

(3) Reviewer's comments:

In the 2nd paragraph of the introduction part, the authors write "This so-called 11 system is so simple that it could be the key structure to disclose the CDW phases with various dimensions simultaneously in one system" with bold and italic font. I don't see there is any relation between a system being "simple" and for "disclosing the CDW phases with various dimensions simultaneously in one system". Also here no reference suggesting that CuTe may host CDW phases with various dimensions is provided. Therefore, I think this sentence is not logically reasonable at here.

Responses:

Thank the reviewer for reminding us. In order to avoid confusion for readers, the sentence "... Te chains-puckered square Cu network-Te chains (see Fig. 1). This so-called "11" system is so simple that it could be the key structure to disclose the CDW phases with various dimensions simultaneously in one system and the origin of CDW phase transitions." has been revised to "... Te chains-puckered square Cu network-Te chains (see Fig. 1), which could be used to reveal the CDW phases with various dimensions simultaneously in one system [1] and the origin of CDW phase transitions." (*Line 3, the 2nd paragraph on page 5 of this revised manuscript*) This sentence describes the main idea and results of this study, which is the first finding in the world to the best of our knowledge. Consequently, there is no reference for this part. However, we have added a related reference for readers, which shows the light-induced dimension changes of the CDW phase in 1T-TiSe₂.

(4) Reviewer's comments:

In the paragraph before the section "Ultrafast dynamics of CDW orders", there is a sentence "The electron relaxation time extracted from $\Delta R/R$ spectra diverges near 320 K because the CDW gap opens. This is significantly higher than the onset temperature (280 K) for collective modes along the a-axis, which indicates the electronic-driven 1D-CDW-chain phase transition in CuTe." Does the onset temperature (280 K) really means that the 320 K transition is purely electronic driven? As I mentioned in the previous comments (in the 4th graph of this review report), the onset temperature below which the folded modes can be observed are often lower than T_c in many studies, because near T_c the modes may be quickly damped or the oscillation strength becomes too weak.

Responses:

Thank the reviewer for reminding us of this point. We agree with the reviewer's comment that the onset temperature below which the folded modes can be observed are often lower than T_c . Therefore, the sentence of "... the CDW gap opens. This is significantly higher than the onset temperature (280 K) for collective modes along the a -axis, which indicates the electronic-driven 1D-CDW-chain phase transition in CuTe." has been revised to "... the CDW gap opens, which is significantly higher than the onset temperature (280 K) for collective modes along the a -axis [5-8].". (Line 6, the 2nd paragraph on page 6 and line 7, the 1st paragraph on page 7 of this revised manuscript)

(5) Reviewer's comments:

In Fig. S2, the ~2.3 THz mode is present in the Raman spectra above 220 K, even though the peaks are lower and broader than at low temperatures. This is against Fig. 3a where this mode is absent above 220 K in the plotted Raman data. The authors should explain such contradiction.

Responses:

In order to clarify this point, the Raman spectra from 180 K to 280 K (marked by a dashed-red rectangle in Fig. R5a) have been enlarged to show the detailed feature of peak evolution in Fig. R5b. As increasing temperatures from 180 K, the 2.25-THz peak can be described well by a single Gaussian function (green-shaded area in Fig. R5b). A black arrow marks the peak positions of the 2.25-THz peak and a single Gaussian function. However, the 2.25-THz peak can no longer be described by a single Gaussian function when the temperature is above 220 K, caused by the splitting of the 2.25-THz peak and marked by two arrows. This indicates new modes to develop above 220 K rather than only a single mode of 2.25 THz. Therefore, the 2.25-THz peak can be only well-identified below 220 K in Raman spectra, consistent with the pump-probe results in Fig. 3a of this revised manuscript.

The above discussion has been added to the Supplementary Information (section S3).

Fig. R5. a Raman spectra along the a -axis ($E//a$ -axis) of a CuTe single crystal at various temperatures. PP: The excitation and the detection polarizations were set along the a -axis of CuTe crystals. **b** Enlargement of the Raman spectra marked by a dashed-red rectangle in **a**. The arrows indicate the positions of peaks, which are consistent with the peak positions of the Gaussian function (the green-shaded area) below 220 K. (*Fig. S4 in Supplementary Information has been updated by this figure*)

In summary, we have addressed and implemented all points raised by the reviewer. The new manuscript version has been improved significantly and has reached the level of thoroughness and quality suitable for publication in Nature Communications.

References

- [1] Cheng, Y. et al. Light-induced dimension crossover dictated by excitonic correlations. *Nat. Commun.* **13**, 963 (2022)
- [2] Luo, C. W. et al. Dichotomy of photoinduced quasiparticle on CuO₂ planes of YBa₂Cu₃O_{7- δ} directly revealed by femtosecond polarization spectroscopy. *J. Appl. Phys.* **102**, 033909 (2007).
- [3] Luo, C. W. et al. Spatial dichotomy of quasiparticle dynamics in underdoped thin-film YBa₂Cu₃O_{7- δ} superconductors. *Phys. Rev. B* **74**, 184525 (2006).
- [4] Luo, C. W. et al. Spatial symmetry of superconducting gap in YBa₂Cu₃O_{7- δ} obtained from femtosecond spectroscopy. *Phys. Rev. B* **68**, 220508R (2003).
- [5] Demsar, J., K. Biljaković & Mihailovic, D. Single particle and collective excitations in the one-dimensional charge density wave solid K_{0.3}MoO₃ probed in real time by femtosecond spectroscopy. *Phys. Rev. Lett.* **83**, 800 (1999).
- [6] Yue, L. et al. Highly anisotropic transient optical response of charge density wave order in ZrTe₃. *Phys. Rev. B* **107**, 165115 (2023).
- [7] Lin, T. et al. Optical spectroscopy and ultrafast pump-probe study on Bi₂Rh₃Se₂: Evidence for charge density wave order formation. *Phys. Rev. B* **101**, 205112 (2020).
- [8] Schmitt, F. et al. Transient electronic structure and melting of a charge density wave in TbTe₃. *Science* **321**, 1649-1652 (2008).

Responses to Reviewer #3:

We appreciate the Reviewer's time and suggestions in the comment letter, which did help us to polish this research work. The detailed responses to the Reviewer's concern are provided as follows.

(1) Reviewer's comments:

The findings are plausible and deserve to be published. Unfortunately, the physical explanation of the driving mechanisms between the different CDW regimes remains rather obscure. In the supporting information, the authors show that two of the CDW instabilities are visible as dips or even soft modes in the phonon dispersion of the primitive (high-symmetry) unit cell of CuTe. Using this dispersion as starting point, the authors should explain why at different temperatures the different soft modes become dominant. Without a clear and understandable explanation of the driving mechanism, the paper does not meet the high criteria of importance for a general audience that one expects from nature communications.

Responses:

The authors are grateful for the suggestion of the reviewer to allow for a chance to comment on the possible thermal effects on the stability of two soft modes. It is well known that fully finite-temperature first-principles calculations, especially for the CDW modulations, including large-scale supercells, are extremely computational demanding, which is out of the scope of the present work. However, it is still possible to explore qualitatively thermal effects by performing calculations in different smearing widths with respect to various artificial electronic temperatures and corresponding lattice constants. As shown in Fig. R1a, it is clear that the soft mode at q_1 (0.4, 0, 0) corresponding to the CDW_{1D} phase is more sensitive to increasing smearing width (artificial temperature) and could be stabilized while the mode at q_2 (0.4, 0, 0.5) responsible for the CDW_{3D} phase is still softening. Also, the calculated phonon spectra of the non-CDW phase of experimental lattice constants at different temperatures, as illustrated in Fig. R1b, show the similar thermal behaviors of the two modes at q_1 and q_2 , respectively. Furthermore, the unfolded phonon spectrum of the CDW_{1D} state in Fig. R1c supports that such $5 \times 1 \times 1$ supercell modulation could only stabilize the soft mode at q_1 rather

than the one at q_2 . Indeed, as discussed in section S6 of Supplementary Information (Fig. S9), only CDW_{3D} modulation derived from the eigenvector of the imaginary frequency at q_2 of the non-CDW phase could stabilize all the phonon instabilities.

Experimentally, the higher frequency 2.26-THz mode that exhibits the same softening behavior as the 1.64-THz is eliminated at temperatures above 220 K in Fig. 3a (this revised manuscript). First-principles calculations show that the 2.26-THz mode is associated with a new A_g mode (see Fig. 3e in this revised manuscript), for which the atomic configuration differs from that of the 1.64-THz mode. There is a lateral shift between two neighboring layers, as shown by the thick arrow in Fig. 3e (this revised manuscript). This indicates an additional interaction between two adjacent layers along the c -axis to form a 3D CDW (CDW_{3D}) at temperatures below 220 K with the atomic configuration shown in Fig. 3e (this revised manuscript). The dimension transition from CDW_{1D} to CDW_{3D} is also accompanied by a reduction in the c -axis shrinking rate as temperatures decrease, as shown in Fig. 3b (solid triangles in this revised manuscript). Namely, the lattice along the c -axis becomes more stable and rigid (lower shrinking rate) as the interval between two neighboring layers (CDW planes) decreases. Sufficient interlayer interactions between two neighboring CDW planes establish a more stable CDW_{3D} phase at temperatures below 220 K. Recently, Kuo *et al.* [1] reported that the temperature-dependent specific heat C_p for CuTe is fitted well by the lattice specific heat at temperatures below ~ 220 K, so the CDW phase at low temperatures is mainly determined by the lattice (or atomic structure). Therefore, the CuTe first suffers dimensional fluctuations in the intermediate temperature region of 220-250 K, in which the peak width of the 1.64-THz mode fluctuates significantly, as denoted by the gray area in Fig. 3a (this revised manuscript). CuTe then forms a stable 3D CDW at temperatures below 220 K.

The above discussion has been added to the 2nd paragraph (line 6-10) on page 11 and the 1st paragraph (line 4) on page 12 of this revised manuscript & the Supplementary Information (section S6).

Fig. R1. The DFPT calculated phonon spectra at room-temperature lattice constants and different smearing widths **a** and lattice coordinates at different temperatures **b** of the non-CDW phase of CuTe. **c** The DFPT calculated phonon spectra of CuTe in the normal phase (blue curves) and CDW_{1D} (red curves). The corresponding unfold weight is shown in grey scale. Phonon soft modes with prominent imaginary frequencies are indicated as q_1 (0.4, 0.0, 0.0) and q_2 (0.4, 0.0, 0.5), respectively. (*This figure has been added to section S6 (Fig. S11) of Supplementary Information.*)

(2) Reviewer's comments:

The last phrase of the abstract ("This study shows that the hidden CDW phases with various dimensions and their transition mechanisms are critical for CDW materials.") does not make sense. What does it mean that the hidden CDW phases are critical for CDW materials? Furthermore, there are materials with just one CDW phase where there are no "hidden" CDW phases.

Responses:

Thank the reviewer for reminding us. In the previous manuscript, the "hidden CDW phases" means the 1D-CDW, 2D-CDW, 3D-CDW phases, which was not disclosed before this study. In order to avoid confusion for readers, the sentence "This study shows that the hidden CDW phases with various dimensions and their transition mechanisms are critical for CDW materials." has been revised to "This study shows that the CDW phases with various dimensions and their transition mechanisms are crucial for CDW materials." (*The last 2 lines in Abstract of this revised manuscript*)

In summary, we have addressed and implemented all points raised by the reviewer. The new manuscript version has been improved significantly and has reached the level of thoroughness and quality suitable for publication in Nature Communications.

Reference

- [1] Kuo, C. N., Huang, R. Y., Kuo, Y. K. & Lue, C. S. Transport and thermal behavior of the charge density wave phase transition in CuTe. *Phys. Rev. B* **102**, 155137 (2020).

List of changes in the revised manuscript

Here, let us briefly summarize the major points and changes:

(1) The co-authors' affiliations have been revised as follows.

Author list on original submission: Nguyen Nhat Quyen¹, Wen-Yen Tzeng², Chin-En Hsu², I-An Lin³, ..., Yu-Ming Chang^{5,8}, Hung-Chung Hsueh³, Chin Shan Lue⁶, Chih-Wei Luo^{1,4,9,10,11}

Author list on this revised manuscript: Nguyen Nhat Quyen¹, Wen-Yen Tzeng², **Chin-En Hsu³**, I-An Lin³, ..., Yu-Ming Chang^{5,8}, **Chin Shan Lue^{*6}**, **Hung-Chung Hsueh^{*3}**, **Chih-Wei Luo^{*1,4,9,10,11}**

The description of author to whom correspondence should be addressed (*line 4 on page 2*) **was added** as following,

*** cwluoep@nycu.edu.tw; hchsueh@gms.tku.edu.tw; cslue@ncku.edu.tw**

(2) **lines 4-14 (page 3, Abstract of the previous manuscript)** “However, there is no complete CDW phase diagram in various dimensions and ..., variously dimensional CDW phases ... (1D) CDW chain phase at T_c of 335 K. At $T=280$ K, electron-phonon coupling creates collective modes along the a -axis, which synchronize via an interchain interaction to establish a two-dimensional (2D) CDW phase on the ab -plane while $T<250$ K. The 2D CDW phase planes are finally locked This study shows that the hidden CDW phases with various dimensions and their transition mechanisms are critical for CDW materials.” **was revised to** “However, there is **yet to be a** complete CDW phase diagram in various dimensions, and ..., **different** dimensional CDW phases ... (1D) **CDW chain** at T_c of 335 K. **When temperature decreases, the shrinking of c -axis length accompanied with the appearance of interchain and interlayer interactions causes the quantum fluctuations (QF) of the CDW phase in dimension until 220 K. At $T<220$ K, the CDWs on the different ab -planes** are finally locked with each other in anti-phase to form a three-dimensional (3D)

CDW phase. This study shows that **the CDW phases** with various dimensions and their transition mechanisms are **crucial** for CDW materials.”

(3) *line 1 (page 4 of the previous manuscript)* “Low-dimensional materials have been subject of much study because they feature...” *was revised to* “Low-dimensional materials have been **the** subject of **many studies** because they feature ...”

(4) *lines 6-9 (page 4 of the previous manuscript)* “Recently, the CDW transition temperature is significantly ... CDW phase transition is sensitive to the dimensionality. For an ideal 1D system, Peierls model ... band gap opening at zone boundary...” *was revised to* “Recently, the CDW transition temperature **has** significantly ... CDW phase transition is sensitive to **dimensionality**. For an ideal 1D system, **the** Peierls model ... band gap opening at **the** zone boundary...”

(5) *lines 14-15 (page 4 of the previous manuscript)* “...1D-CDW scenario and its nature is still moot” *was revised to* “...1D-CDW scenario, and its nature **still needs to be clarified**”

(6) More detailed description of the collective mode of phonons (amplitude, phason) were mentioned in the main text (*lines 16-19 on page 4 & lines 1-9 on page 5 of this revised manuscript*) as following,

“**In correlated electron systems, the order parameter instability with the spontaneously broken symmetry (SBS) has gained attention recently. Owing to the interaction among the charge, spin, orbital, and lattice degrees of freedom, the symmetry-breaking ground state, e.g., charge/spin order and superconductivity, is caused. Usually, the fluctuations in the SBS state exhibit a collective oscillation with the modes of phase (so-called Nambu-Goldstone mode or phasor)¹⁵ or amplitude (so-called Higgs amplitude mode or amplitudon)¹⁶. As shown in Fig. S1 (in Supplementary Information), the phasor is associated with the in-phase motion of ions, which causes the phase (ϕ) modulation of charge density (ρ) wave, and the phase (ϕ) changes with the same V in the minimum of Mexican hat-like (double-well) potential¹⁷. The amplitudon is associated with the out-of-phase motion of ions, which causes the amplitude (Δ) modulation of charge density (ρ) wave and the changes of V with the same ϕ in the Mexican hat-like (double-well) potential.**

As a result, the phase and amplitude modes are IR-active and Raman-active modes, respectively¹⁸.”

- (7) *lines 16-17 (page 4 of the previous manuscript)* “In terms of CDW materials, the vulcanite-phase ... and consists of...” *was revised to* “Regarding CDW materials, the vulcanite-phase ... **It** consists of ...”
- (8) *lines 18-19 (page 4 of the previous manuscript)* “This so-called ”11” system is so simple that it could be the key structure to disclose the CDW phases with various dimensions simultaneously in one system and the origin of CDW phase transitions.” *was revised to* “**which could be used to reveal** the CDW phases with various dimensions simultaneously in one system²¹ and the origin of CDW phase transitions.”
- (9) *lines 2-6 (page 5 of the previous manuscript)* “...measurements - specific heat, thermal conductivity ... T_c of 335 K,²¹ which show that there is electron-phonon coupling in CuTe²². Using density functional theory calculations,²³ Kim et al. proposed that the Coulomb correlation of Cu-3d electrons is also essential for activating the CDW phase transition. Besides, the electronic anisotropy in CuTe is revealed by infrared reflectivity measurements²⁴.” *was revised to* “...measurements - specific heat, thermal conductivity, and the Seebeck coefficient - demonstrate remarkable features near T_c of 335 K,²² which **show electron-phonon coupling** in CuTe²³ (see Supplementary section S6 for more **discussion**). Using density functional theory calculations, Kim et al.²⁴ proposed that the Coulomb correlation of Cu-3d electrons **is essential** for activating the CDW phase transition. **However, this scenario has been revisited and challenged recently by more careful calculations, including quantum anharmonicity.**²⁵ Besides, the electronic anisotropy in CuTe is revealed by infrared reflectivity measurements²⁶”
- (10) *lines 8-9 (page 5 of the previous manuscript)* “...discovered in a pressurized CuTe²⁵. There is debate over the interplay between electronic correlation and electron-phonon interaction and a possible CDW phase diagram in 11-type...” *was revised to* “...discovered in a pressurized CuTe²⁷. There is debate over the interplay between electronic correlation and electron-phonon interaction, and a possible CDW phase diagram **in CuTe.**”
- (11) *lines 13-17 (page 5 of the previous manuscript)* “...to determine ... degrees of freedom²⁶⁻²⁹. There is a unique characterization ability in the time domain ... disentangle

different spectrally overlapping components ... in CDW materials^{4,30-39}.” *was revised to* “...to determine ... degrees of freedom^{28, 29}. There is a unique characterization ability in the time domain, ... disentangle **various** spectrally overlapping ... in CDW materials^{4, 30-39} **(for details, see Supplementary section S1)**.”

- (12) *line 19 (page 5 of the previous manuscript) & lines 1-3 (page 6 of the previous manuscript)* “...crystals (see Fig. 1) to elucidate electron/collective mode dynamics simultaneously in three dimensions and ... ($\Delta R/R$) spectra are clearly observed on *ab*-plane.” *was revised to* “...crystals (see Fig. 1) to **simultaneously elucidate electron/collective mode dynamics in three dimensions** and ... ($\Delta R/R$) spectra **are observed on the *ab*-plane.**”
- (13) *lines 4-6 (page 6 of the previous manuscript)* “...time extracted from ... CDW gap opens. This is significantly higher than the onset temperature (280 K) for collective modes along the *a*-axis, which indicates the electronic-driven 1D-CDW-chain phase transition in CuTe.” *was revised to* “...time extracted from ... CDW gap opens, **significantly higher than the onset temperature (280 K) for collective modes along the *a*-axis^{28, 33-35}.**”
- (14) More detailed description of the wave vectors of 2D- and 3D-CDW phases were mentioned in the main text (*lines 6-11 on page 6 of the previous manuscript*) as following, “STM mapping shows the spatial modulation of CDW with $q_{\text{CDW}}=0.4a^*$ and further reveals a **2D CDW** on the *ab*-plane due to an interchain interaction between 1D-CDW chains. Moreover, a **3D CDW with $q_{\text{CDW}} = 0.4a^* + 0.5c^*$** was first observed at low temperatures using orientation-/time-resolved ultrafast spectroscopy and has been verified by first-principles calculations for a lateral shift between two neighboring **CDW planes.**”
- (15) *lines 13-14 (page 6 of the previous manuscript)* “...the *ab*-plane of CuTe, which is a composite of the damped oscillations and the exponential decay background. In order to figure out ...” *was revised to* “...the **CuTe *ab*-plane, a composite** of the damped oscillations and the exponential decay background. **To** figure out ...”
- (16) *lines 17-18 (page 6 of the previous manuscript)* “After pumping by 3.1-eV photons, there is bleaching around the Fermi surface (E_F), which causes a positive change in $\Delta R/R$ for 1.55-eV probing photons at zero delay time...” *was revised to* “ After pumping by 3.1-eV photons, there is bleaching **(reduction of the electronic population)** around the Fermi surface (E_F), which causes a positive change in $\Delta R/R$ for 1.55-eV probing photons

at zero delay time to reveal the variations of physical characteristics associated with the changes of Fermi surface.”

- (17) *lines 2-3 (page 7 of the previous manuscript)* “which is much higher than the initial lattice (sample) temperature. According to three-temperature...” *was revised to* “much higher than the initial lattice (sample) temperature. According to the three-temperature...”
- (18) *line 12 (page 7 of the previous manuscript)* “The second term describes the dynamics of phonons that have an initial number A_{ph} ...” *was revised to* “The second term describes the dynamics of phonons with an initial number A_{ph} ...”
- (19) *lines 13-14 (page 7 of the previous manuscript)* “Finally, hot electrons and hot phonons, that reach thermal equilibrium at $T_e=T_{ph}$, simultaneously transfer their energy to the lattice...” *was revised to* “Finally, hot electrons and phonons, that reach thermal equilibrium at $T_e=T_{ph}$, simultaneously transfer their energy to the lattice...”
- (20) *lines 1-4 (page 8 of the previous manuscript)* “Figure 2b shows a spectrogram ... Fig. 2a, which is obtained using a short-time Fourier transformation... at 37 K but only one collective mode with much broad width and short lifetime is observed at 280 K” *was revised to* “Figure 2b shows a spectrogram ... Fig. 2a, obtained using a short-time Fourier transformation... at 37 K, but only one collective mode with a much broader width and short lifetime is observed at 280 K.”
- (21) *lines 6-7 (page 8 of the previous manuscript)* “...analyze the experimental data in Fig. 2cd (for details, see section S4 in Supplementary Information).” *was revised to* “...analyze the experimental data in Fig. 2cd (for details, see Supplementary section S5).”
- (22) *line 13 (page 8 of the previous manuscript)* “well of a DWP $V(T_{eL1}, t_1)$ and gradually cool accompanied with the evolution of DWP” *was revised to* “well of a DWP $V(T_{eL1}, t_1)$ and gradually cool, accompanied by the evolution of DWP”
- (23) *line 6 (page 9 of the previous manuscript)* “...the max- κ_L temperature in the high temperature region)²¹. This implies that there is coupling” *was revised to* “...the max- κ_L temperature in the high-temperature region)²². This implies that there is a coupling”
- (24) *line 9 (page 9 of the previous manuscript)* “...the quasi-1D $K_{0.3}MoO_3$ ^{43,44} and rare-earth tritellurides RTe_3 ” *was revised to* “...the quasi-1D $K_{0.3}MoO_3$ ^{43,44} and rare-earth tri-tellurides RTe_3 ”
- (25) *line 12 (page 9 of the previous manuscript)* “Raman spectra in Fig. 3a (for details, see

section S2 in Supplementary Information).” *was revised to* “Raman spectra in Fig. 3a (see Supplementary section S3).”

- (26) *lines 17-18 (page 9 of the previous manuscript)* “... unit cell of 0.31 nm×0.40 nm, which is enclosed by the yellow rectangle in Fig. 3f¹⁹. The line-profile along the red-dashed arrow...” *was revised to* “... unit cell of 0.31 nm×0.40 nm, enclosed by the yellow rectangle in Fig. 3f¹⁹. The line profile along the red-solid arrow...”
- (27) *lines 2-3 (page 10 of the previous manuscript)* “...contains a basis of two atomic rows, followed by a basis of three atomic rows. The displacement of each Te atom is indicated by the blue arrows.” *was revised to* “...contains a basis of two atomic rows (i.e., the number “2” in Fig. 3f), followed by a base of three atomic rows (i.e., the number “3” in Fig. 3f). The yellow and blue arrows indicate the displacement of each basis of atomic rows, respectively.”
- (28) *line 5 (page 10 of the previous manuscript)* “... the reciprocal lattice of the 1×1 unit cell and the red circles indicate...” *was revised to* “... the reciprocal lattice of the 1×1 unit cell, and the red circles indicate...”
- (29) *lines 7-9 (page 10 of the previous manuscript)* “...which is in agreement with the charge density ... (for details, see section S5 in Supplementary Information) and several previous predictions that use other methods^{19,20,23}.” *was revised to* “...which agrees with the charge density ... (for details, see Supplementary section S6) and several previous predictions that use other methods^{19,20,24}.”
- (30) *lines 10-13 (page 10 of the previous manuscript)* “... the 5-Te-atoms modulation chain along *a*-axis is repeated well along the *b*-axis and demonstrates the interchain coupling, as verified by the pump-probe spectroscopy in Fig. 3a. Consequently, the CDW phase at temperatures below 250 K is a 2D-CDW phase (CDW_{2D}) with an atomic configuration that is shown in Fig. 3d.” *was revised to* “... the 5-Te-atoms modulation chain along the *a*-axis is repeated well along the *b*-axis. It demonstrates a 2D CDW formed via the interchain coupling, as verified by the pump-probe spectroscopy along the *b*-axis in Fig. 3a.”
- (31) *line 17 (page 10 of the previous manuscript)* “... that of the 1.64-THz mode and there is a lateral shift ...” *was revised to* “... that of the 1.64-THz mode. There is a lateral shift...”
- (32) *line 19 (page 10 of the previous manuscript)* “...additional interaction between two

neighboring layers along the c -axis ...” *was revised to* “...additional interaction between two adjacent layers along the c -axis ...”

- (33) *lines 2-5 (page 11 of the previous manuscript)* “The phase transition from CDW_{2D} to CDW_{3D} ... as temperatures decreases, as shown in Fig. 3b (solid triangles). That is the lattice along c -axis becomes more stable and rigid (smaller shrinking rate) as the interval between two neighboring layers (CDW_{2D} planes) decreases.” *was revised to* “The dimension transition from CDW_{1D} to CDW_{3D} ... as temperature decreases, as shown in Fig. 3b (solid triangles). Namely, the lattice along the c -axis becomes more stable and rigid (lower shrinking rate) as the interval between two neighboring layers (CDW planes) decreases.”
- (34) *lines 5-7 (page 11 of the previous manuscript)* “Sufficient interlayer interactions between two neighboring CDW_{2D} planes establish a more stable CDW_{3D} phase at temperatures below 220 K. Recently, Kuo *et al.*²¹ reported that...” *was revised to* “Sufficient interlayer interactions between two neighboring CDW planes establish a more stable 3D CDW at temperatures below 220 K. Recently, Kuo *et al.*²² reported that...”
- (35) *lines 10-12 (page 11 of the previous manuscript)* “the CuTe first forms a metastable CDW_{2D} phase in the intermediate temperature region of 220-250 K, ... as denoted by the gray area in Fig.3a. CuTe then forms a stable CDW_{3D} phase at temperatures below 220 K.” *was revised to* “the CuTe first suffers dimensional fluctuations in the intermediate temperature region of 220-280 K, ... as denoted by the gray area & QF in Fig. 3a. CuTe then forms a stable 3D CDW at temperatures below 220 K.”
- (36) *lines 14-15 (page 11 of the previous manuscript)* “...(for details, see section S4 in Supplementary Information).” *was revised to* “...(for details, see Supplementary section S5).”
- (37) *line 19 (page 11 of the previous manuscript)* “...that are obtained from the Raman spectra²².” *was revised to* “that are obtained from the Raman spectra²³.”
- (38) *lines 3-5 (page 12 of the previous manuscript)* “This electron-phonon constant...around 290 K, which is the temperature at which the resistivity is greatest and significantly less than T_c .” *was revised to* “This electron-phonon constant...around 290 K (i.e., the temperature with the greatest resistivity and significantly less than T_c).”
- (39) *line 6 (page 12 of the previous manuscript)* “... the formation of CDW phase just at T_c .” *was revised to* “... the formation of the CDW phase just at T_c .”

- (40) *lines 12-13 (page 12 of the previous manuscript)* “According to the theoretical band structure calculations, the formation of CDW_{1D/2D} (or CDW_{3D}) phase only partially open gaps but not full Brillouin Zone...” *was revised to* “According to the theoretical band structure calculations, the formation of CDW_{1D} (or CDW_{3D}) only partially opens gaps but not the full Brillouin Zone...”
- (41) *line 2 (page 13 of the previous manuscript)* “...photoemission spectroscopy (ARPES) (see section S3 in Supplementary Information).” *was revised to* “...photoemission spectroscopy (ARPES) (see Supplementary section S4).”
- (42) *lines 4-5 (page 13 of the previous manuscript)* “...the electronic phase transition temperature indicated by τ_e is much close to $T_c=335$ K, which is significantly higher than the onset temperature...” *was revised to* “...the electronic phase transition temperature indicated by τ_e is much closer to $T_c=335$ K, significantly higher than the onset temperature...”
- (43) *line 8 (page 13 of the previous manuscript)* “...with a strong Coulomb correlation of Cu-3d electrons²³.” *was revised to* “...with a strong Coulomb correlation of Cu-3d electrons²⁴.”
- (44) *line 11 (page 13 of the previous manuscript)* “...CDW_{1D} phase. This causes metallic behavior...” *was revised to* “... CDW_{1D}. This causes metallic behavior...”
- (45) *line 14 (page 13 of the previous manuscript)* “...indicates that the CDW_{3D} phase is stabilized due to...” *was revised to* “... indicates that the CDW_{3D} is stabilized due to...”
- (46) *lines 16-18 (page 13 of the previous manuscript)* “To determine ... associated with the 3D-CDW phase, pump-probe measurements ..., as shown in Fig. 1.” *was revised to* “To determine ... associated with the CDW_{3D}, pump-probe measurements ..., as shown in Fig. 1 (also Supplementary Fig. S12).”
- (47) *line 19 (page 13 of the previous manuscript) & lines 1-3 (page 14 of the previous manuscript)* “... $E_{//}$ is parallel to the a -axis and E_{\perp} is parallel ... for $\theta=8^\circ$ and 45° . After FFT, two amplitudons are clearly observed for $\theta=8^\circ$ and 45° . Peak fitting reveals that there are two amplitudons for $\theta=8^\circ$ with...” *was revised to* “... $E_{//}$ is parallel to the a -axis, and E_{\perp} is parallel ... for θ from 8° to 45° . After FFT, two amplitudons are observed for $\theta=8^\circ$, 22° , 30° , and 45° . Peak fitting reveals two amplitudons for $\theta=8^\circ$ with...”
- (48) *lines 7-10 (page 14 of the previous manuscript)* “...to 1.97 THz but the low-

frequency ... of A_{HF}/A_{LF} increases to 0.56, which is 18.7 times larger than the value for $\theta=8^\circ$. These results clearly show that the high-frequency amplitudon at around 2 THz can be used as an indicator for the interaction between two neighboring CDW_{2D} layers along the c -axis.” *was revised to* “...to 1.92 THz, which can be well described by a cos function in Supplementary Fig. S12c, but the low-frequency ... of A_{HF}/A_{LF} increases to 0.52, which is 17.3 times larger than the value for $\theta=8^\circ$. These results show that the high-frequency amplitudon at around 2 THz can indicate the interaction between two neighboring CDW layers along the c -axis.”

(49) *line 11 (page 14 of the previous manuscript)* “In summary, the structure of CDW phases of CuTe in real space ...” *was revised to* “The CDW_{3D} structure of CuTe in real ...”

(50) *line 13-19 (page 14 of the previous manuscript) & lines 1-5 (page 15 of the previous manuscript)* “At temperatures below T_c , the CDW_{1D}-chain phase begins to form along the a -axis (the solid-wave lines in Fig. 4c), but there is no interchain interaction. There is then coupling with the phonon subsystem by electron-phonon coupling λ_a to form the amplitudon along the a -axis if $T < 280$ K. At temperatures below 250 K, the CDW_{1D} chains synchronize along the b -axis via an interchain interaction to form a CDW_{2D} phase on the ab -plane. This is shown as in-phase solid-wave lines in Fig. 4c. If the temperature $T < 220$ K, interlayer coupling locks two neighboring CDW_{2D}-phase planes in anti-phase to establish a CDW_{3D} phase, as shown by the dashed squares in Fig. 4c. ... (for details, see section S5 in Supplementary Information). Fig. 4d represents the modulated structure in the CDW_{3D} phase with charge distribution ... dominate the charge distribution difference with out-of-phase feature and have a key role in the interlayer coupling between two neighboring CDW_{2D}-phase planes.” *was revised to* “At temperature below 220 K, the interlayer coupling locks two neighboring CDW planes in anti-phase to establish a 3D CDW, as shown by the dashed squares in Fig. 4c. ... (for details, see Supplementary section S6). Fig. 4d represents the modulated structure in the 3D CDW with charge distribution ... dominate the charge distribution difference with out-of-phase features and have a key role in the interlayer coupling between two neighboring CDW planes.”

(51) The summary for the dimension evolution of the CDW phase from high to low temperatures in CuTe were mentioned in the main text (*lines 13-19 on page 15 of this revised manuscript*) as following,

“The dimension evolution of the CDW phase from high to low temperatures in CuTe can

be concluded in Fig. 4e. At 335 K, a 1D CDW along the a -axis forms with $1c$ periodicity and in-phase along the c -axis. When the temperature decreases, the shrinking of the c -axis length (see Fig. 3b) causes the quantum fluctuations of the CDW phase in dimension until 220 K²¹. The CDW will be further stabilized with $2c$ periodicity and out-of-phase along the c -axis below 220 K to form a 3D CDW due to locking the in-plane CDW phase among layers in the out-of-plane direction.”

- (52) *line 3 (page 16 of the previous manuscript)* “...Te ingots (99.9999%) were mixed with a molar ratio of 35:65 and...” *was revised to* “...Te ingots (99.9999%) were mixed with a molar **ratio 35:65** and...”
- (53) *lines 5-6 (page 16 of the previous manuscript)* “...cooled to 410 °C for 2 h and slowly ... by centrifugation at 345 °C and several plate...” *was revised to* “...cooled to 410 °C for 2 h, and slowly ... by centrifugation at 345 °C, and several plate...”
- (54) *lines 1-4 (page 17 of the previous manuscript)* “A delay stage was used to adjust the optical path length of the pump pulses and to control the time delay ... the fast-axis angles of the half wave plates in front of the samples” *was revised to* “A delay stage was used to adjust **the pump pulses' optical path length** and control the time delay ...the fast-axis angles of the **half-wave** plates in front of the samples”
- (55) *lines 11-12 (page 17 of the previous manuscript)* “...temperature-dependent measurements and were cleaved using a scotch tape prior to the measurements to ensure that the surface was clean and bright.” *was revised to* “...temperature-dependent measurements and were cleaved **using scotch tape before the measurements to ensure the surface was clean and bright.**”
- (56) *lines 14-15 (page 17 of the previous manuscript)* “**Temperature dependent polarized Raman spectroscopy.** The polarized Raman spectroscopy were performed by varying ...” *was revised to* “**Temperature-dependent** polarized Raman spectroscopy. The polarized Raman spectroscopy **was** performed by varying ...”
- (57) *lines 7-10 (page 18 of the previous manuscript)* “...(for details, see section S5 in Supplementary Information) with ultrasoft pseudopotentials... (PBE) functional (for details, see section S5 in Supplementary Information) was employed to approximate...” *was revised to* “...(for details, see **Supplementary section S6**) with ultrasoft pseudopotentials... (PBE) functional (for **more information**, see **Supplementary section S6**) was employed to approximate...”

- (58) *line 14 (page 18 of the previous manuscript)* “...CDW_{1D/2D} and CDW_{3D}, guided by the eigenvectors the soft modes with calculated...” *was revised to* “...CDW_{1D} and CDW_{3D}, guided by the eigenvectors of the soft modes with calculated...”
- (59) *line 18 (page 18 of the previous manuscript)* “...unit cell (20×16×8), CDW_{1D/2D} (4×16×4) and CDW_{3D}...” *was revised to* “...unit cell (20×16×8), CDW_{1D} (4×16×4), and CDW_{3D} ...”
- (60) *line 1 (page 19 of the previous manuscript)* “..., CDW_{1D/2D} and CDW_{3D} phases ...” *was revised to* “..., CDW_{1D}, and CDW_{3D} phases ...”
- (61) *line 3 (page 19 of the previous manuscript)* “...perturbation theory (DFPT) (for details, see section S5 in Supplementary Information).” *was revised to* “...perturbation theory (DFPT) (for details, see Supplementary section S6).”
- (62) *line 6 (page 19 of the previous manuscript)* “...code (for details, see section S5 in Supplementary Information).” *was revised to* “...code (for details, see Supplementary section S6).”
- (63) The description on **Acknowledgments** (*lines 11-17 on page 19 of this revised manuscript*) *was revised* as following, “This work was supported ..., Taiwan (Grant: 109-2112-M-009-020-MY3,...the research support from NSTC, Taiwan, under Grant”
- (64) The description on **Author contributions** (*lines 3-10 on page 20 of this revised manuscript*) *was revised* as following,
“C. W.L. proposed the project. N.N.Q., W.Y.T., C.M.T., and C.W.L. performed the pump-probe experimental measurements and analyzed the data. C.N.K. and C.S.L. prepared the samples...the analysis of TDGL theory. P.H.L., H.H.J., and S.C.W. performed the measurements of ARPES. M.W.C. performed the structure characterization. C.E.H., I.A.L., and H.C.H. performed the first-principles calculations. N.N.Q. and C.W.L. wrote the manuscript. All authors edited and approved the final manuscript.”
- (65) The **Data availability** (*lines 14-16 on page 20 of this revised manuscript*) *was added* as following,
All the data that support the findings of this study are available from the corresponding authors upon reasonable request.
- (66) The **Code availability** (*lines 18-20 on page 20 of this revised manuscript*) *was added* as following,
All the code that support the plots of this study are available from the corresponding authors upon reasonable request.

List of changes references in the revised manuscript

15. Nambu, Y. Energy gap, mass gap, and spontaneous symmetry breaking. *Int. J. Mod. Phys. A* **25**, 4141 (2010).
16. Higgs, P. W. Broken symmetries, massless particles and gauge fields. *Phys. Lett.* **12**, 132-133 (1964).
17. Thomson, M. D. et al. Phase-channel dynamics reveal the role of impurities and screening in a quasi-one-dimensional charge-density wave system. *Sci. Rep.* **7**, 2039 (2017).
18. Grüner, G. Charge Density Wave in Solids, Frontiers in physics Ed D. Pines (Addison-Wesley) (1994).
21. Cheng, Y. et al. Light-induced dimension crossover dictated by excitonic correlations. *Nat. Commun.* **13**, 963 (2022).
22. Kuo, C. N., Huang, R. Y., Kuo, Y. K. & Lue, C. S. Transport and thermal behavior of the charge density wave phase transition in CuTe. *Phys. Rev. B* **102**, 155137 (2020).
23. Wang, S. et al. Observation of room-temperature amplitude mode in quasi-one-dimensional charge-density-wave material CuTe. *Appl. Phys. Lett.* **120**, 151902 (2022).
24. Kim, S., Kim, B. & Kim, K. Role of Coulomb correlations in the charge density wave of CuTe. *Phys. Rev. B* **100**, 054112 (2019).
25. Campetella, M., Marini, G., Zhou, S. J. & Calandra, M. Electron-phonon driven charge density wave in CuTe. *Phys. Rev. B* **108**, 024304 (2023).
26. Li, R. S. et al. Optical spectroscopy and ultrafast pump-probe study of a quasi-one-dimensional charge density wave in CuTe. *Phys. Rev. B* **105**, 115102 (2022).
27. Wang, S. et al. Two distinct charge density wave orders and emergent superconductivity in pressurized CuTe. Preprint at <https://arxiv.org/abs/2210.13896> (2022).

28. Demsar, J., K. Biljaković & Mihailovic, D. Single particle and collective excitations in the one-dimensional charge density wave solid $\text{K}_{0.3}\text{MoO}_3$ probed in real time by femtosecond spectroscopy. *Phys. Rev. Lett.* **83**, 800 (1999).
29. Luo, C. W., Wang, Y. T., Yabushita, A. & Kobayashi, T. Ultrabroadband time-resolved spectroscopy in novel types of condensed matter. *Optica* **3**, 82 (2016).
32. Hellmann, S. et al. Time-domain classification of charge-density-wave insulators. *Nat. Commun.* **3**, 1069 (2012).
33. Yue, L. et al. Highly anisotropic transient optical response of charge density wave order in ZrTe_3 . *Phys. Rev. B* **107**, 165115 (2023).
34. Lin, T. et al. Optical spectroscopy and ultrafast pump-probe study on $\text{Bi}_2\text{Rh}_3\text{Se}_2$: Evidence for charge density wave order formation. *Phys. Rev. B* **101**, 205112 (2020).
35. Schmitt, F. et al. Transient electronic structure and melting of a charge density wave in TbTe_3 . *Science* **321**, 1649-1652 (2008).

List of changes description on Figure in the revised manuscript

- (1) The caption of **Fig. 1** (*lines 9 on page 28 of the previous manuscript*) “...and the incident direction of probe beam.” *was revised to* “... the incident direction of **the** probe beam.”
- (2) The caption of **Fig. 3** *was revised* as following,

lines 5-7 (page 30 of the previous manuscript) “...(for $\mathbf{E} // b$ -axis from **Fig. S1**) and Raman modes (obtained from the Raman spectra in **Fig. S2**) as a function of temperature. Solid lines show the fitting results using TDGL equation, Eq. (S2).” *was revised to* “...(for $\mathbf{E} // b$ -axis from **Fig. S3**) and Raman modes (obtained from the Raman spectra in **Fig. S4**) as a function of temperature. Solid lines show the fitting results using **the** TDGL equation, Eq. (S2). **QF: quantum fluctuations.**”

lines 5-8 (page 31 of the previous manuscript) “...ARPES: obtained from the ARPES images in **Fig. S3. d, e** Modulated structure in the one-dimensional CDW ($\text{CDW}_{1\text{D}/2\text{D}}$) and three-dimensional CDW ($\text{CDW}_{3\text{D}}$) phases that is proposed by first-principles

simulations (for details, see section S5 in Supplementary Information). **f** High resolution STM image...” *was revised to* “...ARPES (k_x): obtained from the ARPES images in Fig. S5. ARPES (k_z): obtained from the ARPES images in Fig. S6. **d, e** Modulated structure in the one-dimensional CDW (CDW_{1D}) and three-dimensional CDW (CDW_{3D}) that is proposed by first-principles simulations (for details, see Supplementary section S6). **f** High-resolution STM image”

line 11 (page 31 of the previous manuscript) “... 1×1 unit cell and the red circles...” *was revised to* “... 1×1 unit cell, and the red circles...”

lines 14-15 (page 31 of the previous manuscript) “...(for details, see section S5 in Supplementary Information).” *was revised to* “...(for details, see Supplementary section S6).”

(3) The caption of **Fig. 4** *was revised* as following,

lines 10 (page 32 of the previous manuscript) “...value of 0.007 e/bohr^3 is adopted whereas the positive ... (for details, see section S5 in Supplementary Information” *was revised to* “...value of 0.007 e/bohr^3 is adopted, whereas the positive ... (for details, see Supplementary section S6). **e** The dimension evolution of CDW phase from high to low temperatures in CuTe.”

REVIEWER COMMENTS

Reviewer #1 (Remarks to the Author):

The authors have made necessary changes and clarified the doubts.

The only point missed out to clarify earlier are,

(i) the possibility of pure phase of CuTe. As CuTe/Cu₂Te has an interesting phase diagram with composition [ACS Applied Energy Materials 3, 3, 2175(2020)], thus a comparison and comment on obtaining the phase pure CuTe would be good.

(ii) Raman spectra in the SI is not elaborated for the peak identifications. CDW has several interesting modes [Physical Review Research, 02, 033118 (2020)] including amplitude and phase modes. Some discussion on the assignment of the modes in SI would make it better. Authors may like to consider it.

The paper can be considered for further proceedings.

Reviewer #2 (Remarks to the Author):

In the revised manuscripts, the authors have put great efforts to improve the context according to the reviewers' comments, which should be appreciated. The authors have properly addressed most of the comments I raised. However, there's some remaining concerns which I think of key importance.

1. My first concern is still about the "3D CDW" proposed by the paper. In this 1D system, the "3D" CDW means increased CDW dimensionality along the other two directions, namely the c and b directions, which I will elaborate separately about my concern as followings:

As for the c direction, the authors used ARPES data to show the CDW gap feature in this direction. Combined with the behavior of collective modes in the pump probe spectroscopy, I can agree that there may be increased CDW dimensionality along c axis. In previous publications, the qCDW was always reported to be $(0.4a^*, 0, 0.5c^*)$. However, the authors said there is a change of qCDW from $1c^*$ to $0.5c^*$ below 220 K, which I have not seen in the paper direct experimental evidence of how the different wavevectors are determined. The one evidence of the wavevector change is the

theoretical calculation, but it can only be viewed as a “theoretical prediction” without any experimental evidence.

As for the b direction. I go through the paper and find almost no proof of the CDW dimensionality along the b-direction. A weak proof is claimed on page 11 as “In the STM image, the 5-Te-atoms modulation chain along the a-axis is repeated well along the b-axis. It demonstrates a 2D CDW formed via the interchain coupling”. I think the repetition of chains in the b direction cannot prove the 2D CDWs, since in STM measurements on other 1D CDW materials the 1D CDW chains are also repeated. Besides, in the in-plane anisotropic optical conductivity measurements (ref 26), the CDW gap feature is only present in a direction and always absent in b direction, which is against the “2D” CDWs in this paper. Therefore, even if the claim of CDW along c direction can possibly be true, the proposed “3D” CDW is not reasonable considering lack of evidence of the b direction.

2. About the results shown in Fig. 4ab: The shift of the mode frequency with increasing incident angle looks quite novel and I have not seen such results in previous pump probe studies in CDW systems. For me this is one of the most interesting results of this work and deserves to be discussed more in the paper. But I am puzzled by the explained $\lambda\theta=1/f\theta$ scenario in Supplementary section S7, which looks quite unreasonable. According to the scenario, does it seem that such frequency shift could appear in any CDW systems? And, when rotating the E field direction in the ab plane (instead of changing incident angle), will the mode frequency also shift according to $\lambda\theta=1/f\theta$? Indeed, I have never seen such examples in CDW systems before. Moreover, what is the relation between the frequency shift and the CDW dimensionality? This is almost not discussed in the main text except for a concluding sentence “These results show that the high-frequency amplitudon at around 2 THz can indicate the interaction between two neighboring CDW layers along the c-axis.”

In all, since the authors do not convince me of the proposed “3D” CDW, which is the main idea of the paper, I think the paper is not suitable for publication in Nature Communications. However, since the work contains some meaningful results, it can still be suitable for publication in a more specialized journal.

Reviewer #3 (Remarks to the Author):

Overall, the authors made a big effort to answer the referees' questions, most of them, in particular the technical questions, convincingly. Unfortunately, some answers still remain obscure and are not easy to understand. I list below the most important points:

Reply to ref. 1, question about amplitudon and phason

The authors insert an explanation of the concept of amplitudons and phasons in the manuscript. Yet, the connection to the observed 1D, 2D, 3D CDW phases remains open. From the presented eigenmodes, I fail to see a clear connection to either an amplitudon or phason.

Reply to ref. 1: point 6:

The referee asked about the wave vectors of the 2D and 3D CDW phases, given that only the wave vector $q_{\text{CDW}} = 0.4a^*$ of the 1CDW was given. The authors answer that the wave vectors of the 2D- and 3D-CDW phases are $q_{\text{CDW}} = 0.4a^*$ and $q_{\text{CDW}} = 0.4a^* + 0.5c^*$, respectively. But that means that the wave vector of the 1D and 2D CDW are the same. What is then the difference between the 1D and 2D CDW? Maybe there is an overall misconception (of the reader or of the authors) of the dimensionality of the CDW. Looking at Fig. 4 e of the revised manuscript, it seems that the relation between "1D" and "3D" CDWs is an oscillation in-phase or out-of-phase between neighboring layers (separated by an intermediate regime with loose phase relation between neighboring layers). If this is correct, then I would not call the different CDW regimes "1D" and "3D" but rather in phase and out of phase.

The authors' reply to question 1 of referee 2 (about the "dimensionality" of the CDWs) seems to indicate that they have somehow changed/adapted their interpretation of their data. However, this is not yet reflected in a change of the abstract which continues to contain statements like "However, there is yet to be a complete CDW phase diagram in various dimensions" which insinuates that this manuscript would contain such an analysis which is, however, not the case.

Reply to ref. 3, point 2 (question about last statement of the abstract):

The phrase has been replaced by essentially the same phrase (only deleting the word "hidden"). My point was that this phrase "This study shows that the CDW phases with various dimensions and their transition mechanisms are crucial for CDW materials" is a drastic and unsubstantiated overstatement of the results of the current manuscript. While it is certainly interesting that CuTe contains apparently different CDW regimes and while it is a good research work to measure those and shine some light on their mechanism, this does not mean that they are "crucial" for the understanding of other CDW materials in which the mechanism is often completely different and most of which only show exactly one CDW transition.

In summary: My impression is that the experimental data has been carefully measured and deserves publication to a wider audience. Yet, the interpretation and theoretical discussion of the CDW regimes remains obscure and below the level that warrants publication in Nature Communications.

My main advice to the authors: please make sure that your abstract and final conclusions accurately reflect what the paper achieves (and what it does not achieve): the paper is potentially a very valuable contribution to clarifying the different CDW regimes of CuTe (though I probably would not

call them 1D and 3D). It is NOT a "complete CDW phase diagram in various dimensions" of CDW materials in general.

Responses to Reviewer #1:

We appreciate the Reviewer's time and support. The detailed responses to the Reviewer's additional suggestions are provided as follows.

(1) Reviewer's comments:

the possibility of pure phase of CuTe. As CuTe/Cu₂Te has an interesting phase diagram with composition [ACS Applied Energy Materials 3, 3, 2175(2020)], thus a comparison and comment on obtaining the phase pure CuTe would be good.

Responses:

Thank the reviewer's suggestion. We have added the characterization results of CuTe crystals to the *Supplementary Information (section S8)* and made a comparison with the Cu₂Te, which was also mentioned at the end of the *Sample preparation section in the Methods of the main text*.

(2) Reviewer's comments:

Raman spectra in the SI is not elaborated for the peak identifications. CDW has several interesting modes [Physical Review Research, 02, 033118 (2020)] including amplitude and phase modes. Some discussion on the assignment of the modes in SI would make it better. Authors may like to consider it.

Responses:

Thank the reviewer for reminding us of the identifications of the modes in Raman spectra. The discussion on the assignment of the Raman modes has been added in *the 2nd paragraph and Fig. S4 in section S3 of Supplementary Information*.

Responses to Reviewer #2:

We appreciate the Reviewer's time and comments, which helped us to clarify the missing point in the previous manuscript. The detailed responses to the Reviewer's concern are provided as follows.

(1) Reviewer's comments:

My first concern is still about the "3D CDW" proposed by the paper. In this 1D system, the "3D" CDW means increased CDW dimensionality along the other two directions, namely the c and b directions, which I will elaborate separately about my concern as followings:

As for the c direction, the authors used ARPES data to show the CDW gap feature in this direction. Combined with the behavior of collective modes in the pump probe spectroscopy, I can agree that there may be increased CDW dimensionality along c axis. In previous publications, the q_{CDW} was always reported to be $(0.4a^*, 0, 0.5c^*)$. However, the authors said there is a change of q_{CDW} from $1c^*$ to $0.5c^*$ below 220 K, which I have not seen in the paper direct experimental evidence of how the different wavevectors are determined. The one evidence of the wavevector change is the theoretical calculation, but it can only be viewed as a "theoretical prediction" without any experimental evidence.

As for the b direction. I go through the paper and find almost no proof of the CDW dimensionality along the b-direction. A weak proof is claimed on page 11 as "In the STM image, the 5-Te-atoms modulation chain along the a-axis is repeated well along the b-axis. It demonstrates a 2D CDW formed via the interchain coupling". I think the repetition of chains in the b direction cannot prove the 2D CDWs, since in STM measurements on other 1D CDW materials the 1D CDW chains are also repeated. Besides, in the in-plane anisotropic optical conductivity measurements (ref 26), the CDW gap feature is only present in a direction and always absent in b direction, which is against the "2D" CDWs in this paper. Therefore, even if the claim of CDW along c direction can possibly be true, the proposed "3D" CDW is not reasonable considering lack of evidence of the b direction.

Responses:

Before explicitly addressing the reviewer's concerns on the CDW dimensionality, we first clarify one argument of "there is a change of q_{CDW} from $1c^*$ to $0.5c^*$ below 220 K" mentioned by the reviewer. We performed the non-resonant x-ray diffraction to measure the CDW peak with the modulation vector $q_{CDW} = (0.4, 0, 0.5)$. The acquisition of sufficient data necessitates

the high-brightness synchrotron x-ray source. Therefore, all the non-resonant XRD measurements were executed at TPS 09A in NSRRC (National Synchrotron Radiation Research Center, <https://www.nsrcc.org.tw/english/index.aspx>) with an x-ray energy of 12 keV ($\lambda=1.0332 \text{ \AA}$). The CuTe single crystal was mounted in a closed-cycle cryostat on a six-circle Huber diffractometer, enabling temperature control from 100 K to 300 K. Fig. R1a shows that the L-scan across CDW modulation vector $q_{\text{CDW}} = (0.4, 0, 0.5)$ were measured along the $(0, 0, L)$ direction of one CuTe single crystal at different temperatures. The $(0, 0, 0.5L)$ peak significantly disappears while $T > 200 \text{ K}$, which clearly indicates the phase transition along the c -axis accompanying the c -axis doubling (i.e., $1c \text{ unit} \rightarrow 2c \text{ unit}$) at low temperatures. This change of spatial periodicity further creates zone-folded (ZF) modes near the zone center (see Fig. S4 in Supplementary section S3). Moreover, the $(0.4, 0, 2)$ peak can still be observed at 300 K, as shown in Fig. R1b. This demonstrates that the $q_{\text{CDW}} = 0.4a^*$ can subsist at $T > 200 \text{ K}$, even if there is no $q_{\text{CDW}} = 0.5c^*$.

Fig. R1. **a** L-scan across CDW modulation vector $q_{\text{CDW}} = (0.4, 0, 0.5)$ was measured along the $(0, 0, L)$ direction of one CuTe single crystal at different temperatures, indicating the phase transition along the c -axis. **b** The x-ray diffraction (θ - 2θ scan) peak of $(0.4, 0, 2)$ at 300 K. (*This figure has been added to section S8 (Fig. S14) of Supplementary Information.*)

Second, we refer to Fig. 5.5 in the cornerstone book on CDWs by G. Grüner [1] and copy it in the following (see Fig. R2) for the convenience of our discussions. As indicated in Fig. R2, CDWs are subject to ubiquitous fluctuations at high temperatures and render pertinent

transitions into a three-stage character by the set-in of respective ordering features at T_{MF} , T^* , and $T_{3\text{D}}$, though one conventionally considers T_{CDW} alone, with eventually $T_{\text{CDW}} \approx T^*$ ($T_{\text{CDW}} = 335$ K in CuTe). We start our arguments from the derivation of T_{MF} in accordance with the classical mean-field equation of $2\Delta = 3.2k_B T_{\text{MF}}$ (2Δ , the CDW gap size; k_B , the Boltzmann's constant). With the gap of ~ 150 meV in CuTe, T_{MF} is estimated to be ~ 494 K. At $T_{\text{CDW}} \approx T^* < T < T_{\text{MF}}$, CuTe would readily commence to form quasi-one-dimensional chains along a -axis with the a -component of q_{CDW} being the reported $\sim 0.4a^*$. In this temperature regime, it is nonetheless noted that the chains are to be aligned (namely, in-phase as pointed out by Reviewer #3) neither along the a -axis nor along the b - and c -axes of CuTe due to the ubiquitous CDW fluctuations. At the lower temperature of $T_{\text{CDW}} \approx T^*$, the fluctuations mitigate and the CDW chains turn to be in-phase along the a -axis, casting 1D CDWs that correspond to the 1D CDW of CuTe discussed in this work. With further decreasing temperatures to $T_{3\text{D}} < T < T_{\text{CDW}} \approx T^*$, an anti-phase coupling of the neighboring CDW chains in the vertical direction (i.e., c -axis) would assist in optimizing the overall electrostatic energy and commences to develop as characterized by the q_{CDW} of $(0.4a^*, 0, 0.5c^*)$ in CuTe. Meanwhile, the CDW chains in CuTe have to be perfectly in phase along the b -axis, since no superlattice modulation along b^* has ever been reported. At $T_{3\text{D}} < T < T_{\text{CDW}} \approx T^*$, CuTe thus features a nicely in-phase ordering of CDW chains in ab -plane, i.e., the 2D CDW we refer to, and mediates an anti-phase chain coupling along c -axis, but not fully aligned yet along c -axis. Therefore, we called this regime the “quantum fluctuations (QF)” (see Fig. 4e). At $T_{3\text{D}}$, fully in-phase coupled CDW chains in ab -plane and coherently anti-phase coupled ones along the c -axis are to be established in CuTe and the CDWs were coined as 3D CDWs by G. Grüner. In CuTe, such a CDW transition with the 3D identity is found in our spectroscopic measurements below ~ 220 K, suggesting $T_{3\text{D}} \sim 220$ K, and we borrowed all the above classical knowledge by G. Grüner for the descriptions of 1D, 2D(QF), and 3D CDWs, regrettably without having properly narrated the details in the last version.

In this revision, we have accordingly improved all relevant texts (including Abstract) and Fig. 4e.

Fig. R2. Fluctuations in a coupled chain system in the 1D and 3D fluctuation regime and the corresponding long-range correlations below T_{3D} . The main features of the diffraction pattern are also displayed in the different temperature regions. [1]

(2) Reviewer's comments:

About the results shown in Fig. 4ab: The shift of the mode frequency with increasing incident angle looks quite novel and I have not seen such results in previous pump probe studies in CDW systems. For me this is one of the most interesting results of this work and deserves to be discussed more in the paper. But I am puzzled by the explained $\lambda\theta=1/f\theta$ scenario in

Supplementary section S7, which looks quite unreasonable. According to the scenario, does it seem that such frequency shift could appear in any CDW systems? And, when rotating the E field direction in the ab plane (instead of changing incident angle), will the mode frequency also shift according to $\lambda\theta=1/\theta$? Indeed, I have never seen such examples in CDW systems before. Moreover, what is the relation between the frequency shift and the CDW dimensionality? This is almost not discussed in the main text except for a concluding sentence “These results show that the high-frequency amplitudon at around 2 THz can indicate the interaction between two neighboring CDW layers along the c-axis.

Responses:

Thank the reviewer for pointing out the missing part, which we needed to have properly narrated in the last responses. A detailed discussion of this issue is provided as follows.

Fig. R3a represents the configuration of a CDW probed by a probe beam with various incident angles θ . First, in the normal incident case with $\theta = 0^\circ$, a propagation direction of probe pulses $\mathbf{k} // c$ -axis and electric field $\mathbf{E} // a$ -axis ($\perp c$ -axis and \mathbf{P}_c). Here, the \mathbf{P}_c is an electric dipole, which is caused by the displacements of Cu and Te atoms along the c -axis, as shown in **Fig. S7** (in Supplementary section S6). Thus, there is no electric field \mathbf{E} to interact with the \mathbf{P}_c according to the relation of $\Delta R/R_c \propto \mathbf{E} \cdot \mathbf{P}_c = |\mathbf{E}||\mathbf{P}_c|\cos(90-\theta) = 0$ with $\theta = 0^\circ$.

For the case of oblique incidence, the \mathbf{E} field of probe pulses can be decomposed into two perpendicular components: \mathbf{E}_1 and \mathbf{E}_2 . \mathbf{E}_2 is parallel to the a -axis and \mathbf{P}_a (caused by the displacements of Cu and Te atoms along the a -axis as shown in **Fig. S7** of Supplementary section S6), and \mathbf{E}_1 is parallel to the c -axis and \mathbf{P}_c , which discloses the CDW along the c -axis with a wavevector of q_c ($\because q_c // \mathbf{P}_c$). Under this configuration, the $\Delta R/R_c \propto \mathbf{E} \cdot \mathbf{P}_c = |\mathbf{E}||\mathbf{P}_c|\cos(90-\theta) \neq 0$ with $\theta \neq 0^\circ$ and the $\Delta R/R_c$ will become larger when increasing the incident angle θ as demonstrated by **Fig. 4b** in the main text. Meanwhile, the effective propagation vector of \mathbf{E}_1 is \mathbf{k}_1 , which increases with increasing the incident angle θ . According to the assignment of the

Raman modes in Fig. S4 (in Supplementary section S3), the 2.25-THz mode is a zone-folded (ZF) mode A_{ZF1} due to the c -axis doubling in Fig. R1a below 220 K, e.g., the red lines in the inset of Fig. R3c. The crossing point between phonon dispersion (a red line in the inset of Fig. R3c) and photon dispersion (green solid and dot-dashed lines in the inset of Fig. R3c) curves marked by the circle dots, where the phonon modes can be detected by probe pulses. If the k_1 increases with increasing the incident angle θ , the circle dots (in the inset of Fig. R3c) will move to a new position with a larger momentum k and lower energy E (or frequency), e.g., from the yellow dot to the green dot in the inset of Fig. R3c. This result has been observed in Fig. R3c and Fig. 4b in the main text.

According to the above scenario, we expect no frequency shift on the ab -plane. Fig. R4 shows the Fourier transform spectra of the oscillation components of $\Delta R/R$ spectra at different ϕ_1 (ϕ_2) angles, i.e., on ab -plane ($\phi_1 = \phi_2 = 0^\circ$ for a -axis; $\phi_1 = \phi_2 = 90^\circ$ for b -axis), in Fig. 1. As we can see, the A_{HF} (~ 2.28 THz) peak is gradually disappearing when ϕ_1 ($= \phi_2$) increases from 0° to 90° with the same frequency. This is because the incident angle θ of probe pulses is fixed to keep the same propagation direction \mathbf{k} , which has the same crossing point between phonon dispersion and photon dispersion curves in the inset of Fig. R3c.

In summary, if a CDW with a specific direction q is parallel with an electric dipole \mathbf{P} (caused by the displacements of Cu and Te atoms), it can be detected by a laser pulse with \mathbf{E} field according to the relation of $\Delta R/R \propto \mathbf{E} \cdot \mathbf{P}$. Meanwhile, if the propagation direction of probe pulses \mathbf{k} also changes, the crossing point between phonon dispersion and photon dispersion curves will vary, resulting in the frequency shift, which depends on the photon and phonon dispersion curves. Consequently, it will not be observed in all CDW systems and dimensions.

The above discussion has been added to the 2nd paragraph on page 14 and the 1st paragraph on page 15 of this revised manuscript and updated in the Supplementary Information (section S7).

Fig. R3. **a** Schematics of the configuration of a 3D CDW probed by a probe pulse with various incident angles θ and electric field $\mathbf{E} \perp b$ -axis. \mathbf{E}_1 ($\parallel c$ -axis and $\parallel \mathbf{P}_c$) and \mathbf{E}_2 ($\parallel a$ -axis and $\parallel \mathbf{P}_a$) are the components of the electric field \mathbf{E} of a probe pulse. \mathbf{k} is the propagation direction of the probe pulses. \mathbf{k}_1 ($\parallel a$ -axis and $\perp \mathbf{P}_c$) and \mathbf{k}_2 ($\parallel c$ -axis and $\perp \mathbf{P}_a$) are the components of the propagation direction \mathbf{k} of probe pulses. **b** Schematics of a 3D CDW probed by a probe pulse with various incident angles θ and electric field $\mathbf{E} \parallel b$ -axis. \mathbf{k} is the propagation direction of the probe pulses. **c** Frequency of A_{HF} peak in Fig. 4b (in main text) as a function of incident angles θ of the probe pulses. Inset: schematic demonstrating the change in the phonon dispersion (blue line) from the undistorted state with a zone edge at the gray dashed line to the CDW state where the spatial periodicity increases, thereby decreasing the first Brillouin zone to the black dashed line. The red lines show how the phonon bands change and create zone-folded (ZF) modes near the zone center (see Fig. S4 in Supplementary section S3). The circle

dots show the crossing points between phonon dispersion (blue line) and photon dispersion (green solid and dot-dashed lines) curves, where the phonon modes can be detected by probe pulses. k_1 is the component of the propagation direction \mathbf{k} of probe pulses in **a**. **d** Normalized oscillation components in $\Delta R/R$ spectra of a CuTe single crystal at different incident angles, θ (ϕ_1 and $\phi_2 = 90^\circ$, i.e., the configuration in **b**) at $T = 60$ K. Inset: Fourier transform spectra of the oscillation spectra in **d**. (*This figure has been updated to section S7 (Fig. S12) of Supplementary Information.*)

Fig. R4. Fourier transform spectra of the oscillation components of $\Delta R/R$ spectra at different ϕ_1 (ϕ_2) angles, i.e., on ab -plane ($\phi_1 = \phi_2 = 0^\circ$ for a -axis; $\phi_1 = \phi_2 = 90^\circ$ for b -axis), in Fig. 1 (main text).

(3) Reviewer's comments:

In all, since the authors do not convince me of the proposed “3D” CDW, which is the main idea of the paper, I think the paper is not suitable for publication in Nature Communications. However, since the work contains some meaningful results, it can still be suitable for publication in a more specialized journal.

Responses:

From the above discussion and analyses, especially G. Grüner's scenario, we can figure out that the changes in periodicity of the atoms (i.e., the superlattice or a CDW wavevector) in Peierls' model cannot be applied to all CDW systems, especially for the high dimensional cases [2-5]. This means that if the CDW phases do not accompany the superlattice or without a

specific CDW wavevector, it cannot be directly resolved by conventional methods, such as diffraction spectroscopy, etc. For example, except for the Bragg peak from the original lattice, there is no report to show the new superlattice or CDW wavevector along the b -axis in CuTe. However, it has been unambiguously disclosed by the orientation- and time-resolved ultrafast spectroscopy in this work, which shows the novelty and uniqueness of studies in high-dimensional CDW systems.

In summary, the new manuscript version has addressed and clarified all points raised by the reviewer, which has reached the level of thoroughness and quality suitable for Nature Communications. We really appreciate the Reviewer's encouragement and desire your support for the publication in Nature Communications.

References

- [1] G. Grüner, Density waves in solid. Frontiers in physics Ed D. Pines (Addison-Wesley), p.99 (1994).
- [2] Zhu, X., Cao, Y., Zhang, J., Plummer, E. W. & Guo, J. Classification of charge density waves based on their nature. *Proc. Natl. Acad. Sci.* **112**, 2367-2371 (2015).
- [3] Cudazzo, P. & Wirtz, L. Collective electronic excitations in charge density wave systems: The case of CuTe. *Phys. Rev. B* **104**, 125101 (2021).
- [4] Kogar, A. et al. Signatures of exciton condensation in a transition metal dichalcogenide. *Science* **358**, 1314-1317 (2017).
- [5] Zhu, X., Guo, J., Zhang, J. & Plummer, E. W. Misconceptions associated with the origin of charge density wave. *Advances in Physics: X* **2**, 622-640 (2017).

Responses to Reviewer #3:

We appreciate the Reviewer's time and suggestions in the comment letter again. The detailed responses to the Reviewer's concern are provided as follows.

(1) Reviewer's comments:

Reply to ref. 1, question about amplitudon and phason

The authors insert an explanation of the concept of amplitudons and phasons in the manuscript. Yet, the connection to the observed 1D, 2D, 3D CDW phases remains open. From the presented eigenmodes, I fail to see a clear connection to either an amplitudon or phason.

Responses:

The amplitudons and phasons introduced in section S1 of Supplementary Information (Fig. S1) are based on the 1D crystal chain, which means that all the amplitudons and phasons (i.e., a kind of electronic order parameter and CDW) are modulated in 1D space, e.g., along the z-axis in Fig. S2 of Supplementary Information. Additionally, this electronic order parameter (i.e., CDW, q) would couple with a 1D phonon with a lattice vibration, which further forms an electric dipole \mathbf{P} . In our case, for example, the electric dipole \mathbf{P}_a is caused by the displacements of Cu and Te atoms along the a -axis, as shown in Fig. S7 (in Supplementary section S6).

According to the interaction between the electric field \mathbf{E} of probe pulses and an electric dipole \mathbf{P} , we can have the results of $\Delta R/R_a \propto \mathbf{E} \cdot \mathbf{P}_a$ (or $\mathbf{E} \cdot \mathbf{q}_a$). Besides, the $\Delta R/R_b \propto \mathbf{E} \cdot \mathbf{q}_b$ and $\Delta R/R_c \propto \mathbf{E} \cdot \mathbf{q}_c$ can be further realized via the polarization manipulation of probe pulses as described in Supplementary section S7, which is the novelty in this work. If only one of the $\Delta R/R_a$, $\Delta R/R_b$, and $\Delta R/R_c$ along three orientations (dimensions) can be observed, it is just a kind of 1D CDW. If two of the $\Delta R/R_a$, $\Delta R/R_b$, and $\Delta R/R_c$ along three orientations (dimensions) can be observed, it should be a 2D CDW. Finally, if all the $\Delta R/R_a$, $\Delta R/R_b$, and $\Delta R/R_c$ along three orientations (dimensions) can be observed, it is a 3D CDW.

The above discussion has been added to the last paragraph in section S7 of Supplementary Information.

(2) Reviewer's comments:

Reply to ref. 1: point 6:

The referee asked about the wave vectors of the 2D and 3D CDW phases, given that only the wave vector $q_{\text{CDW}} = 0.4a^*$ of the 1CDW was given. The authors answer that the wave vectors of the 2D- and 3D-CDW phases are $q_{\text{CDW}} = 0.4a^*$ and $q_{\text{CDW}} = 0.4a^* + 0.5c^*$, respectively. But that means that the wave vector of the 1D and 2D CDW are the same. What is then the difference between the 1D and 2D CDW? Maybe there is an overall misconception (of the reader or of the authors) of the dimensionality of the CDW. Looking at Fig. 4 e of the revised manuscript, it seems that the relation between "1D" and "3D" CDWs is an oscillation in-phase or out-of-phase between neighboring layers (separated by an intermediate regime with loose phase relation between neighboring layers). If this is correct, then I would not call the different CDW regimes "1D" and "3D" but rather in phase and out of phase.

Responses:

The authors appreciate the reviewer for raising the crucial point about CDW dimensionality. First, we would like to emphasize that the CDW wavevector $q_{\text{CDW}} = 0.4a^* + 0.5c^*$ is the result of periodicity changes of the atoms along a specific orientation based on Peierls' model. However, Peierls' scenario with a particular CDW wavevector q_{CDW} and Fermi surface nesting (FSN) cannot be applied to all CDW systems, especially for the high dimensional cases [1-4]. This means that we can still observe the CDW phases in some CDW systems or along some specific orientations even without a particular CDW wavevector q_{CDW} or with the same periodicity of Bragg lattice (in which we cannot distinguish it from the periodicity of CDW modulation). For example, CuTe is the most interesting system among CDW systems, in which only two CDW wavevectors (q_{CDW}) can be found along a - and c -axes by diffraction technique. However, the CDW along the b -axis had been observed by the orientation- and time-resolved ultrafast spectroscopy in this study, which can disclose the CDW phases, whether there is a CDW wavevector q_{CDW} (based on Peierls' model) or not. This is the novelty and uniqueness of studying the high-dimensional CDW systems in this work.

To explicitly address the reviewer's concerns on the issue of the difference between 1D and 2D, we refer to Fig. 5.5 in the cornerstone book on CDWs by G. Grüner [5] and copy it in the following (see Fig. R1) for the convenience of our discussions. As indicated in Fig. R1,

CDWs are subject to ubiquitous fluctuations at high temperatures and render pertinent transitions into a three-stage character by the set-in of respective ordering features at T_{MF} , T^* , and $T_{3\text{D}}$, though one conventionally considers T_{CDW} alone, with eventually $T_{\text{CDW}} \approx T^*$ ($T_{\text{CDW}} = 335$ K in CuTe). We start our arguments from the derivation of T_{MF} in accordance with the classical mean-field equation of $2\Delta = 3.2k_B T_{\text{MF}}$ (2Δ , the CDW gap size; k_B , the Boltzmann's constant). With the gap of ~ 150 meV in CuTe, T_{MF} is estimated to be ~ 494 K. At $T_{\text{CDW}} \approx T^* < T < T_{\text{MF}}$, CuTe would readily commence to form quasi-one-dimensional chains along a -axis with the a -component of q_{CDW} being the reported $\sim 0.4a^*$. In this temperature regime, it is nonetheless noted that the chains are to be aligned (namely, in-phase as pointed out by the Reviewer) neither along the a -axis nor along the b - and c -axes of CuTe due to the ubiquitous CDW fluctuations. At the lower temperature of $T_{\text{CDW}} \approx T^*$, the fluctuations mitigate and the CDW chains turn to be in-phase along the a -axis, casting 1D CDWs that correspond to the 1D CDW of CuTe discussed in this work. With further decreasing temperatures to $T_{3\text{D}} < T < T_{\text{CDW}} \approx T^*$, an anti-phase coupling of the neighboring CDW chains in the vertical direction (i.e., c -axis) would assist in optimizing the overall electrostatic energy and commences to develop as characterized by the q_{CDW} of $(0.4a^*, 0, 0.5c^*)$ in CuTe. Meanwhile, the CDW chains in CuTe have to be perfectly in phase along the b -axis, since no superlattice modulation along b^* has ever been reported. At $T_{3\text{D}} < T < T_{\text{CDW}} \approx T^*$, CuTe thus features a nicely in-phase ordering of CDW chains in ab -plane, i.e., the 2D CDW we refer to, and mediates an anti-phase chain coupling along c -axis, but not fully aligned yet along c -axis. Therefore, we called this regime the “quantum fluctuations (QF)” (see Fig. 4e). At $T_{3\text{D}}$, fully in-phase coupled CDW chains in ab -plane and coherently anti-phase coupled ones along the c -axis are to be established in CuTe and the CDWs were coined as 3D CDWs by G. Grüner. In CuTe, such a CDW transition with the 3D identity is found in our spectroscopic measurements below ~ 220 K, suggesting $T_{3\text{D}} \sim 220$ K, and we borrowed all the above classical knowledge by G. Grüner for the descriptions of 1D, 2D(QF), and 3D CDWs, regrettably without having properly narrated the details in the last version.

In this revision, we have accordingly improved all relevant texts (including Abstract) and Fig. 4e.

Fig. R1. Fluctuations in a coupled chain system in the 1D and 3D fluctuation regime and the corresponding long-range correlations below T_{3D} . The main features of the diffraction pattern are also displayed in the different temperature regions. [5]

(3) Reviewer's comments:

The authors' reply to question 1 of referee 2 (about the "dimensionality" of the CDWs) seems to indicate that they have somehow changed/adapted their interpretation of their data. However, this is not yet reflected in a change of the abstract which continues to contain statements like "However, there is yet to be a complete CDW phase diagram in various dimensions" which insinuates that this manuscript would contain such an analysis which is, however, not the case.

Responses:

Thank the reviewer's reminder. In order to avoid confusion for readers, the sentence "However, there is yet to be a complete CDW phase diagram in various dimensions, ..." has been revised to "However, CDW phase dynamics in various dimensions are yet to be studied, ...". (*The 3-4 lines in the Abstract of this revised manuscript*)

(4) Reviewer's comments:

Reply to ref. 3, point 2 (question about last statement of the abstract):

The phrase has been replaced by essentially the same phrase (only deleting the word "hidden"). My point was that this phrase "This study shows that the CDW phases with various dimensions and their transition mechanisms are crucial for CDW materials" is a drastic and unsubstantiated overstatement of the results of the current manuscript. While it is certainly interesting that CuTe contains apparently different CDW regimes and while it is a good research work to measure those and shine some light on their mechanism, this does not mean that they are "crucial" for the understanding of other CDW materials in which the mechanism is often completely different and most of which only show exactly one CDW transition.

Responses:

The authors are grateful for the suggestion of the reviewer to allow for a chance to revise the abstract again. The sentence "This study shows that the CDW phases with various dimensions and their transition mechanisms are crucial for CDW materials." has been revised to "This study shows the dimension evolution of CDW phases in one CDW system and their stabilized mechanisms in different temperature regimes.". (*The last 2 lines in the Abstract of this revised manuscript*)

(5) Reviewer's comments:

In summary: My impression is that the experimental data has been carefully measured and deserves publication to a wider audience. Yet, the interpretation and theoretical discussion of

the CDW regimes remains obscure and below the level that warrants publication in Nature Communications.

My main advice to the authors: please make sure that your abstract and final conclusions accurately reflect what the paper achieves (and what it does not achieve): the paper is potentially a very valuable contribution to clarifying the different CDW regimes of CuTe (though I probably would not call them 1D and 3D). It is NOT a "complete CDW phase diagram in various dimensions" of CDW materials in general.

Responses:

The authors are grateful for the reviewer's kind advice and encouragement. We have carefully clarified and implemented all suggestions and points raised by the reviewer throughout the whole manuscript. The new manuscript version has been polished significantly based on the reviewer's advice and has reached the level of thoroughness and quality suitable for publication in Nature Communications.

References

- [1] Zhu, X., Cao, Y., Zhang, J., Plummer, E. W. & Guo, J. Classification of charge density waves based on their nature. *Proc. Natl. Acad. Sci.* **112**, 2367-2371 (2015).
- [2] Cudazzo, P. & Wirtz, L. Collective electronic excitations in charge density wave systems: The case of CuTe. *Phys. Rev. B* **104**, 125101 (2021).
- [3] Kogar, A. et al. Signatures of exciton condensation in a transition metal dichalcogenide. *Science* **358**, 1314-1317 (2017).
- [4] Zhu, X., Guo, J., Zhang, J. & Plummer, E. W. Misconceptions associated with the origin of charge density wave. *Advances in Physics: X* **2**, 622-640 (2017).
- [5] G. Grüner, Density waves in solid. *Frontiers in physics* Ed D. Pines (Addison-Wesley), p.99 (1994).

List of changes in the revised manuscript

Here, let us briefly summarize the major points and changes as follows.

(1) The co-author's affiliations have been revised respectively.

Author list on previous submission: Nguyen Nhat Quyen¹, ..., Cheng-En Liu¹, Wei-Liang Chen⁵, ..., Ping-Hui Lin⁴, Chang-Yang Kuo^{1,4}, ..., Chih-Wei Luo^{* 1,4,9,10,11}

Author list on revised manuscript: Nguyen Nhat Quyen¹, ..., Cheng-En Liu¹, **Yu-Sheng Chen⁴**, Wei-Liang Chen⁵, ..., Ping-Hui Lin⁴, **Shih-Chang Weng⁴**, Chang-Yang Kuo^{1,4}, ..., Chih-Wei Luo^{* 1,4,9,10,11}

(2) *lines 4-5 (page 3, Abstract of previous manuscript)* “However, there is yet to be a complete CDW phase diagram in various dimensions, and ... currently moot.” *was revised to* “**However, CDW phase dynamics in various dimensions are yet to be studied,** and ... currently moot.”

(3) *lines 13-14 (page 3, Abstract of previous manuscript)* “This study shows that the CDW phases with various dimensions and their transition mechanisms are crucial for CDW materials.” *was revised to* “**This study shows the dimension evolution of CDW phases in one CDW system and their stabilized mechanisms in different temperature regimes.**”

(4) *lines 4 (page 4 of the previous manuscript)* “..., 2D transition metal dichalcogenides (TMDCs)...” *was revised to* “..., **two-dimensional (2D)** transition metal dichalcogenides (TMDCs)...”

(5) *lines 16-17 (page 4 of the previous manuscript)* “..., the order parameter (OP) instability with the spontaneously broken symmetry (SBS) has gained attention recently.” *was revised to* “..., the order parameter instability with **spontaneously broken symmetry (SBS)** has gained attention recently.”

(6) *lines 2-3 (page 5 of the previous manuscript)* “As shown in Fig. S1 (in Supplementary Information), the phasor...” *was revised to* “**As shown in Fig. S1,** the phasor...”

(7) *lines 4 (page 5 of the previous manuscript)* “...the phase (ϕ) modulation of charge density (ρ) wave, and...” *was revised to* “...the phase (ϕ) modulation of **CDW (ρ)**, and...”

(8) *lines 6-7 (page 5 of the previous manuscript)* “...the amplitude (Δ) modulation of charge density (ρ) wave and the changes of V with the same ϕ in the Mexican hat-like (double-well) potential. As a result, the phase and amplitude modes are IR-active and Raman-active modes, respectively¹⁸.” *was revised to* “...the amplitude (Δ) modulation of **CDW (ρ)** and the changes of V with the same ϕ **in the double-well potential**. As a result, the phase and amplitude modes are **IR-** and Raman-active modes, respectively¹⁸.”

- (9) *lines 10 (page 5 of the previous manuscript)* “...(with a CDW gap of ~190 meV)...” *was revised to* “...(with **a gap** of ~190 meV)...”
- (10) *lines 16 (page 5 of the previous manuscript)* “...(see Supplementary section S6 for more discussion). Using...” *was revised to* “...(see **Supplementary section S6**). Using...”
- (11) *lines 5-6 (page 6 of the previous manuscript)* “... to determine the hidden CDW phases with various...” *was revised to* “...to determine **the CDW phases** with various...”
- (12) *lines 12-13 (page 6 of the previous manuscript)* “...in CDW materials^{4,30-39} (for details, see Supplementary section S1).” *was revised to* “...in CDW materials^{4,30-39} (**see Supplementary section S1**).”
- (13) *lines 1-5 (page 7 of the previous manuscript)* “STM mapping ... CDW with $q_{\text{CDW}}=0.4a^*$ and further reveals a 2D CDW on the ab -plane due to an interchain interaction between 1D-CDW chains. Moreover, a 3D CDW with $q_{\text{CDW}} = 0.4a^* + 0.5c^*$ was first observed at low temperatures using orientation-/time-resolved ultrafast spectroscopy and has been verified by first-principles calculations for...” *was revised to* “STM mapping ... **CDW with $q_{\text{CDW}}=0.4a^*$** . Moreover, a **3D CDW** was first observed at low temperatures using orientation-/time-resolved ultrafast spectroscopy **and verified by** first-principles calculations for...”
- (14) *lines 10-11 (page 7 of the previous manuscript)* “...made along the a -axis and the b -axis (see Fig. S1a in Supplementary Information) at various temperatures,...” *was revised to* “...made along the **a - and b -axes** (see **Fig. S3a**) at various temperatures,...”
- (15) *lines 1 (page 9 of the previous manuscript)* “...data in Fig. 2cd (for details, see Supplementary section S5). After...” *was revised to* “...data in Fig. 2cd (**see Supplementary section S5**). After...”
- (16) *lines 6 (page 10 of the previous manuscript)* “...associated with the A_1 mode, as...” *was revised to* “...associated with the A_1 (**A_{am}**) mode, as...”
- (17) *lines 3 (page 11 of the previous manuscript)* “...using first-principles calculations (for details, see Supplementary section S6) and several...” *was revised to* “...using first-principles calculations (**see Supplementary section S6**) and several...”
- (18) *lines 8-10 (page 11 of the previous manuscript)* “The higher frequency 2.26-THz mode that exhibits the same softening behavior as the 1.64-THz is eliminated at temperatures above 220 K. First-principles ... with a new A_g mode (see Fig. 3e), for which...” *was*

- revised to* “The 2.26-THz mode is eliminated at $T > 220$ K. First-principles ... with a new A_g (A_{ZF1}) mode (see Fig. 3e and Fig. S4), for which...”
- (19) *lines 13-14 (page 11 of the previous manuscript)* “...to form a 3D CDW (CDW_{3D}) at temperatures below 220 K with the atomic configuration that is shown in Fig. 3e. The...” *was revised to* “...to form a 3D CDW (CDW_{3D}) at $T < 220$ K with the atomic configuration shown in Fig. 3e. The...”
- (20) *lines 9 (page 11 of the previous manuscript)* “...a more stable 3D CDW at temperatures below 220 K.” *was revised to* “...a more stable 3D CDW at $T < 220$ K, consistent with XRD results in Fig. S14a.”
- (21) *lines 2 (page 12 of the previous manuscript)* “...by the lattice specific heat at temperatures below ~ 220 K,...” *was revised to* “...by the lattice-specific heat at $T < \sim 220$ K,...”
- (22) *lines 4 (page 12 of the previous manuscript)* “...suffers dimensional fluctuations in the intermediate...” *was revised to* “...suffers dimensional fluctuations (from 1D to 3D via the formation of 2D) in the intermediate...”
- (23) *lines 6 (page 12 of the previous manuscript)* “...a stable 3D CDW at temperatures below 220 K.” *was revised to* “...a stable 3D CDW at $T < 220$ K.”
- (24) *lines 8-9 (page 12 of the previous manuscript)* “...the data in Fig. 3a (for details, see Supplementary section S5).” *was revised to* “...the data in Fig. 3a (see Supplementary section S5).”
- (25) *lines 9 (page 13 of the previous manuscript)* “...1D systems¹⁰. At temperatures below ~ 300 K, which is...” *was revised to* “...1D systems¹⁰. At $T < \sim 300$ K, which is...”
- (26) *lines 3 (page 14 of the previous manuscript)* “..., such as the A_1 mode,...” *was revised to* “..., such as the A_{am} mode,...”
- (27) *lines 7 (page 14 of the previous manuscript)* “Finally, the appearance of 2.28-THz amplitudon indicates that...” *was revised to* “Finally, the appearance of 2.26-THz mode indicates that ...”
- (28) *lines 8-9 (page 14 of the previous manuscript)* “...along the c -axis at temperatures below 220 K.” *was revised to* “...along the c -axis at $T < 220$ K.”
- (29) *lines 10 (page 14 of the previous manuscript)* “To determine whether the 2.28-THz

amplitudon in Fig. 3a...” *was revised* “To determine whether the 2.26-THz A_{ZF1} mode in Fig. 3a...”

(30) *lines 12 (page 14 of the previous manuscript)* “..., as shown in Fig. 1 (also Supplementary Fig. S12)...” *was revised to* “..., as shown in Fig. 1 (also Fig. S12)...”

(31) *lines 14-16 (page 14 of the previous manuscript)* “..., which discloses the contribution of the amplitudon along the c -axis. Fig. 4a shows the $\Delta R/R$ spectra for θ from 8° to 45° . After FFT, two amplitudons are observed for $\theta=8^\circ, 22^\circ, 30^\circ$, and 45° .” *was revised to* “..., which discloses the CDW along the c -axis, q_c . Fig. 4a shows the $\Delta R/R$ spectra for θ from 8° to 45° . After FFT, two modes are observed for $\theta=8^\circ, 22^\circ, 30^\circ$, and 45° .”

(32) *lines 16-18 (page 14 of the previous manuscript)* “...Peak fitting reveals two amplitudons for $\theta=8^\circ$ with frequencies of 1.61 and 2.32 THz. The high-frequency amplitudon peak (A_{HF}) is much shorter than that for the 1.61-THz amplitudon peak (A_{LF}): $A_{HF}/A_{LF}\sim 0.03$.” *was revised to* “...Peak fitting reveals two modes for $\theta=8^\circ$ with frequencies of 1.61 and 2.32 THz. The high-frequency peak (A_{HF}) is much shorter than the low-frequency peak (A_{LF}): $A_{HF}/A_{LF}\sim 0.03$.”

(33) *lines 18-19 (page 14 of the previous manuscript) & lines 1-5 (page 15 of the previous manuscript)* “...As the value of θ increases from 8° to 45° , the high-frequency amplitudon peak undergoes a significant redshift from 2.32 to 1.92 THz, which can be well described by a cos function in Supplementary Fig. S12c, but the low-frequency amplitudon peak exhibits almost no shift. Surprisingly, the value of A_{HF}/A_{LF} increases to 0.52, which is 17.3 times larger than the value for $\theta=8^\circ$. These results show that the high-frequency amplitudon at around 2 THz can indicate the interaction between two neighboring CDW layers along the c -axis.” *was revised to* “...As the θ increases from 8° to 45° , the A_{HF} undergoes a significant redshift from 2.32 to 1.92 THz due to the variation of the propagation vector (\mathbf{k}) of the \mathbf{E}_1 field and its coupling with the zone-folded (ZF) phonon bands (see Supplementary section S7). The increase of the ratio of \mathbf{E}_1 to \mathbf{E}_2 by increasing θ further enhances the value of A_{HF}/A_{LF} to 0.52, which is 17.3 times larger than that of $\theta=8^\circ$. This demonstrates that the c -axis CDW q_c can be unambiguously revealed via the coupling between q_c and \mathbf{E} field along the c -axis. Moreover, the width of the A_{HF} mode is wider than that of the A_{LF} mode to show a shorter lifetime, which is consistent with the result in Fig. 2b and implies a stronger CDW interaction along the a -axis compared with the c -axis.”

- (34) *lines 6-7 (page 15 of the previous manuscript)* “At temperature below 220 K, the interlayer...” *was revised to* “At $T < 220$ K, the interlayer...”
- (35) *lines 9-10 (page 15 of the previous manuscript)* “...by first-principles calculations (for details, see Supplementary section S6).” *was revised to* “...by first-principles calculations (see Supplementary section S6).”
- (36) *lines 12 (page 15 of the previous manuscript)* “...difference with out-of-phase features and...” *was revised to* “...difference with anti-phase features and...”
- (37) *lines 14-15 (page 15 of the previous manuscript)* “...At 335 K, a 1D CDW along the a -axis forms with $1c$ periodicity and in-phase along the c -axis. When...” *was revised to* “...At 335 K, a 1D CDW along the a -axis forms with $1c$ periodicity. When...”
- (38) *lines 17-18 (page 15 of the previous manuscript)* “... CDW phase in dimension until 220 K²¹. The CDW will be further stabilized with $2c$ periodicity and out-of-phase along the c -axis below 220 K to form a 3D CDW due to...” *was revised to* “... CDW phase in dimension (from 1D to 3D accompanying the formation of 2D CDW on ab -plane) until 220 K^{18, 21}. The CDW will be further stabilized with $2c$ periodicity and anti-phase along the c -axis below 220 K to form a 3D CDW¹⁸ due to...”
- (39) *lines 8-10 (page 16 of the previous manuscript)* “The powder XRD pattern was obtained using a Bruker D2 Phaser with Cu $K\alpha$ radiation at room temperature. The chemical composition...X-ray (EDX) spectroscopy.” *was revised to* “The XRD measurements were carried out at TPS 09A in NSRRC with an x-ray energy of 12 keV ($\lambda = 1.0332 \text{ \AA}$) at room temperature (the results in Supplementary section S8). The chemical composition ... x-ray (EDX) spectroscopy (the results in Supplementary section S8).”
- (40) The description of **Author contributions** (*lines 1-9 on page 20*) *was revised* as follows,
“C.W.L. proposed the project. N.N.Q., ... performed the measurements of STM. C.Y.K., C.E.L., Y.S.C., and S.C.W. performed the measurements of XRD. W.L.C and ... All authors edited and approved the final manuscript.”
- (41) The description of **Code availability** *was updated* on page 20 as follows,
“All the codes that support the plots of this study are available from the corresponding authors upon reasonable request.”

REVIEWER COMMENTS

Reviewer #2 (Remarks to the Author):

The revised manuscript and the rebuttal file provide answers to most of the reviewer's questions. I appreciate the authors' big effort and additional measurements to address the reviewer's comments. However, I still have some concerns about the manuscript:

1. About the "conception" of the "3D CDW" questioned by review #2 and review #3, now the authors use the scenario in the textbook (Fig. R2) to illustrate the "3D CDW" proposed in the manuscript. However, the physical interpretation and experimental finding of this work essentially bring little new understanding to CDW physics, and lacks scientific importance. Besides, while CDW along a- and c- axes are seen by diffraction technique, I'm still not convinced of the CDW along b-axis given only the ultrafast pump probe measurements. Also in the referenced paper [Phys. Rev. B 105, 115102 (2022)], in-plane optical conductivity clearly shows that the CDW feature is absent in the b direction, this is against the authors claim of CDW along b^* . How should the authors explain the contradiction?

2. About the authors' explanation of the frequency shift described in Fig. R3c: Optical phonons have quadratic dispersion near Brillouin zone center, which can be approximated as a constant at small wavevector. Since the wavevector of probe light is very small compared to the crystal's reciprocal wavevector, the measured phonon frequency should be nearly the same, as in most pump probe measurements where the optical phonons oscillate at frequencies independent of the propagation direction k of probe light [e.g. J. Chem. Phys. 120, 4755-4758 (2004)]. Under the authors' scenario, the phonon should disperse sharply at small k , what is the reason for the unusual sharp dispersion? The author should give an explanation.

Page 19 of the supplementary material writes: "If only one of the $\Delta R/R_a$, $\Delta R/R_b$, and $\Delta R/R_c$ along three orientations (dimensions) can be observed, it is just a kind of 1D CDW. If two of the $\Delta R/R_a$, $\Delta R/R_b$, and $\Delta R/R_c$ along three orientations (dimensions) can be observed, it should be a 2D CDW. Finally, if all the $\Delta R/R_a$, $\Delta R/R_b$, and $\Delta R/R_c$ along three orientations (dimensions) can be observed, it is a 3D CDW." I disagree with this argument. In some other standard 1D CDW systems, the amplitude mode oscillations can be measured on multiple directions, see [Phys. Rev. B 107, 165115 (2023)], [Phys. Rev. Lett. 83, 800 (1999)].

In all, the current manuscript do not meet the publication criteria in terms of scientific importance and credibility, therefore I think it is not suitable for publication in Nature Communications.

Reviewer #3 (Remarks to the Author):

The authors have answered in great detail to my questions and the one of my fellow referees. One might deplore that the final manuscript did not change much. Nevertheless, the explanations made the modelization part more clear to me and I am now more confident that the overall theoretical explanations are correct and in-line with the experimental data.

Overall, the manuscript is now in a form that can be published in Nature communications. Nevertheless, I would encourage the authors to consider the following points before publication (no more referring from my side needed, though).

Major remarks on the interpretation:

- Concerning my remark about the transitions to 1D -> 2D -> 3D CDW regimes, the authors make it clear that what I believed to be the 2D regime should be rather considered a "fluctuation regime" before the "full locking" in alternating planes along the c-axis sets in. This makes sense, but leaves nevertheless a questions concerning the assignment of dimensionalities of the different CDW regimes. If the first regime (below 335 K) is a 1D CDW (corresponding to the wave vector $q_1=(0.4,0,0)$ of the unstable mode, then the wave vector $q_2 = (0,4, 0, 0.5)$ which represents the next phase transition (after crossing the "fluctuation regime") should point to a 2D CDW and not a 3D CDW.

- The introduction contains a paragraph about amplitudons and phasons (not "phasors" as written in the text). What is the connection to the later described 1D and 3D (or 2D) CDW regimes? Since those correspond to a rather short wave vector, I do not see the usefulness of describing them in terms of amplitude or phase variations. They seem to correspond to distinct eigenvectors, but this remains somewhat unclear, because the drawings of the eigenvectors in Figs. 3d and 3e are not very clear.

Some minor details:

- abstract, line 11: "CDW phase in dimension" -> "... in one dimension" (?)

- supplementary material, S6, choice of unit cells: if the k-point grid of the primitive unit cell is $(20 \times 16 \times 8)$, then the one of the 1D CDW should be $(4 \times 16 \times 8)$, multiplying the cell in real space by the factor 5 and dividing consequently in reciprocal space by 5 in x-direction. The k-point grid of the 3D CDW should then be $(4 \times 16 \times 4)$, dividing the number of k-points in c-direction by 2. The order of the latter two k-point grids seems to be reversed in the current version of the SM.

Responses to Reviewer #2:

We appreciate the Reviewer's time and comments. The detailed responses to the Reviewer's concern are provided as follows.

(1) Reviewer's comments:

About the "conception" of the "3D CDW" questioned by review #2 and review #3, now the authors use the scenario in the textbook (Fig. R2) to illustrate the "3D CDW" proposed in the manuscript. However, the physical interpretation and experimental finding of this work essentially bring little new understanding to CDW physics, and lacks scientific importance. Besides, while CDW along a - and c -axes are seen by diffraction technique, I'm still not convinced of the CDW along b -axis given only the ultrafast pump probe measurements. Also in the referenced paper [Phys. Rev. B 105, 115102 (2022)], in-plane optical conductivity clearly shows that the CDW feature is absent in the b direction, this is against the authors claim of CDW along b^* . How should the authors explain the contradiction?

Responses:

As the argument in Ref. [1], the optical conductivity σ_1 spectra along the a -axis in Fig. R1(a) show the feature of CDW mean-field gap opening below T_{CDW} , and there is no similar feature of CDW mean-field opening along the b -axis (see Fig. R1(b)). However, we believe it is not the whole story. If we enlarged the optical conductivity σ_1 spectra along the b -axis in Fig. R1(b), one could clearly observe a significant suppression of optical conductivity σ_1 spectra in the region of 1000 - 4000 cm^{-1} and the enhancement in the region of 7000 - 8000 cm^{-1} as decreasing temperatures (see Fig. R1(d)) due to the density of states (DOS) reduction around Fermi surface (E_F). Similar results have also been observed in $2H\text{-TaSe}_2$ [2], as shown in Fig. R1(c), which led to the pseudogap-like feature at ~ 0.07 eV and was associated with fluctuation effects with short-range CDW ordering. Actually, this gap anisotropy has been directly visualized in ARPES images of CuTe [3], as shown in Fig. R2. If the k_y is fixed at 0.3 \AA^{-1} , there are completely no states around E_F and along k_x (see the dashed square in Fig. R2(d)), i.e., along the a -axis in real space. Meanwhile, if the k_x is fixed at 0.4 \AA^{-1} , there are still some states (close to small k_y) around E_F and along k_y (see the dashed square in Fig. R2(h)), i.e., along the b -axis in real space, which is not a full gap as observed along the a -axis in Fig. R2(d).

The dispersion along the line segment suggests that this band still has some extent of 2D characteristics caused by the hybridization of the Te p_x orbital with other orbitals.

Fig. R1. The optical conductivity σ_1 at different temperatures along the (a) a -axis and (b) b -axis, respectively. [1] The inset of (a) displays $\sigma_1(\omega)$ along the a -axis at 10 K together with a Drude-Lorentz fit. (c) The optical conductivity σ_1 of $2H$ -TaSe $_2$ at relevant temperatures below and above the CDW phase transitions at 122 K and 90 K, respectively. The inset enlarges the frequency range below 0.7 and 0.2 eV. [2] (d) The enlarged optical conductivity σ_1 spectra in (b) from 500 to 8500 cm^{-1} .

Once the reduction of DOS appears around E_F , the relaxation channels are less effective in increasing the relaxation time of photoexcited electrons. As shown in Fig. R3 (marked by a black arrow), we do observe the significant increase of relaxation time $\tau_{e,b}$ of photoexcited electrons along the b -axis at ~ 270 K due to the suppression of optical conductivity σ_1 spectra from 1000 to 4000 cm^{-1} in Figs. R1(b) and (d), i.e., the reduction of DOS around E_F . Moreover,

the giant divergence of $\tau_{e,a}$ (along the a -axis, see Fig. R3 marked by a black arrow) is caused by the opening of the mean-field gap, i.e., full reduction of DOS around E_F (see Fig. R1(a) and ARPES data in Fig. S5) and has been unambiguously demonstrated by the theory of Kabanov *et al.* [4] with the mean-field gap as the fitting in Fig. 3c. Compared with the strongly temperature-dependent $\tau_{e,a}$, the weak temperature-dependent $\tau_{e,b}$ cannot be well described by the theory of Kabanov *et al.* with mean-field gap. This result is also consistent with the pseudogap-like feature in $2H\text{-TaSe}_2$ [2]. Not only in CDW materials, the coexistence of superconducting (mean-field) gap and pseudogap along different orientations has also been observed in superconductors, e.g., YBCO [5].

Fig. R2. (a)-(g) Band dispersions along the k_x direction at fixed $k_y = 0, 0.15, 0.25, 0.3, 0.45, 0.55, 0.77 \text{ \AA}^{-1}$, respectively. The CDW gap size is labeled. (h) Band dispersion along the quasi-1D line segment at $k_x=0.4 \text{ \AA}^{-1}$. The locations of (a)-(h) momentum cuts are marked by blue dashed lines in (i). Gap sizes extracted from (a)-(g) are appended as symbols. (j) Extracted gap size as a function of k_y . The error bars are smaller than the markers. (k)-(o) Evolution of band dispersions at $k_y=0.47 \text{ \AA}^{-1}$ with temperature from 20 K to 350 K with a photon energy of 60 eV. (p) Extracted gap size as a function of temperature. The gray dashed line is the fitting curve using a BCS-type gap equation. [3]

In summary, the anisotropic dynamics along the a -axis and b -axis in our orientation-/time-resolved ultrafast spectra are in good agreement with the results in Ref. [1]. Namely, a new 5-Te-atoms modulation (a spatial modulation of $q_{\text{CDW}}=0.4a^*$, see STM image in Fig. 3f) forms a CDW mean-field gap (see Fig. R1(a) and Fig. S5) along the a -axis. Even though there is no extra modulation (only the original lattice periodicity; see STM image in Fig. 3f), the interchain coupling of CDW chains along the a -axis does cause the reduction of DOS around E_F and shows the pseudogap-like feature along the b -axis. This result is just one of the novelties of our studies in the CuTe system, which was never reported in the literature, especially for the anisotropic dynamics along various dimensions. Therefore, this paper does show a new understanding of CDW physics and its scientific importance.

Above discussion has been added to lines 10-13 on Page 13 of this revised manuscript.

Fig. R3. The relaxation time τ_e of photoexcited electrons along the a -axis (and b -axis) as a function of temperature and obtained by fitting the $\Delta R/R$ spectra in Fig. 2a (and Fig. S3a) with Eq. (1). Solid line shows the fitting of Eq. (28) in Ref. [4]. (*Figure 3c in the revised manuscript has been updated by this figure.*)

(2) Reviewer's comments:

About the authors' explanation of the frequency shift described in Fig. R3c: Optical phonons have quadratic dispersion near Brillouin zone center, which can be approximated as a constant at small wavevector. Since the wavevector of probe light is very small compared to the crystal's reciprocal wavevector, the measured phonon frequency should be nearly the same, as in most pump probe measurements where the optical phonons oscillate at frequencies independent of the propagation direction k of probe light [e.g. J. Chem. Phys. 120, 4755-4758 (2004)]. Under the authors' scenario, the phonon should disperse sharply at small k , what is the reason for the unusual sharp dispersion? The author should give an explanation.

Responses:

A close look at the data in Fig. R4(a), we found that the data points at small incident angles of $\theta < 30^\circ$ are almost the same frequency, which is consistent with the results of Ref. [6] as the reviewer mentioned. When the θ increases to 45° , however, the frequency drops significantly from ~ 2.3 THz to ~ 1.97 THz, which is different from the results of Ref. [6]. In order to figure out this issue, we further enlarged the DFPT calculated phonon spectra of CuTe in the CDW_c phase around Γ point in Fig. S9, as shown in the inset of Fig. R4(a). Actually, there are many phonon bands near Γ point, including A_g , B_{1g} , B_{2g} , B_{3g} , B_{1u} , B_{2u} , and B_{3u} modes (red/green curves in the inset of Fig. R4(a)). But only the A_g mode with the c -axis components of eigenvectors (i.e., \mathbf{P}_c , see Figs. R4(b)-(d)) can be detected by our linear-polarized probe beam according to $\Delta R/R_c \propto \mathbf{E} \cdot \mathbf{P}_c$. Besides, the $A_{g,1}$ mode has the largest \mathbf{P}_c . Therefore, the $A_{g,1}$ mode with a frequency of ~ 2.39 THz would dominate the $\Delta R/R_c$ spectra at small incident angles with a small \mathbf{k}_1 vector of the probe beam, e.g., the dashed line (photon dispersion) in the inset of Fig. R4(a). For the larger incident angles, the photon dispersion line (e.g., dot-dashed line in the inset of Fig. R4(a)) may cross with $A_{g,2}$ and $A_{g,3}$ modes but not $A_{g,1}$ mode. However, the c -axis components of $A_{g,2}$ mode eigenvectors are smaller than those of $A_{g,3}$ mode, as shown in Figs. R4(c) and (d). Consequently, the $A_{g,3}$ mode with a frequency of 1.93 THz could be detected, which agrees with our data point (~ 1.97 THz) at 45° .

In summary, the dramatic shrink of mode frequency at larger incident angles θ is possibly caused by probing the lower frequency modes. This also indicates that our orientation-/time-

resolved ultrafast spectroscopy can reveal the CDW dynamics along the c -axis, which is one of the novelties of our studies in the CuTe system and never reported in the literature. Therefore, this paper definitely shows the new understanding of CDW physics in various dimensions and its scientific importance.

Section S7 in the Supplementary Information has been updated with the above discussion.

Fig. R4. (a) Frequency of A_{HF} peak in Fig. 4b as a function of incident angles θ of the probe pulses. Inset: Enlargement of the DFPT calculated phonon spectrum of CuTe in CDW_c phase around Γ point in Fig. S9 (in Supplementary Information). Red curves indicate the dispersion of $A_{g,1}$, $A_{g,2}$, and $A_{g,3}$ modes. Green curves show the B_{1g} , B_{2g} , B_{3g} , B_{1u} , B_{2u} , and B_{3u} modes, respectively. Red dashed and dot-dashed lines are the photon dispersion. (b), (c), (d) Calculated eigenvectors of $A_{g,1}$, $A_{g,2}$, and $A_{g,3}$ modes in the inset of (a). Copper and tellurium atoms are denoted as blue and golden spheres, where red arrows depict atomic displacements. (*Figure S12 in the revised Supplementary Information has been updated by this figure.*)

(3) Reviewer's comments:

Page 19 of the supplementary material writes: “If only one of the $\Delta R/R_a$, $\Delta R/R_b$, and $\Delta R/R_c$ along three orientations (dimensions) can be observed, it is just a kind of 1D CDW. If two of the $\Delta R/R_a$, $\Delta R/R_b$, and $\Delta R/R_c$ along three orientations (dimensions) can be observed, it should be a 2D CDW. Finally, if all the $\Delta R/R_a$, $\Delta R/R_b$, and $\Delta R/R_c$ along three orientations (dimensions) can be observed, it is a 3D CDW.” I disagree with this argument. In some other standard 1D CDW systems, the amplitude mode oscillations can be measured on multiple directions, see [Phys. Rev. B 107, 165115 (2023)], [Phys. Rev. Lett. 83, 800 (1999)].

Responses:

This is because the interchain coupling results in coherent oscillations in $\Delta R/R$ that can also be observed along the perpendicular direction of a CDW chain, not only the direction along a CDW chain. Take well-known quasi-1D $K_{0.3}MoO_3$ (blue bronze) as an example [6]; the coherent oscillations in ΔR can be observed along both parallel (probe polarization $E//b$) and perpendicular (probe polarization $E\perp b$) to the directions of CDW chains ($//b$), as shown in Fig. R5(a). The results of perpendicular ($E\perp b$) to the CDW chain have been demonstrated by the interchain coupling, which is a critical mechanism to further form the 3D CDW with long-range order in $K_{0.3}MoO_3$ [6, 7] and has been taken as an example in Grüner's book [Chapter 5 in 8]. However, $K_{0.33}MoO_3$ (red bronze) has a similar structure as $K_{0.3}MoO_3$ but no interchain coupling. Therefore, there are no coherent oscillations in ΔR when the probe polarization is perpendicular to the chain direction ($E\perp b$), as shown in Fig. R5(b). This indicates that the probe polarization can detect the ordering or order parameters exactly along the polarization direction in CDW materials.

In order to avoid confusion for readers, we have revised the sentences “If only one of the $\Delta R/R_a$, $\Delta R/R_b$, and $\Delta R/R_c$ along three orientations (dimensions) can be observed, it is just a kind of 1D CDW. If two of the $\Delta R/R_a$, $\Delta R/R_b$, and $\Delta R/R_c$ along three orientations (dimensions) can be observed, it should be a 2D CDW. Finally, if all the $\Delta R/R_a$, $\Delta R/R_b$, and $\Delta R/R_c$ along three orientations (dimensions) can be observed, it is a 3D CDW.” (*on page 19 of Supplementary Information*) to “This indicates that the orientation-/time-resolved ultrafast spectroscopy developed in this study can be used to reveal the ordering or order parameters in various orientations (dimensions) in materials.”. Additionally, the sentence “Additionally, the $\Delta R/R_a \propto \mathbf{E}\cdot\mathbf{q}_a$ and $\Delta R/R_b \propto \mathbf{E}\cdot\mathbf{q}_b$ can also be realized via the polarization rotation of probe pulses on ab -plane, respectively.” was moved to lines 10-12 on page 19.

Fig. R5. Transient reflectivity changes ΔR at 400 nm probe wavelength: **(a)** with the polarization vector E parallel ($E \parallel b$) and perpendicular ($E \perp b$) to the chain direction in $K_{0.3}MoO_3$ at 30 K. The inset shows the anisotropy of the oscillation period in the $a'-b$ plane. **(b)** Anisotropy of ΔR for different probe polarization angles in the $a'-b$ plane of $K_{0.33}MoO_3$ at 290 K. [6]

In summary, the authors are grateful for the reviewer's time and insightful comments. We have carefully addressed and clarified all points raised by the reviewer. The new manuscript version has been updated significantly according to the reviewer's comments. We really appreciate the reviewer's suggestions and desire your support for the publication in Nature Communications.

References

- [1] Li, R. S. et al. Optical spectroscopy and ultrafast pump-probe study of a quasi-one-dimensional charge density wave in CuTe. *Phys. Rev. B* **105**, 115102 (2022).
- [2] Vescoli, V., Degiorgi, L., Berger, H. & Forró, L. Dynamics of correlated two-dimensional materials: the 2H-TaSe₂ case. *Phys. Rev. Lett.* **81**, 453 (1998).
- [3] Zhang, K. et al. Evidence for a quasi-one-dimensional charge density wave in CuTe by angle-resolved photoemission spectroscopy. *Phys. Rev. Lett.* **121**, 206402 (2018).
- [4] Kabanov, V. V., Demsar, J., Podobnik, B. & Mihailovic, D. Quasiparticle relaxation dynamics in superconductors with different gap structures: theory and experiments on YBa₂Cu₃O_{7-δ}. *Phys. Rev. B* **59**, 1497 (1999).
- [5] Luo, C. W. et al. Spatial dichotomy of quasiparticle dynamics in underdoped thin-film YBa₂Cu₃O_{7-δ} superconductors. *Phys. Rev. B* **74**, 184525 (2006).
- [6] Ren, Y., Xu, Z., & Lüpke, G. Ultrafast collective dynamics in the charge-density-wave conductor K_{0.3}MoO₃. *J. Chem. Phys.* **120**, 4755-4758 (2004).
- [7] Demsar, J., Biljakovic, K., & Mihailovic, D. Single particle and collective excitations in the one-dimensional charge density wave solid K_{0.3}MoO₃ probed in real time by femtosecond spectroscopy. *Phys. Rev. Lett.* **83**, 800 (1999).
- [8] G. Grüner, Density waves in solid. *Frontiers in physics* Ed D. Pines (Addison-Wesley), p.99 (1994).

Responses to Reviewer #3:

We do appreciate the Reviewer's encouragement and support. The detailed responses to the Reviewer's additional suggestions are provided as follows.

(1) Reviewer's comments:

Concerning my remark about the transitions to 1D \rightarrow 2D \rightarrow 3D CDW regimes, the authors make it clear that what I believed to be the 2D regime should be rather considered a "fluctuation regime" before the "full locking" in alternating planes along the c -axis sets in. This makes sense, but leaves nevertheless a question concerning the assignment of dimensionalities of the different CDW regimes. If the first regime (below 335 K) is a 1D CDW (corresponding to the wave vector $q_1=(0.4,0,0)$ of the unstable mode, then the wave vector $q_2 = (0,4, 0, 0.5)$ which represents the next phase transition (after crossing the "fluctuation regime") should point to a 2D CDW and not a 3D CDW.

Responses:

Thank the reviewer's support and reminder. In order to avoid confusion for readers, we have revised all terms of 1D CDW (CDW_{1D}) and 3D CDW (CDW_{3D}) to CDW along the a -axis (CDW_a) and CDW along the c -axis (CDW_c), respectively, in manuscript and Supplementary Information.

(2) Reviewer's comments:

The introduction contains a paragraph about amplitudons and phasons (not "phasors" as written in the text). What is the connection to the later described 1D and 3D (or 2D) CDW regimes? Since those correspond to a rather shortwave vector, I do not see the usefulness of describing them in terms of amplitude or phase variations. They seem to correspond to distinct eigenvectors, but this remains somewhat unclear, because the drawings of the eigenvectors in Figs. 3d and 3e are not very clear.

Responses:

Thank you for the reviewer's correction and suggestions. First, the typo of "phasors" (*Lines 1 & 3 on Page 5 of this revised manuscript*) has been corrected. Second, the drawings of the eigenvectors in **Figs. 3d** and **3e** have been changed in this revised manuscript to be clear

for readers. Finally, the connection between amplitudons/phasons and the oscillation modes in 1D/3D (or 2D) CDW regimes has been added in *lines 1-2 on page 10* and *lines 5-6 on page 11 of this revised manuscript*. The detailed discussion can also be found in *the last paragraph of section S6 in Supplementary Information*.

(3) Reviewer's comments:

abstract, line 11: "CDW phase in dimension" -> "... in one dimension" (?)

Responses:

In order to avoid confusion for readers, the sentence "... the CDW phase in dimension until ..." has been revised to "... the CDW phase until ...". (*Line 11 in the Abstract of this revised manuscript*)

(4) Reviewer's comments:

supplementary material, S6, choice of unit cells: if the k-point grid of the primitive unit cell is (20x16x8), then the one of the 1D CDW should be (4x16x8), multiplying the cell in real space by the factor 5 and dividing consequently in reciprocal space by 5 in x-direction. The k-point grid of the 3D CDW should then be (4x16x4), dividing the number of k-points in c-direction by 2. The order of the latter two k-point grids seems to be reversed in the current version of the SM.

Responses:

Thank the reviewer's reminder. The typos of "CDW_{1D} (4×16×4), and CDW_{3D} (4×16×8)" have been corrected to "CDW_{1D} (4×16×8), and CDW_{3D} (4×16×4)" in this version. (*Line 14 on page 11 of Supplementary Information*)

List of changes in the revised manuscript

Here, let us briefly summarize the major points and changes as follows.

- (1) The co-author's affiliations have been revised respectively.
Author list on previous submission: Nguyen Nhat Quyen¹,..., Shih-Chang Weng⁴, Chang-Yang Kuo^{1,4}, ..., Chih-Wei Luo*^{1,4,9,10,11}
Author list on revised manuscript: Nguyen Nhat Quyen¹,..., Shih-Chang Weng⁴, **Cheng-Maw Cheng⁴**, Chang-Yang Kuo^{1,4}, ..., Chih-Wei Luo*^{1,4,9,10,11}
- (2) **lines 5 (page 3, Abstract of previous manuscript)** "...be studied, and the phase transition mechanism is currently moot." **was revised to** "...be studied, and **their** phase transition mechanism is currently moot."
- (3) **lines 8-9 (page 3, Abstract of previous manuscript)** "...CuTe is also demonstrated to drive the formation of one-dimensional (1D) CDW chain ... When temperature decreases, ..." **was revised to** "...CuTe is also demonstrated to drive the formation of **one-dimensional CDW** chain... When **the** temperature decreases, ..."
- (4) **lines 11-13 (page 3, Abstract of previous manuscript)** "...quantum fluctuations (QF) of the CDW phase in dimension until 220 K. At $T < 220$ K, ... in anti-phase to form a three-dimensional (3D) CDW phase." **was revised to** "...quantum fluctuations (QF) of the **CDW phase until 220 K**. At $T < 220$ K, ... in anti-phase to form **a CDW phase along the c-axis**."
- (5) **lines 1 (page 5 of the previous manuscript)** "... (so-called Nambu-Goldstone mode or phasor)" **was revised to** "... (so-called Nambu-Goldstone mode or **phason**)"
- (6) **lines 3 (page 5 of the previous manuscript)** "... (phasor is associated with ...)" **was revised to** "... (**phason** is associated with ...)"
- (7) **lines 14-15 (page 5 of the previous manuscript)** "...features near T_c of 335 K,²² which show electron-phonon coupling in CuTe²³ (see Supplementary section S6). Using density functional..." **was revised to** "...features near **$T_c=335$ K**,²² which show electron-phonon coupling in **CuTe²³**. Using density functional..."
- (8) **lines 10-11 (page 6 of the previous manuscript)** "...the dynamics of collective modes (amplitudon, phason) in CDW materials^{4,30-39} (see Supplementary section S1)." **was revised to** "...the dynamics of **collective modes in** CDW materials^{4,30-39} (see Supplementary section S1)."
- (9) **lines 12 (page 6 of the previous manuscript)** "This study uses orientation- and time-resolved ultrafast spectroscopy in (001) CuTe..." **was revised to** "This study uses **orientation-/time-resolved ultrafast spectroscopy (otrUS)** in (001) CuTe..."

- (10) *lines 19 (page 6 of the previous manuscript) and lines 1 (page 7 of the previous manuscript)* “...Moreover, a 3D CDW was first observed at low temperatures using orientation-/time-resolved ultrafast spectroscopy and...” *was revised to* “...Moreover, **a CDW along the c -axis** was first observed at low temperatures using **otrUS** and...”
- (11) *lines 10 (page 7 of the previous manuscript)* “...physical characteristics associated with the changes of Fermi surface. Since electrons...” *was revised to* “...physical characteristics associated with the changes of E_F . Since electrons...”
- (12) *lines 1 (page 10 of the previous manuscript)* “...by first-principles calculations (see Fig. 3d) and...” *was revised to* “...by first-principles calculations (see Fig. 3d & Supplementary section S1/S6, the arrows indicate the motion of atoms to form the amplitudon) and...”
- (13) *lines 1-2 (page 11 of the previous manuscript)* “.... It demonstrates a 2D CDW formed via the interchain coupling, as verified by the pump-probe spectroscopy along the b -axis in Fig. 3a.” *was revised to* “...It demonstrates **a CDW plane** formed via the interchain coupling, as verified by the **otrUS** along the b -axis in Fig. 3a”
- (14) *lines 4 (page 11 of the previous manuscript)* “.... new A_g (A_{ZF1}) mode (see Fig. 3e and Fig. S4), for which...” *was revised to* “...new A_g (A_{ZF1}) mode (see Fig. 3e, Fig. S4 & Supplementary section S6, the arrows indicate the motion of atoms to form the amplitudon), for which...”
- (15) *lines 7-9 (page 11 of the previous manuscript)* “...two adjacent layers along the c -axis to form a 3D CDW (CDW_{3D}) at $T < 220$ K ... transition from CDW_{1D} to CDW_{3D} is also accompanied by a reduction...” *was revised to* “...two adjacent layers along the c -axis to form **a CDW along the c -axis (CDW_c)** at $T < 220$ K ... transition from **a CDW along the a -axis (CDW_a) to CDW_c** is also accompanied by a reduction...”
- (16) *lines 15 (page 11 of the previous manuscript)* “...establish a more stable 3D CDW at $T < 220$ K...” *was revised to* “...establish a more **stable CDW** at $T < 220$ K...”
- (17) *lines 17-18 (page 11 of the previous manuscript) and lines 1 (page 12 of the previous manuscript)* “... (from 1D to 3D via the formation of 2D) in the intermediate temperature ... CuTe then forms a stable 3D CDW...” *was revised to* “... (from **CDW_a** to **CDW_c** via the formation of **a CDW plane on ab -plane**) in the intermediate temperature ... CuTe then forms a stable **CDW_c** ...”
- (18) *lines 10 (page 12 of the previous manuscript)* “...(with a bare critical phonon frequency

of $\omega_0=2.2$ THz, $2\Delta=900$ meV) in LaTe₃...” *was revised to* “...(with a bare critical phonon frequency of $\omega_0=2.2$ THz) in LaTe₃...”

- (19) *lines 19 (page 12 of the previous manuscript)* “... The value for τ_e suddenly increases...” *was revised to* “...The value for $\tau_{e,a}$ along the *a*-axis suddenly increases...”
- (20) *lines 2 (page 13 of the previous manuscript)* “... the formation of CDW_{1D} (or CDW_{3D})...” *was revised to* “...the formation of CDW_{*a*} (or CDW_{*c*})...”
- (21) *lines 4 (page 13 of the previous manuscript)* “... which the resistivity is greatest, the τ_e ...” *was revised to* “...which the resistivity is greatest, the $\tau_{e,a}$...”
- (22) *lines 7 (page 13 of the previous manuscript)* “...the temperature-dependent τ_e ...” *was revised to* “...the temperature-dependent $\tau_{e,a}$...”
- (23) *lines 8-10 (page 13 of the previous manuscript)* “...The temperature-dependent gap size that is obtained by pump-probe spectroscopy is consistent with the results for angle-resolved photoemission spectroscopy (ARPES) (see Supplementary section S4).” *was revised to* “...The temperature-dependent gap size that is obtained by **otrUS** is consistent with the results for angle-resolved photoemission spectroscopy (ARPES) (see Supplementary section S4). **Additionally, the reduction of the density of states around E_F and along the *b*-axis²⁶ (pseudogap-like feature⁴), caused by a certain ordering due to the interchain coupling of CDW chains along the *a*-axis, further results in a slight increase of $\tau_{e,b}$ (along the *b*-axis) around 270 K.**”
- (24) *lines 11 (page 13 of the previous manuscript)* “Electronic-driven phase transition and 3D-CDW orders...” *was revised to* “Electronic-driven phase transition and **CDW order along the *c*-axis**...”
- (25) *lines 12 (page 13 of the previous manuscript)* “phase transition temperature indicated by τ_e is much...” *was revised to* “...phase transition temperature indicated by $\tau_{e,a}$ is much...”
- (26) *lines 18 (page 13 of the previous manuscript)* “...and the high-temperature CDW_{1D}...” *was revised to* “...the high-temperature **CDW_{*a*}**...”
- (27) *lines 3 (page 14 of the previous manuscript)* “...CDW_{3D} is stabilized due to...” *was revised to* “...**CDW_{*c*}** is stabilized due to...”
- (28) *lines 4 (page 14 of the previous manuscript)* “...**Fig. 3a** is associated with the CDW_{3D}, ...” *was revised to* “...**Fig. 3a** is associated with the **CDW_{*c*}**, ...”

- (29) *lines 1 (page 15 of the previous manuscript)* “The CDW_{3D} structure of CuTe...” *was revised to* “The CDW_c structure of CuTe...”
- (30) *lines 2 (page 15 of the previous manuscript)* “...in anti-phase to establish a 3D CDW, ...” *was revised to* “...in anti-phase to establish a CDW_c, ...”
- (31) *lines 5 (page 15 of the previous manuscript)* “...the modulated structure in the 3D CDW with...” *was revised to* “...the modulated structure in the CDW_c with...”
- (32) *lines 9 (page 15 of the previous manuscript)* “...At 335 K, a 1D CDW along...” *was revised to* “At 335 K, a CDW_a along...”
- (33) *lines 11-12 (page 15 of the previous manuscript)* “...in dimension (from 1D to 3D accompanying the formation of 2D CDW on *ab*-plane) until 220 K^{18,21}...” *was revised to* “...in dimension (from CDW_a to CDW_c accompanying the formation of a CDW on *ab*-plane) until 220 K^{18,21}...”
- (34) *lines 14 (page 15 of the previous manuscript)* “...form a 3D CDW¹⁸ due to locking...” *was revised to* “...form a CDW_c¹⁸ due to locking...”
- (35) *lines 14 (page 18 of the previous manuscript)* “...CDW modulations, CDW_{1D} and CDW_{3D}, guided by...” *was revised to* “...CDW modulations, CDW_a and CDW_c, guided by...”
- (36) *lines 18-19 (page 18 of the previous manuscript)* “...a primitive unit cell (20×16×8), CDW_{1D} (4×16×4), and CDW_{3D} (4×16×8) were set to perform corresponding BZ integrations...” *was revised to* “...a primitive unit cell (20×16×8), CDW_a (4×16×8), and CDW_c (4×16×4) were set to perform corresponding BZ integrations...”
- (37) *lines 1 (page 19 of the previous manuscript)* “...phonon dispersions of the normal, CDW_{1D}, and CDW_{3D} phases...” *was revised to* “...phonon dispersions of the normal, CDW_a, and CDW_c phases...”
- (38) The description of **Author contributions** (*lines 1-9 on page 20*) *was revised* as follows, “C.W.L. proposed the project. N.N.Q., ... performed the measurements of STM. C.Y.K., C.E.L., Y.S.C., S.C.W, and C.M.C. performed the measurements of XRD. ... All authors edited and approved the final manuscript.”
- (39) A reference was changed in this revised manuscript as follows,

“4. Demsar, J., Forró, L., Berger, H. & Mihailovic, D. Femtosecond snapshots of gap-forming charge-density-wave correlations in quasi-two-dimensional dichalcogenides 1T-TaS₂ and 2H-TaSe₂. *Phys. Rev. B* **66**, 041101R (2002).” *was changed to*

“4. Vescoli, V., Degiorgi, L., Berger, H. & Forró, L. Dynamics of correlated two-dimensional materials: the 2H-TaSe₂ case. *Phys. Rev. Lett.* **81**, 453 (1998).”

List of changes on Figure captions in the revised manuscript

The caption of **Fig. 3** *was revised* as follows,

(40) *lines 5 (page 31 of the previous manuscript)* “...**Fig. 2a** with Eq. (1). CDW gap size as...” *was revised to* “...**Fig. 2a (Fig S3a for b-axis)** with Eq. (1). CDW gap size as...”

(41) *lines 8-9 (page 31 of the previous manuscript)* “...Modulated structure in the one-dimensional CDW (CDW_{1D}) and three-dimensional CDW (CDW_{3D}) that is proposed by...” *was revised to* “...Modulated structure in the **CDW along the a-axis (CDW_a)** and **CDW along the c-axis (CDW_c)** that is proposed by...”

The caption of **Fig. 4** *was revised* as follows,

(42) *lines 3 (page 32 of the previous manuscript)* The description of “**Fig. 4 Ultrafast dynamics along the c-axis of CuTe and schematics of a 3D CDW.**” *was revised to* “**Fig. 4 Ultrafast dynamics along the c-axis of CuTe and schematics of a CDW in various dimensions.**”

(43) *lines 7 (page 32 of the previous manuscript)* “...Schematics of 3D CDW for which two layers along...” *was revised to* “... Schematics of **a CDW in various dimensions** for which two layers along...”

(44) *lines 8 (page 32 of the previous manuscript) and lines 1 (page 33 of the previous manuscript)* “...Modulated structure of the 3D CDW with charge density difference between CDW_{3D} and non-CDW...” *was revised to* “...Modulated structure of the **CDW in c** with charge density difference between **CDW_c** and non-CDW...”